# In vivo HIV-1 nuclear condensates safeguard against cGAS and license reverse transcription

Selen Ay[1,7], Julien Burlaud-Gaillard[2,3,7], Anastasia Gazi [4,7], Yevgeniy Tatirovsky[5,6], Celine Cuche[1], Jean-Sebastien Diana[1], Viviana Scoca[1], James P Di Santo [5], Philippe Roingeard [2,3], Fabrizio Mammano [2] & Francesca Di Nunzio [1]✉

## Abstract

Entry of viral capsids into the nucleus induces the formation of biomolecular condensates called HIV-1 membraneless organelles (HIV-1-MLOs). Several questions remain about their persistence, in vivo formation, composition, and function. Our study reveals that HIV-1-MLOs persisted for several weeks in infected cells, and their abundance correlated with viral infectivity. Using an appropriate animal model, we show that HIV-1-MLOs were formed in vivo during acute infection. To explore the viral structures present within these biomolecular condensates, we used a combination of double immunogold labeling, electron microscopy and tomography, and unveiled a diverse array of viral core structures. Our functional analyses showed that HIV-1-MLOs remained stable during treatment with a reverse transcriptase inhibitor, maintaining the virus in a dormant state. Drug withdrawal restored reverse transcription, promoting efficient virus replication akin to that observed in latently infected patients on antiretroviral therapy. However, when HIV-1 MLOs were deliberately disassembled by pharmacological treatment, we observed a complete loss of viral infectivity. Our findings show that HIV-1 MLOs shield the final reverse transcription product from host immune detection.

**Keywords** HIV-1 Cores; Post-nuclear Entry Steps; Innate Immunity; Nuclear Reverse Transcription; Biomolecular Condensates
**Subject Categories** Microbiology, Virology & Host Pathogen Interaction; Organelles

## Introduction

Biomolecular condensates play a pivotal role in facilitating the replication cycle of RNA viruses (Alberti et al, 2019; Cubuk et al, 2021; Di Nunzio, 2023; Iserman et al, 2020; Nevers et al, 2020; Perdikari et al, 2020; Risso-Ballester et al, 2021; Savastano et al, 2020; Scoca et al, 2022; Zhang et al, 2023). Viral RNA molecules can persist for a long time during chronic viral infection and stress-induced condensates reactivate dormant viruses (Di Nunzio, 2023; Fang et al, 2023; Zhang et al, 2023). Retroviruses, such as the Human Immunodeficiency Virus (HIV-1), need to reverse transcribe their RNA genome into DNA for integration into the host chromatin to replicate (Flint et al, 2020). Recent studies have challenged the traditional view that the only genetic material of HIV-1 entering the nucleus is the reverse transcribed DNA (Burdick et al, 2020; Dharan et al, 2020; Rensen et al, 2021; Scoca et al, 2022; Selyutina et al, 2020). Instead, incoming viral RNA genomes accumulate in nuclear niches (Rensen et al, 2021) (referred to as HIV-1-membraneless organelles or HIV-1-MLOs) triggered by the virus to navigate the nuclear space (Scoca et al, 2022). They eventually reach active sites within the host chromatin, promoting the generation of an active provirus (Li et al, 2021; Scoca et al, 2022). The establishment of HIV-1-MLOs is dependent on the interaction between the viral capsid (CA) and the Cleavage Polyadenylation Subfactor 6 (CPSF6), a component of the Pre-Messenger RNA Cleavage Factor I (CFIm) complex (Buffone et al, 2018; Lee et al, 2010; Price et al, 2012; Wei et al, 2022), which likely favors oligomerization of CPSF6 through multivalent interactions between its disordered regions, leading to the formation of condensates (Ay and Di Nunzio, 2023; Di Nunzio et al, 2023). Furthermore, it is unknown whether these structures form in in vivo settings and whether they serve as exclusive sites for nuclear reverse transcription.

Here, we studied the viral conformations hosted inside these structures and the role of HIV-1-MLOs in helping the final product of the reverse transcription (RT), the double-stranded viral DNA (dsDNA), to evade the innate immune response (Cingoz and Goff, 2019), enabling the virus to evolve into a pandemic strain capable of persistent host infection (Sandler et al, 2014; Zuliani-Alvarez et al, 2022). Understanding these early infection steps is critical, as they can significantly contribute to the establishment of viral reservoirs that are seeded shortly after the initial infection (Luzuriaga et al, 2015), representing the primary obstacle to finding a cure.

[1]Institut Pasteur, Advanced Molecular Virology Unit, Department of Virology, Université Paris Cité, 75015 Paris, France. [2]Inserm U1259 MAVIVHe, Université de Tours and CHRU de Tours, Tours, France. [3]Plate-Forme IBiSA de Microscopie Electronique, Université de Tours and CHRU de Tours, Tours, France. [4]Institut Pasteur, Université Paris Cité, Ultrastructural Biolmaging Facility, 75015 Paris, France. [5]Innate Immunity Unit, Institut Pasteur, Université Paris Cité, Inserm U1223, Paris, France. [6]Vaccine Research Institute, Université Paris Est, Inserm U955, Créteil, France. [7]These authors contributed equally: Selen Ay, Julien Burlaud-Gaillard, Anastasia Gazi. ✉E-mail: dinunzio@pasteur.fr

# Results

## HIV-1 nuclear condensates persist over time during reverse transcription inhibitor treatment

Virus-induced nuclear condensates are typically formed in the early stages of infection (Francis et al, 2020; Luchsinger et al, 2023; Rensen et al, 2021; Scoca et al, 2022), but how long they can persist in infected cells is unclear.

To preserve the stability of the nuclear HIV-1-MLOs and prevent their loss during cell division, we chose to infect macrophage-like cells (THP-1), which are non-dividing cells supporting low viral replication. However, these cells are also less susceptible to HIV-1 infection than active lymphocytes due to the presence of restriction factors such as SAMHD1 and a lower amount of dNTPs, as previously reported (Laguette et al, 2011). Due to their low susceptibility to infection, we infected these cells with a high multiplicity of infection (MOI), and maintained them in culture for 25 days. To block reverse transcription and potential uncoating, which would induce the disassembly of the HIV-1-MLOs, we treated the THP-1 cells with nevirapine (NEV) for the entire duration of the experiment. Immunofluorescence microscopy was conducted using antibodies against the endogenous CPSF6 and the viral integrase (IN) that contains an HA tag (Blanco-Rodriguez et al, 2020; Petit et al, 2000), which revealed the presence of nuclear HIV-1-MLOs (Scoca et al, 2022), characterized by the clustering of CPSF6 and viral components, inside the host nucleus for up to 25 days post-infection (d.p.i.) (Fig. 1A). Although they may persist for even longer, we stopped the experiment at 25 days, when we detected initial signs of cell suffering. We revealed the majority of CPSF6 clusters or nuclear HIV-1-MLOs at 4 d.p.i. (avg. of CPSF6 clusters/cell 2.74), and there was no significant difference in the detection of CPSF6 clusters from the 11th d.p.i. until the last day of the experiment (avg. CPSF6 clusters/cell were 1.36, 1.10, and 0.94 at 11, 18, and 25 d.p.i., respectively) (Fig. 1B). However, the number of analyzed cells decreased between day 4 ($n$ = 107) and day 11 post-infection and remained similar until the 25th d.p.i. ($n$ = 33, $n$ = 40, $n$ = 33) (Fig. 1B). This decrease in the number of cells may be due to the death of highly infected cells that carry more CPSF6 clusters. In most of the detected clusters, the IN was present (Fig. 1C). Importantly, no clustering of CPSF6 was detected in uninfected cells (Fig. 1A–C). Taken together, our data demonstrate that these biomolecular condensates endure for an extended period of time in non-dividing cells during treatment with NEV, a component of the current antiretroviral therapy.

## The HIV-1 nuclear condensates a beacon of functional infection

HIV-1-MLOs are generated after viral nuclear entry (Scoca et al, 2022), here we evaluate their viral RNA (vRNA) composition and their role in the progression of viral life cycle. We characterized HIV-1-MLOs vRNA content by using two different imaging tools, immune RNA-FISH (Rensen et al, 2021; Scoca et al, 2022) and the MCP-MS2 system (Tantale et al, 2016) (Fig. 2A), in infected macrophage-like cells treated or not with NEV. Using RNA-FISH, in NEV-treated cells, where only incoming vRNA is present, the viral genome was exclusively detected inside CPSF6 clusters

(Fig. 2B). In untreated cells, instead, vRNA puncta also appear outside CPSF6 clusters, presumably representing vRNA transcriptional foci. Thus, RNA-FISH detects the total vRNA (in average ~5 events per cell), but the incoming vRNA can be distinguished from vRNA in transcriptional foci based on their nuclear spatial location (Fig. 2B,C; Appendix Fig. S1A). To corroborate these data we applied another imaging tool, MCP-MS2 system, designed to specifically detect the RNA (Bertrand et al, 1998). Interestingly, MCP-MS2 was able to form bright puncta only with the vRNA located outside of CPSF6 clusters, corresponding to newly transcribed vRNA (Scoca et al, 2022), which represents a smaller population of the total vRNA foci (avg. 2 events/cell) (Fig. 2B,C; Appendix Fig. S1A). These data suggest that the incoming vRNA is shielded by viral structures, rendering it inaccessible to MCP-GFP, despite its random distribution in the nucleus and within CPSF6 clusters (Fig. 2B,C; Movies EV1 and EV2). However, it can be labeled by RNA-FISH since samples are permeabilized, allowing probe penetration. In more detail, 50% of the vRNA foci detected by RNA-FISH associate with CPSF6 clusters, while approximately 45% are detected outside of CPSF6 clusters, and associate with MCP-GFP forming bright puncta. About 4% of the vRNA foci did not associate with either CPSF6 or MCP-GFP puncta (Fig. 2C). This latter population could represent incoming viral genomes just prior to be enrolled to establish CPSF6 condensates, or newly generated full-length vRNA genomes that are protected by proteins that obstruct MCP-GFP from binding to the MS2 loops. Further analysis revealed that 100% of untreated cells that presented MCP-GFP puncta were also positive for CPSF6 clusters (Fig. 2D; Appendix Fig. S1A), suggesting that viral transcribed RNA is a consequence of prior CPSF6 clustering. Next, we correlated the presence of CPSF6 clusters with the efficiency of infection in single cells (Fig. 2E–G; Appendix Fig. S1B).

We observe that only a few cells were productively infected (24.14%) using an MOI of 10 (Fig. 2G), as they expressed the reporter gene GFP, which is carried by the HIV-1 genome. Interestingly, we observe a positive correlation between incoming vRNA genome located in CPSF6 clusters and viral transcription foci, which localize outside CPSF6 clusters (Scoca et al, 2022) ($P$-value < 0.0001, R = 0.5238) and between the percentage of cells presenting CPSF6 clusters and GFP-positive cells (more than 80% of GFP + cells show CPSF6 clusters) ($P$-value < 0.0001, R = 0.7514) (Fig. 2F,G), again supporting the association between the detection of incoming viral RNA (CPSF6 associated) and de novo viral RNA transcription. The error bars represent the 95% confidence interval of the simple linear regression curve calculated using Prism. Importantly, macrophages, exhibit a slower rate of viral infection compared to activated CD4+ T cells (Koppensteiner et al, 2012). This characteristic may contribute to the observation that approximately 70% of GFP-negative cells harbor CPSF6 clusters (Fig. 2G). These clusters likely represent pre-integration viral reservoirs within HIV condensates that can persist for extended periods, as shown in Fig. 1. Despite the persistence of CPSF6 clusters within these cells, our findings reveal that only a small fraction (~24%), mainly carrying CPSF6 clusters, progresses to a productive infection, as indicated in Fig. 2G.

Taken together, our data indicate that the presence of incoming viral RNA genomes clustered in HIV-1-MLOs is established prior to RT and integration as a pre-requisite for canonical post-nuclear RT events.

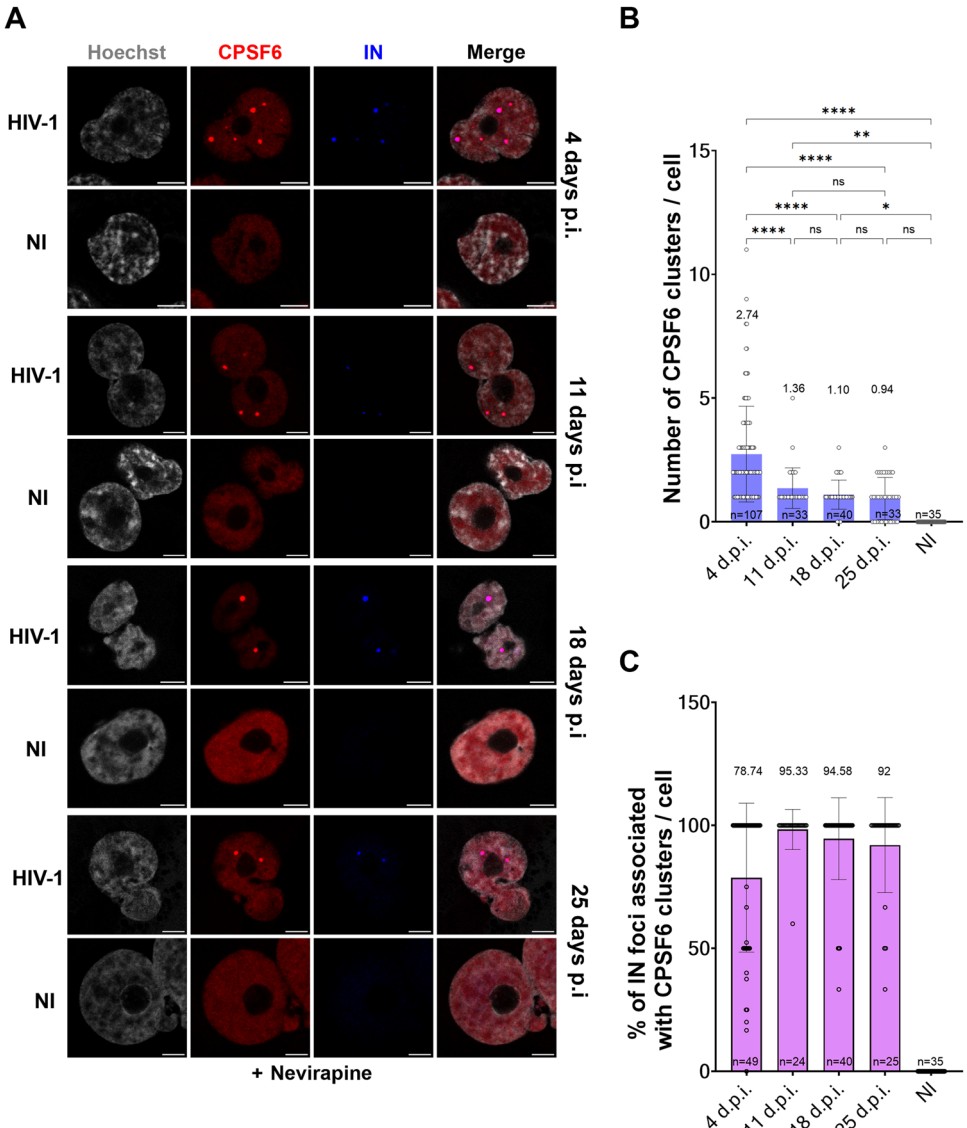

**Figure 1. HIV-1-MLOs persist in macrophage-like cells in presence of nevirapine.**

(A) THP1 cells differentiated with phorbol 12-myristate 13-acetate (PMA) and treated with Nevirapine (NEV) were infected with HIV-1 ΔEnv pseudotyped with VSV-G and carrying the IN with an HA tag (MOI 10). At different times post infection (p.i.), cells were fixed and labeled with antibodies to detect CPSF6 (in red) and IN (in blue). Nuclei were stained with Hoechst (in gray). Scale bar = 5 µm. (B) The histogram shows the quantification of the number of CPSF6 clusters per cell at different times p.i. Data from two independent experiments are shown as the mean ± SD (Ordinary one-way ANOVA test; ****: (4 d.p.i. vs 11 d.p.i. *p*-value = 0.0000149), (4 d.p.i. vs 18 d.p.i. *p*-value = 0.000000013), (4 d.p.i. vs 25 d.p.i. *p*-value = 0.00000004146790), 4 d.p.i. vs N.I. *p*-value = 0.000000000000277); **: *p*-value 0.0031; *: *p*-value 0.0216; ns (not significant *p*-value >0.05): (11 d.p.i. vs 18 d.p.i. *p*-value = 0.929), 11 d.p.i. vs 25 d.p.i. *p*-value = 0.73), (18 d.p.i. vs 25 d.p.i. *p*-value = 0.98), (25 d.p.i. vs N.I. *p*-value = 0.08). (C) Percentage of IN foci associated with CPSF6 clusters at different times p.i. The data derive from two datasets of biological replicates. Source data are available online for this figure.

## HIV-1 nuclear condensates form during acute infection in vivo

Several laboratories have observed nuclear HIV-1-MLOs in cells infected in vitro (Bejarano et al, 2019; Francis et al, 2020; Rensen et al, 2021; Scoca et al, 2022; Selyutina et al, 2020) using high viral MOI. However, the importance of these structures under physiological infection conditions has not been demonstrated. We infected Balb/c Rag2−/−Il2rg−/−SirpaNOD (BRGS) humanized (HIS) mice with the HIV-1 NLAD8 molecular clone as previously

described (Li et al, 2018). Xenotransplantation of human CD34+ hematopoietic stem cell progenitors into BRGS hosts results in the development of various human lymphocyte populations (B, T, and NK cells) as well as myeloid cells (Legrand et al, 2011; Li et al, 2018; Li et al, 2016) (Appendix Fig. S2). BRGS HIS mice were efficiently infected with an intraperitoneal injection of $10^5$ tissue culture infective dose (TCID50) of a CCR5-tropic viral strain (NLAD8) (Li et al, 2018; Vicenzi et al, 1999) (Figs. 3A and EV1A). Blood samples were harvested from infected and uninfected BRGS HIS mice at 10 d.p.i. to assess viral replication during the acute phase of

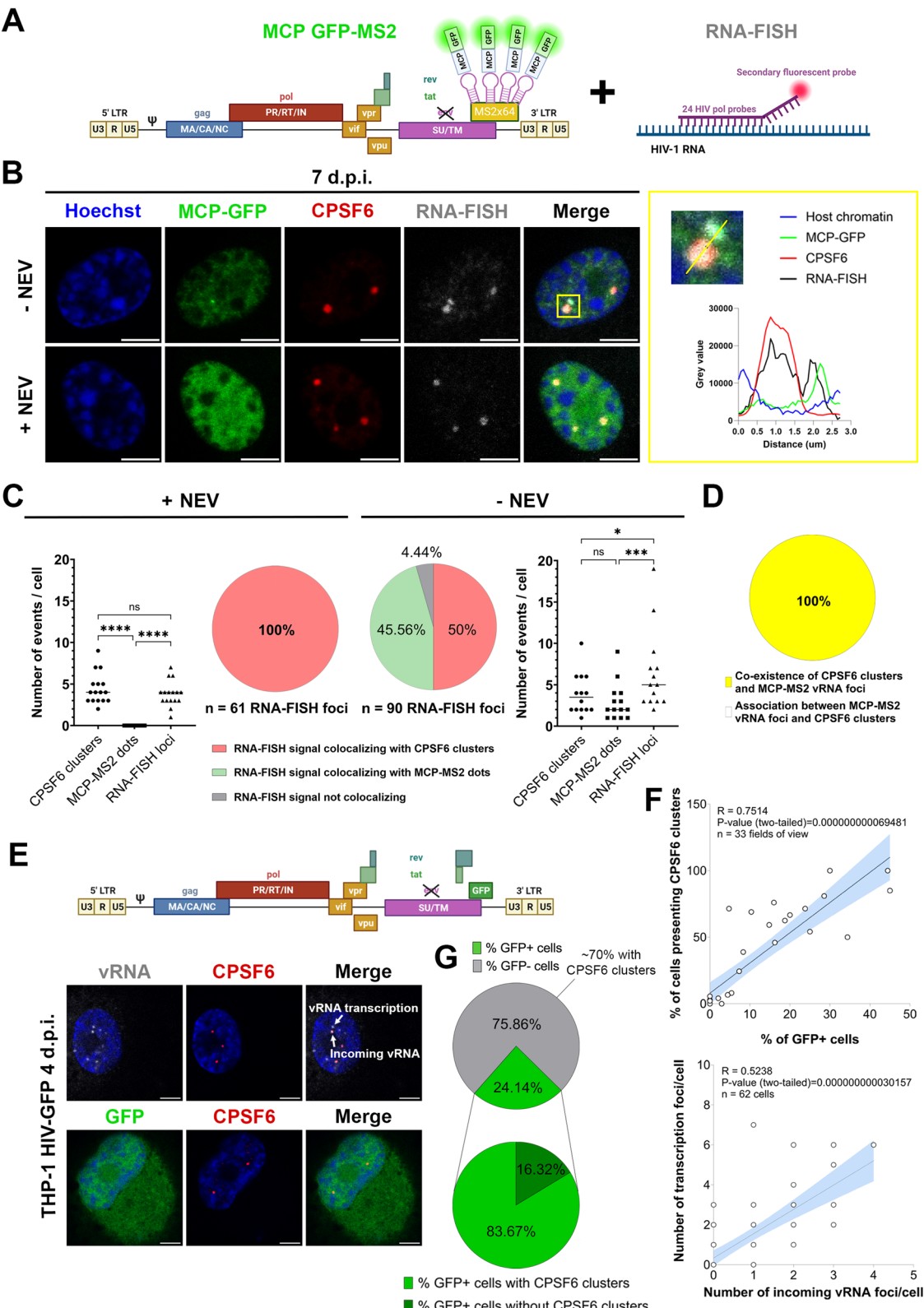

**Figure 2. CPSF6 condensates co-exist with active proviruses and shield viral RNA genomes.**

(A) Schematic representation of the two systems (MCP-MS2 and RNA-FISH) applied for the detection of the viral RNAs (done with BioRender). The cartoon on the left illustrates the backbone of the virus carrying MS2 loops at the place of nef used for the experiments. The cartoon on the right represents a schema of RNA-FISH approach. (B) Comparison between infection (MOI 30) in presence or in absence of NEV and co-labeling of the vRNA with MCP GFP-MS2 (in green) and RNA FISH (in gray). CPSF6 clusters are detected by the antibody against CPSF6 (in red) (scale bar = 5 μm). On the right, the zoom of the two spots analyzed and the graph displaying the intensity profile of vRNA detected with both approaches, both inside and outside of the CPSF6 clusters. (C) Number of events (CPSF6 clusters and vRNA detected by MCP-GFP or RNA-FISH) in presence (left) or in absence (right) of NEV. Data from two datasets of biological replicates are shown as the mean ± SD (Ordinary one-way ANOVA test; data +NEV: ns $p$-value = 0.415, ****: $p$-value=0.000000000031536, 0.000000002132483; data −NEV: ***: $p$-value = 0.0008; *: $p$-value = 0.0120; ns: $p$-value = 0.0522). Percentage of RNA-FISH detections colocalizing with CPSF6 clusters or MCP-GFP-MS2 dots or alone, in presence (left) or in absence of NEV (right). (D) In the absence of NEV, a percentage of cells displaying MCP-MS2 puncta also showed CPSF6 clusters, and no association between MCP-MS2 puncta and CPSF6 clusters was detected, across a sample size of 14 cells. (E) THP-1 cells differentiated with PMA were infected with HIV-1 ΔEnv pseudotyped with VSV-G and carrying the GFP reporter gene (schema made with BioRender). At 4 d.p.i., cells were fixed and labeled with an antibody to detect CPSF6 (in red) and with RNA-FISH probes (specific to the HIV-1 POL RNA sequence) to detect viral RNA (vRNA, in white) incoming or being transcribed by the cell. Productively infected cells expressed the GFP reporter protein (in green) and nuclei were stained with Hoechst (in blue) (scale bar = 5 μm). (F) Linear regression between the % of cells hosting CPSF6 clusters and the % of GFP-positive cells (GFP+), in 33 fields of view (R2 = 0.7514, $P$-value < 0.0001) (up). Linear regression between the number of vRNA transcription foci (outside CPSF6 clusters) and the number of incoming vRNA foci (co-localizing with CPSF6 clusters) per cell, in 62 cells (R2 = 0.5238, $P$-value <0.0001) (bottom). The error bars represent the 95% confidence interval of the simple linear regression curve calculated using Prism. (G) Proportion of GFP+ cells in the total cell population. The percentage of GFP negative cells carrying CPSF6 clusters (top) and proportion of cells presenting or not CPSF6 clusters in the GFP+ cells population (bottom) is indicated in pie charts. The data derive from two datasets of biological replicates. Source data are available online for this figure.

infection by RT-PCR (Figs. 3B and EV1B). BRGS HIS mice supported robust HIV-1 replication, with viremia reaching ~$10^7$ copies per milliliter at 10 d.p.i. (Figs. 3B and EV1B), as previously observed (Li et al, 2018). Infected and uninfected BRGS HIS mice were sacrificed, and monocytes were isolated from the bone marrow (BM), a lymphoid organ targeted by HIV-1, and immediately imaged to identify the presence of CPSF6. Notably, we exclusively observed CPSF6 clusters in infected mice, with no such clusters found in uninfected mice (Fig. EV1C).

Next, we differentiated monocytes derived from bone marrow into human monocyte-derived macrophages (MDMs) (Mlcochova et al, 2017; Rensen et al, 2021; Scoca et al, 2022). MDMs were subjected to immuno RNA-FISH to detect CPSF6 and vRNA. Interestingly, we were able to visualize incoming vRNA inside CPSF6 clusters (Fig. 3B,C; Appendix Fig. S3A), even though the majority of cells (>80%) did not display clusters and were probably non-infected, indeed they were vRNA negative (Appendix Fig. S3B,C). Interestingly, our ex vivo results align with those observed in cellulo (Fig. 2); indeed, cells isolated from infected HIS mice are categorized into two types: those containing only incoming vRNA and those exhibiting both incoming vRNA and vRNA from transcription foci. The latter cells thus carry viruses at different steps of the viral life cycle, from a step prior to reverse transcription to the post-integration step (Fig. 3C; Appendix Fig. S3A, Movies EV3 and EV4). The different locations of vRNA foci with respect to the nuclear location of CPSF6 clusters indicate that cells containing both forms of vRNA foci are actively undergoing productive viral replication (Figs. 2E and 3C). We further studied the property of these CPSF6 clusters in cells derived from infected HIS mice. Due to surface tension, biomolecular condensates are typically spherical, however, they can deform under physical force. Upon contact, they often merge before returning to their spherical shape (Hyman et al, 2014). Since in the majority of cases they are spherical, we evaluated the sphericity of these CPSF6 clusters in 3D by ICY software (Scoca et al, 2022). All clusters showed >95% sphericity, and a more stringent analysis showed that 92% of CPSF6 clusters had >98% sphericity (Fig. 3D). These results show that biomolecular condensates efficiently form under physiological conditions (Figs. 3A–D and EV1).

## Exploring the viral morphologies present in HIV-1 nuclear condensates

Viral cores from single round HIV-1 pseudotyped with glycoprotein G from vesicular stomatitis virus (VSV-G) have been detected in the host nucleus (Li et al, 2021; Muller et al, 2021; Zila et al, 2021), however, their morphological composition and distribution in nuclear condensates is unknown. Here, we used immunogold labeling with transmission electron microscopy (TEM) on thin sectioned macrophage-like cells that were infected with a single round HIV-1 VSV-G pseudotyped virus preparation for several days (3 or 7 days) in the absence or in the presence of NEV. The presence of NEV fully blocks reverse transcription, so all incoming vRNAs should remain trapped inside HIV-1-MLOs, according to our previous results (Rensen et al, 2021; Scoca et al, 2022). However, it is unclear whether this viral genome remains associated or not with viral components and why it does not diffuse outside of these condensates. We co-labeled the viral CA (gold beads with 6 nm diameter) and CPSF6 (10 nm gold beads) on the 80 nm ultrathin sections to target the nuclear HIV-1-MLOs (Fig. 4A). Of note, CPSF6 condensates were only visible in infected cells and not in uninfected cells (Fig. 4A; Appendix Fig. S4A,B), as also shown by fluorescence microscopy (Fig. 1A–C). Surprisingly, we were able to observe several conical shapes ~100 nm in length located near CPSF6 and CA gold beads in both samples treated and untreated with the NEV (Fig. 4B,C). Indicating that viral cores enter the nucleus, where they accumulate and persist inside the nuclear microenvironment several days post-infection (Fig. 4B,C) and remain functional for several days as shown when the NEV is removed (Fig. EV2A,B) but also by the increase of GFP+ cells during the time post-infection (Fig. EV2C; Appendix Fig. S4C). Next, ultrathin sections were co-labeled with antibodies to reveal CPSF6 (10 nm gold particles) and Integrase carrying the HA tag (IN$_{HA}$) (6 nm gold particles) (Blanco-Rodriguez et al, 2020; Petit et al, 2000) at seven d.p.i. (Fig. 4D; Appendix Fig. S4D). When we quantified the number of IN proteins revealed by immunogold labeling, we detected almost twice as many in samples without drug than in NEV-treated cells (Fig. 4D), indicating that IN proteins are less shielded by the capsid in the absence of the drug

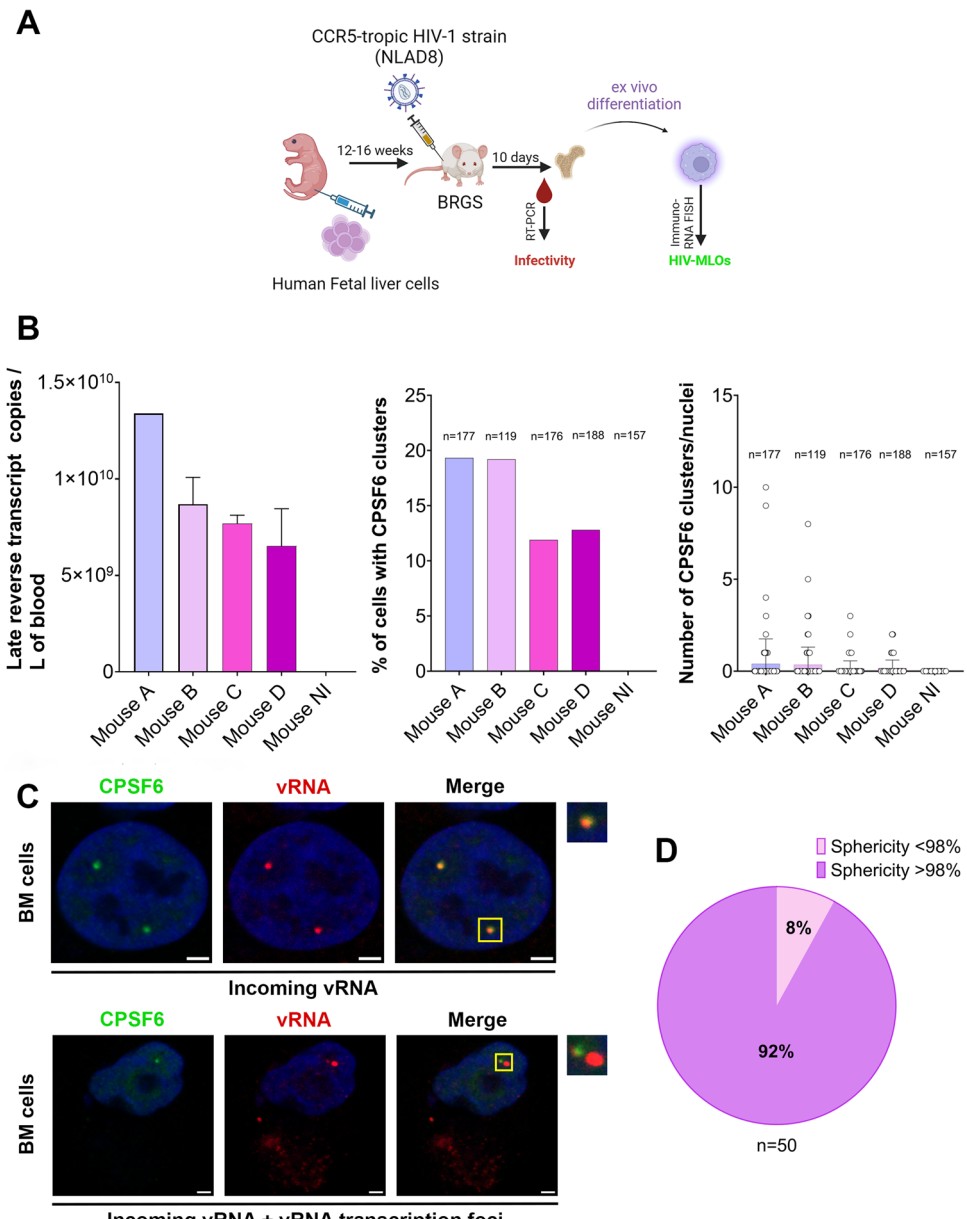

**Figure 3. HIV-1-MLOs build during in vivo infection.**

(A) Schema of CCR5-tropic HIV-1 strain (NLAD8) infection in BRGS mice (done with BioRender). (B) Graphs showing viral infectivity in BRGS mice by quantitative RT PCR, measuring the late reverse transcripts copies per mL of blood in each mouse (left) (Data are shown as the mean ± SD of two replicates), quantification of the percentage of cells presenting CPSF6 clusters per mouse (middle) and the number of CPSF6 clusters per nuclei in individual mice (right) (the n corresponds to the number of cells analyzed). (C) Mononucleated cells derived from BM of infected BRGS mice were differentiated ex vivo and labeled for the detection of the vRNA (in red) and CPSF6 (in green). Nuclei were stained with Hoechst (in blue) (scale bar = 2 μm). (D) Sphericity analysis conducted on CPSF6 clusters by ICY software. Source data are available online for this figure.

and protrude outside the viral core. These results show that reverse transcription is followed by the formation and the exposure of pre-integration complexes (PICs) carrying multiple IN proteins (Ballandras-Colas et al, 2017; Passos et al, 2017). The progression of viral nuclear steps in HIV-1-MLOs correlated with the observation of smaller viral condensates, which had diameters significantly smaller than those seen in the samples treated with the drug. These drug-treated samples exhibited frozen dynamics of

the RT and featured larger nuclear condensates, with diameters ranging from approximately 0.5 to 1.2 μm, with a mean of 0.843 in drug-treated samples compared to a mean of 0.711 μm in untreated samples (T-test, **$p$-value = 0.0045) (Fig. 4E). It is likely that their large size is a result of the fusion typical of biomolecular condensates, as we previously demonstrated (Scoca et al, 2022). Furthermore, in NEV-treated samples, we observed a significantly higher density of gold labeling for CPSF6 compared to

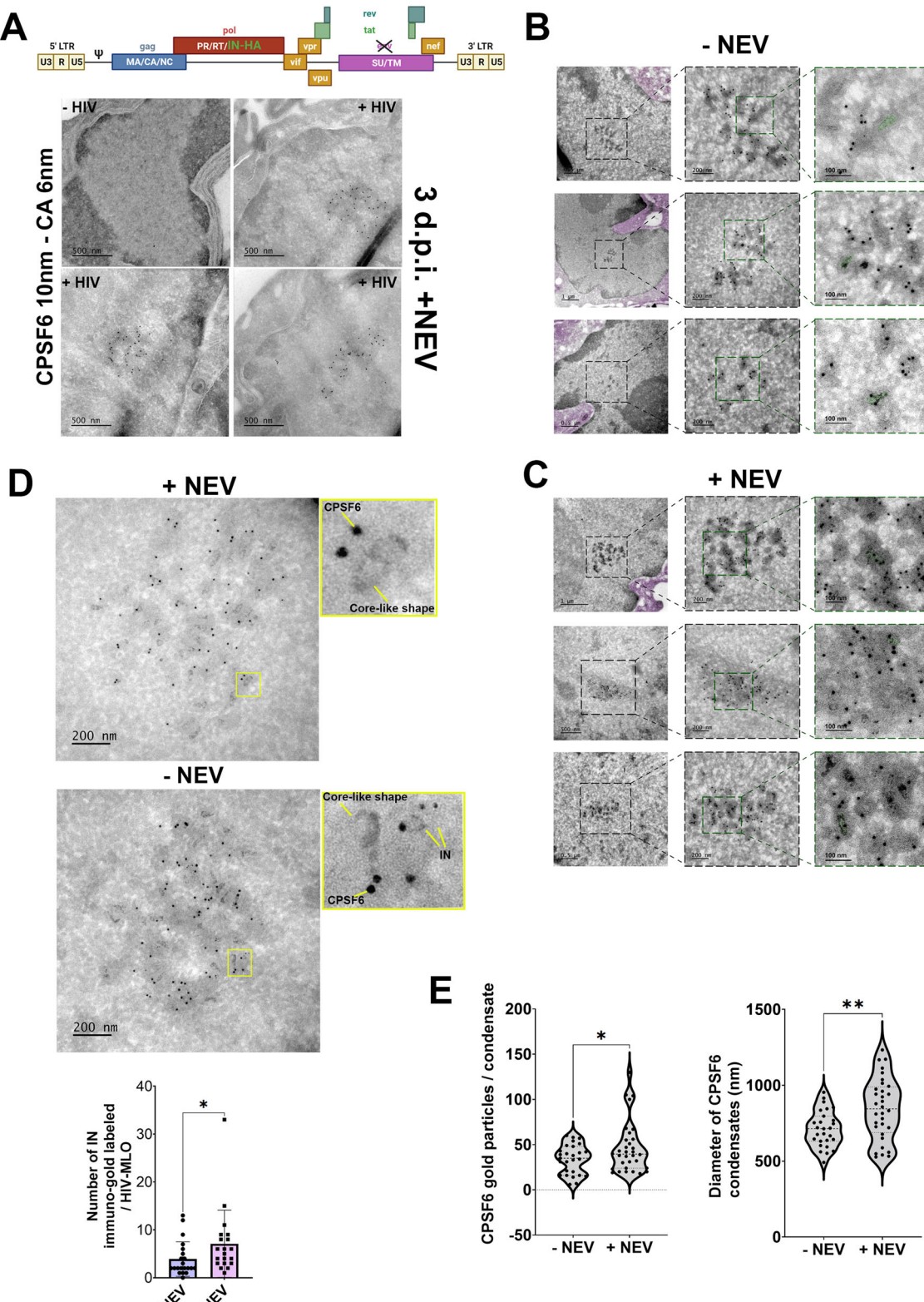

untreated cells (T-test, *p*-value = 0.0435). The mean number of CPSF6 gold particles per condensate increased from 33.89 in untreated cells to 46.63 in drug-treated cells (Fig. 4E). Our observations suggest changes in viral nuclear condensates throughout the HIV-1 replication cycle (Fig. 4A–E).

## Uncovering the structural diversity of viral morphologies in HIV-1 nuclear condensates

While recent in vitro studies have shown that intact viral cores can serve as a protective cage for RT, and that viral cores uncoat when

**Figure 4.  Viral components in nuclear condensates.**

(A) At the top is a schematic representation of the viral genome used for electron microscopy (EM) experiments (done with BioRender). THP-1 cells infected or not for 3 days in presence of NEV. Sections were co-labeled with antibodies against CPSF6 and CA. (B) Three areas showing HIV-1-MLOs in the nucleus of infected cells in absence of NEV at 7 d.p.i. In the left column (overview panels) the cytoplasm is pseudocolored in purple to be easily discriminated from the nucleus (remaining in gray). In the middle column higher magnification projection images of the corresponding HIV-1-MLO sites are displayed. In the third column, close up views from the middle column images are shown. Some of the HIV-1 cores are easily recognized even in 2D projection images, one such core per image is pseudocolored in green (right column panels) for easier visualization. Diameter of CPSF6 immunogold beads = 10 nm, diameter of CA immunogold beads = 6 nm. Scale bar is shown in each panel. (C) Similar to (B) but cells were infected in presence of NEV for 7 days. (D) Cells were infected for 7 days and labeled with antibodies against CPSF6 and IN, followed by labeling with secondary antibodies conjugated to gold particles of 10 nm and 6 nm in size, respectively. The histogram shows the quantification of the number of integrase (IN) proteins detected by gold particles in the presence or absence of NEV. The statistical analysis was performed using the Mann–Whitney test (*$p$-value = 0.0212). Data are shown as the mean ± SD of two independent datasets. (E) Comparison between HIV-1-MLOs size and number of CPSF6 dots detection is shown in violins graphs. Two independent datasets from biological replicates have been included in the analysis. Data are shown as the mean ± SD (T test *$p$ value = 0.0435; **$p$ value = 0.0045). Source data are available online for this figure.

the inner pressure due to the completion of RT is increased (Christensen et al, 2020; Yu et al, 2022), the cellular relevance of this aspect and the cellular compartment where this phenomenon of the viral life cycle occurs is still unclear. To explore this issue, we conducted dual-axis electron tomography and volume annotation of viral core-like structures located in HIV-1-MLOs identified by double CPSF6 and CA gold labeling, enabling the visualization and quantitative morphological analysis of changes in molecular crowding during viral infection or when the RT process is impeded by NEV (Fig. 5A). Cells were infected for 7 days in the presence or in the absence of the NEV, and we obtained two hundred eighty-five core-like structures found inside nuclear HIV-1-MLOs. Upon tomogram reconstruction, we observed three different capsid categories classified according to their inner electron density signal, which populate HIV-1-MLOs (Fig. 5A; Appendix Fig. S4E, Fig. EV3A,B, Movies EV5, EV6, EV7 and EV8). Cores were classified into three categories per site/tomogram: i. Ghosts were not identifiable in projection images but observable in dual-axis tomographic volumes, with interiors matching their exterior peripheral space pattern; ii. Dense cores were easily identified in projection images and in dual-axis tomograms, exhibiting dark areas usually throughout the core; iii. Lighter cores were less easily identified in projection images, showing dark spots in some parts after reconstruction. Indeed, high magnification observations revealed dense cores (annotated in magenta), lighter cores (in blue), and what appeared as empty cores (that we refer to as 'ghosts'; in yellow) (Fig. 5A, Movies EV5 and EV6). We differentiated and quantified the shapes, classifying them as either conical or tubular, based on the shapes observed in the viral preparation (Fig. EV3A,B). Our observations showed that the majority are conical: 80% cones versus 20% tubes in the absence of NEV, and 70% cones versus 30% tubes in presence of NEV (Fig. EV3A,B). Ghosts maintained a similar shape to lighter and dense cores but their interior had instead the same low-density contrast and pattern as the surrounding background (Fig. 5A; Movies EV5 and EV6), giving the impression of them being empty. Ghosts retain their conical shape, likely, due to the preservation of the core itself. Indeed, we observed a faint electron-dense contour enveloping these ghost capsids, which might be stabilizing them. CPSF6 surrounds these structures, as it was clearly shown by gold labeling (Fig. 5A). Thus, we quantified and classified the most visible core shapes in our tomograms using a combination of pixel classification with manual tracing (Fig. EV3A,B; Movies EV11 and EV12). In the absence of NEV, one hundred twenty-one core-like

shapes were annotated, and we found that most of them were ghosts (64%), followed by light cores (23%) and a few remaining dense cores (13%). In the presence of NEV, we found and annotated one hundred sixty-three core-like shapes and found 49% ghosts. There were 18.4% of dense cores and 32.5% of light cores (Fig. 5B). In cells without drug, where the virus undergoes internal core pressure triggered by the RT, dense and light cores were less represented inside HIV-1-MLOs than in treated cells (36% vs. 50.9%) (Fig. 5B). Of note, we found ghosts in both conditions, implying that this capsid category forms during the viral cytoplasmic and nuclear journey and it is unlikely that they are already present in the viral preparation (Mattei et al, 2016) as we show by the comparison of viral cores in the viral preparation and inside HIV-1-MLOs (Fig. EV4A). We found that the mean intensity within the cores of the viral preparation more closely resembled that of dense cores rather than empty cores (Fig. EV4B). A recent study, using coarse-grained molecular dynamics simulations, suggests that some viral cores might lose their content during nuclear translocation (Kreysing et al, 2024) supporting our hypothesis. On the other hand, it is also plausible that the presence of ghosts in NEV-treated cells could result from either limited drug penetration in HIV-1-MLOs or by capsid damage caused by the movement of highly infected cells (Bhargava et al, 2021). Nevertheless, we noted a 15% increase in the number of ghosts in untreated cells compared to those treated with NEV, suggesting that they may, at least in part, result from the release of viral DNA. Intriguingly, in a tomogram of NEV untreated infected cells, we observed an electron dense signal immediately adjacent the ghost's head. This direct observation might suggest that viral content is indeed released from the HIV cores that appear as ghosts in our preparation (Fig. EV4C; Movies EV9 and EV10). Although this was a rare event in our dataset, it closely resembled previous observations of in vitro cores undergoing endogenous reverse transcription (Christensen et al, 2020). The detection of ghosts also suggests that the viral core does not fully disassemble and that uncoating occurs as a gradual process that takes place within nuclear HIV-1-MLOs. It's worth noting that in macrophages, viral reverse transcription is slower compared to T cells, and the infection process is hindered by the presence of numerous restriction factors, limiting the ability of only a few viruses to establish infection. Our data indicate that core structures exhibit higher integrity (as indicated by denser and lighter cores) when RT activity is inhibited, implying that they can remain preserved for an extended period before reactivation occurs upon NEV removal

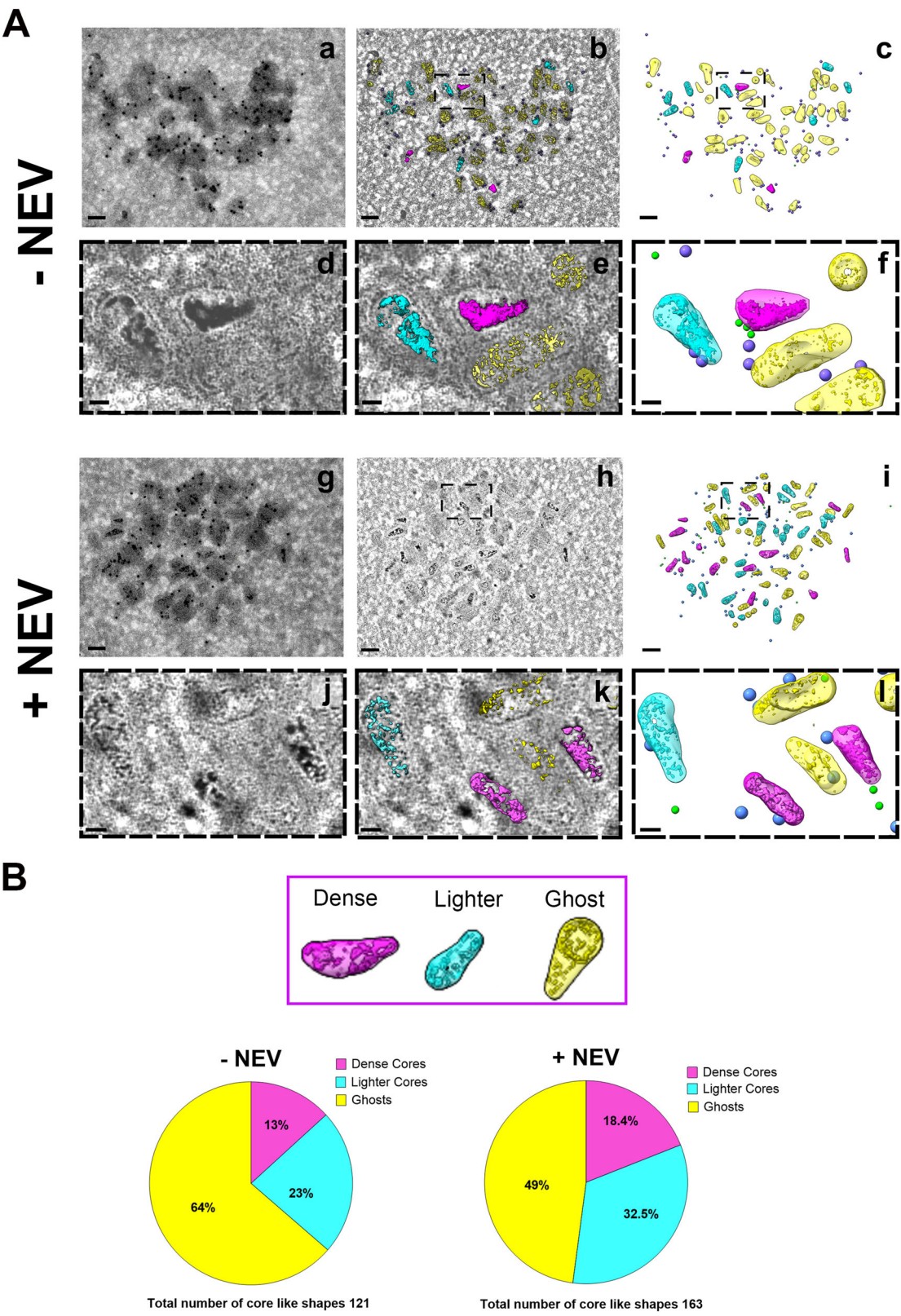

**Figure 5. Viral nuclear condensates host different viral core morphologies.**

(A) Different morphologies of HIV-1 core-like shapes inside CPSF6 clusters immuno-gold labeled are shown in the host cell nucleus. Ultrathin sections immunolabelled for CPSF6 (10 nm gold beads, annotated as purple spheres) and the capsid protein (6 nm gold beads, annotated in green) in presence and absence of NEV were screened using TEM. Large clusters of CPSF6 immunogold were observed in the cell nucleus. In (a), 2D projection image of a cluster observed in absence of NEV. In (b), one of the central slices of the dual-axis tomographic volume in the same area, revealing the presence of at least three different kinds of cores. In (c), full annotation of the different cores (very dense cores in magenta, less dense in cyan and ghosts in yellow). In (d) and (e), the zoomed area of the panel in (b) with and without the iso-surface rendering of the core interiors. In (f), the equivalent zoomed area of (d) and (e) only with the different kind of cores and beads annotated. In (g), 2D projection image of a cluster observed in presence of nevirapine. In (h), one of the central slices of a dual-axis tomographic volume in the same area. In (i), the full annotation of the different cores. In (j) and (k), the zoomed area of the panel in (h) with and without the iso-surface rendering of the core interiors. In (l), the equivalent zoomed area of (j) and (k) only with the different kind of cores and beads annotated. Color coding follows the one of panels (a–f). Scale bar for panels (a–c) and (g–l) is 100 nm. Scale bar for panels (d–f) and (j–l) is 20 nm. (B) Percentage of dense cores, lighter cores and ghosts in cells treated (right) and untreated (left) with NEV from 284 total cores. Source data are available online for this figure.

(Fig. EV2; Appendix Fig. S4C). At the same time, the persistence of ghosts could contribute to generate a favorable microenvironment for viral DNA synthesis, by maintaining viral nuclear condensates (Figs. 5A,B and EV3; Appendix Fig. S4E). Alternatively, "ghost" virus cores could be remnants without further function or serve as a reservoir for some viral components without a protective function.

## The HIV-1 nuclear condensate: the main site for nuclear reverse transcription

We then investigated whether HIV-1 nuclear condensates create a major microenvironment for viral nuclear reverse transcription. We exposed macrophage-like cells to HIV-1, which carried the GFP reporter gene, for 120 h, in presence or in absence of NEV (Fig. 6A). In both conditions, we observed the formation of CPSF6 condensates. Notably, the number of condensates was approximately twice as high in the presence of NEV compared to untreated cells (Fig. 6B). This increase is attributed to the virus being on hold and sequestered within the nuclear condensates, which leads to the persistence of CPSF6 condensates for a longer duration (Fig. 1). Simultaneously, we leveraged the reversibility of NEV to investigate whether HIV-1 nuclear condensates are the main sites for the initiation of reverse transcription within the nucleus. We infected the cells with NEV for 48 h, after which it was removed. This allowed the incoming vRNA to accumulate in the nucleus. We treated some of the samples with the compound PF-3450074 (PF74), known for its ability to dissolve CPSF6 condensates only at a high dose (Figs. 6A and EV5; Appendix Fig. S5) (Selyutina et al, 2022). Although high doses of PF74 lead to the disassembly of CPSF6 clusters, they conversely were shown to stabilize viral cores in vitro (Bhattacharya et al, 2014; Dostalkova et al, 2020; Fricke et al, 2017), however, the effect of this compound in capsid located inside CPSF6 clusters has never been evaluated. Since we did not observe viral core structures outside HIV-1-MLOs (Figs. 4, 5 and EV3; Appendix Fig. S4B) we expect that the effect of PF74 high dose should be mainly related only to CPSF6 clusters and their content.

Subsequently, we assessed infectivity in correlation with the presence of CPSF6 condensates at 120 h post-infection. When the NEV was removed, we observed that 7.48% of cells were GFP positive, representing a 32% restoration in infectivity compared to cells infected without the drug, which had an infectivity rate of 23.2%. In contrast, cells treated with high dose of PF74 after the removal of NEV were unable to restore effective infection (Fig. 6A). The loss of infectivity can be attributed to a block in the reverse transcription process, as supported by the qPCR data (Fig. 6B). Interestingly, NEV did not completely block all catalytic sites of the

reverse transcriptase enzymes hosted within the CPSF6 condensates, as evidenced by the presence of a residual reverse transcription activity giving rise to a low amount of ERT (early reverse transcripts), detectable by qPCR that did not form LRT (late reverse transcripts) at least within the first 120 h of infection (Fig. 6B). This indicates that CPSF6 clusters create a favorable environment for reverse transcription, impeding optimal drug penetration. The detection of ERT in NEV-treated samples aligns with the observation of 'ghosts' in NEV-treated samples in electron microscopy experiments (Fig. 5A,B). Overall, the addition of PF74 disassembles the CPSF6 condensates, disrupting the microenvironment required for the nuclear reverse transcription and impeding the formation of ERT and LRT (Fig. 6A,B). We observed that after 2 h of PF74 treatment, there were numerous capsid puncta (in cells fixed with PFA, the antibody against capsid can only detect it if the epitope is not occupied by CPSF6 (Muller et al, 2021) (Fig. 6C). This suggests that the capsids were not fully disassembled by the high dose of PF74, consistent with previous in vitro data (Bhattacharya et al, 2014; Dostalkova et al, 2020; Fricke et al, 2017). However, we cannot rule out the possibility of minor structural changes in the capsids within CPSF6 clusters. Intriguingly, the number of CPSF6 clusters in untreated infected THP-1 cells closely matched the number of CA puncta revealed in infected THP-1 cells after 2 h of high-dose PF74 treatment (Fig. 6C).

These results underscore the significance of HIV-1 nuclear condensates, primarily composed of CPSF6 proteins and viral cores, as main and favorable environment for nuclear reverse transcription.

## The HIV-1 nuclear condensate: a shield against innate immunity

The final product of reverse transcription is dsDNA, formed in the host nucleus (Burdick et al, 2020; Dharan et al, 2020; Rensen et al, 2021; Scoca et al, 2022; Selyutina et al, 2020). We investigated whether it could be recognized by cyclic GMP-AMP synthase (cGAS) (Gao et al, 2013), leading to the production of the secondary messenger cyclic GMP-AMP (cGAMP), thereby activating innate immune responses. Interestingly, even in the presence of Vpx, which degrades the restriction factor SAMHD1 (Laguette et al, 2011) the HIV-1 capsid evades sensing of dsDNA before integration (Lahaye et al, 2018; Zuliani-Alvarez et al, 2022).

Hence, we posed the question of whether the innate immune response could be impeded by the presence of nuclear condensates that shield the newly synthesized viral dsDNA. To explore this possibility,

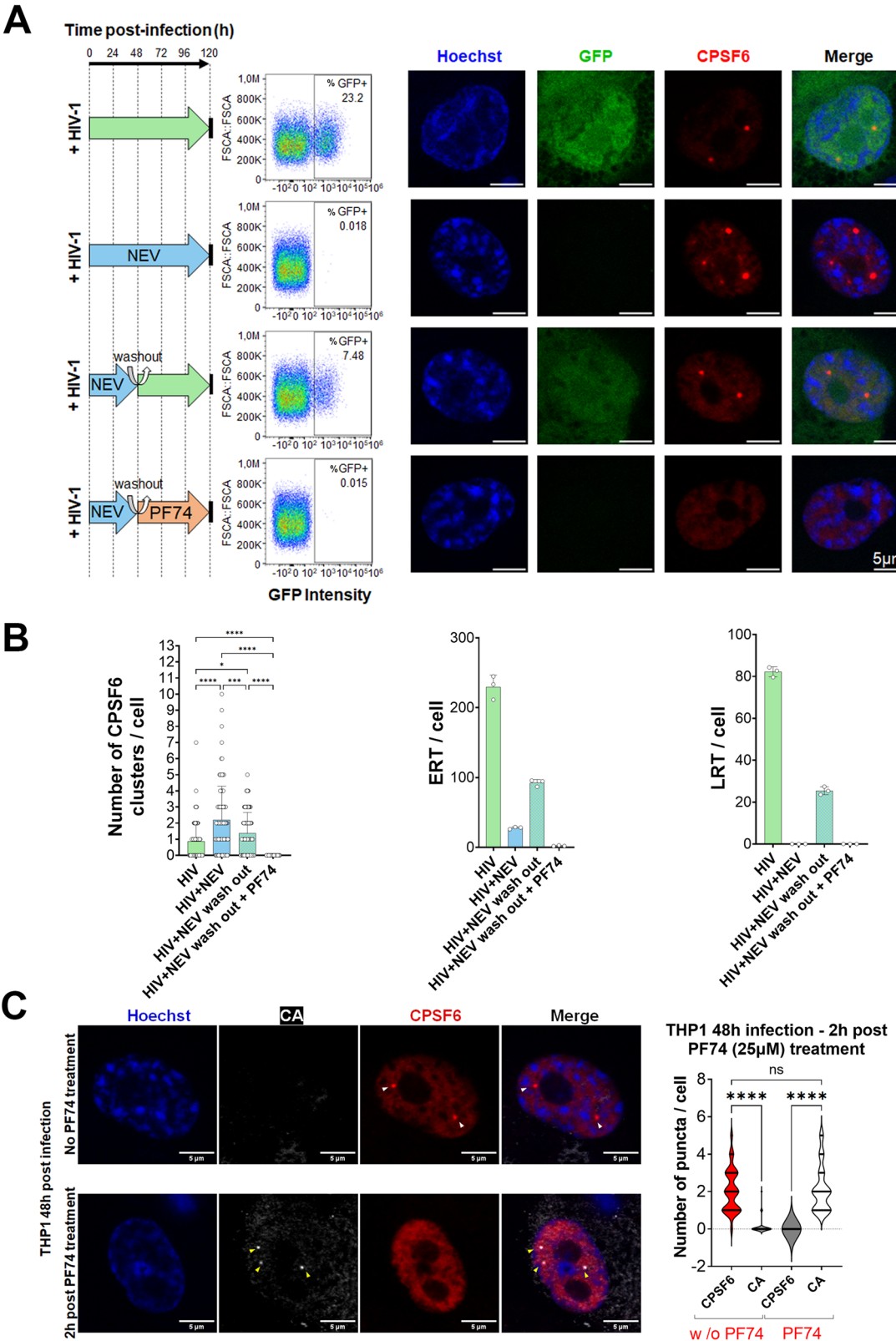

**Figure 6. CPSF6 condensate disassembly impedes nuclear viral reverse transcription after nevirapine removal.**

(A) THP1 cells differentiated with PMA were infected (with HIV-1 ΔEnv pseudotyped with VSV-G and carrying the GFP reporter gene, schema Fig. 2E) (MOI 10) for 120 h without any treatments, or treated with NEV for 120 h, or treated with NEV and washed out after 48 h adding or not PF74 (25 μM) and then left without any treatments for 72 h. Cells were fixed 120 h p.i. (first panel from the left). Productive infection was quantified by FACS through GFP expression intensity (2nd panel starting from the left). Fixed cells were labeled with an antibody to detect CPSF6 (in red). Productively infected cells expressed the GFP reporter protein (in green) and nuclei were stained with Hoechst (in blue) (scale bar = 5 μm) (3rd panel from the left). (B) Histograms showing the quantification of CPSF6 clusters per cell (almost 100 cells per condition have been analyzed) from two datasets of biological replicates (top panel). Data are shown as the mean ± SD (Ordinary one-way ANOVA test; ****: *p*-value = 0.000000000273883, 0.000059345976010, <0.00000000000001, 0.000000000021184; ***: *p*-value = 0.0002; *: *p*-value = 0.0473). Quantitative PCR: early viral reverse transcripts (ERT) per cell (middle panel) or late viral reverse transcripts (LRT) per cell (lower panel) in the different culture conditions. A representative experiment is shown, based on two independent experiments, each performed with three replicates. (C) THP-1 cells were treated with a high dose of PF74 (25 μM) at 48 h post-infection (h.p.i.) (MOI 10), while a control group remained untreated. The number of puncta from CPSF6 (indicated by white arrows) and CA (indicated by yellow arrows) were quantified in both cell populations, with the results presented in the violin graph (number of cells analyzed = 222). The data are derived from three independent experiments and shown as the mean ± SD. An ordinary one-way ANOVA test was used for statistical analysis; **** indicates a *p*-value = 0.000000000000076, 0.000000000000081 and 'ns' denotes not significant *p*-value = 0.947850878211297. Scale bar: 5 μm. Source data are available online for this figure.

we investigated whether cGAS could be activated upon exposure to viral DNA by destroying HIV-1-MLOs. While it was commonly believed that cGAS primarily operates in the cytoplasm, recent studies have shown that cGAS also localizes to the nucleus (Zhao et al, 2020), participating in nuclear processes such as DNA repair (Liu et al, 2018) and innate immunity against HIV-2 (Lahaye et al, 2018).

The viral DNA can be detected in HIV-1 nuclear condensates using EdU and click chemistry (Rensen et al, 2021) (Fig. 7A). We thus infected macrophage-like cells for 48 h, when the vDNA is observed in HIV-1-MLOs (Rensen et al, 2021; Scoca et al, 2022) (Fig. 7A) and subsequently a high dose of PF74 was added to destroy the HIV-1 nuclear condensates (Fig. EV5). Ten hours after PF74 treatment, we quantified the product of cGAS activation, cGAMP. Our findings showed about a ~4-fold increase in average of cGAMP levels in samples treated with PF74 compared to uninfected cells. In contrast, cells infected without PF74 maintained their HIV-1 nuclear condensates, which served as a protective shield for the viral DNA against the innate immune response. Indeed, no increase in cGAMP levels was observed between cells treated with NEV and those left untreated (Fig. 7B). We further demonstrate that PF74 does not increase cGAMP levels by itself, nor through episomal forms that predominate when viral integration is inhibited by an anti-integrase drug like raltegravir (RAL) (Appendix Fig. S6). It has been observed that once the viral dsDNA is formed, it is released from the viral capsid (Christensen et al, 2020). Therefore, the activation of cGAS likely results from the exclusive exposure of the viral DNA once the CPSF6 clusters are disassembled by the high dose of PF74. Indeed, newly synthesized viral DNA is primarily visualized within CPSF6 clusters (Fig. 7A). Interestingly, an increase in cGAMP is detectable as early as 2 h after PF74 treatment (Fig. 7C), when viral capsid puncta are still present (Fig. 6C). This increase is also observed at 10 h post-treatment (Fig. 7C). Furthermore, cells knocked out for CPSF6 exhibit an increase in cGAMP compared to control cells following infection (Fig. 7D). Notably, the activation of cGAS cannot be attributed to damage to the viral cores by PF74, as this compound was not used during the infection. These observations support the protective role of CPSF6 clusters in the nucleus against innate immunity. Indeed, in the absence of CPSF6 clusters, cGAS activity targets viral dsDNA that once formed is released from the core shell (Fig. EV4C; Movies EV9 and EV10) (Christensen et al, 2020) within the host's nucleus (Burdick et al, 2024; Muller et al, 2021; Scoca et al, 2022). cGAMP is a second messenger that activates STING (stimulator of interferon

genes), which initiates signaling cascades that lead to the production of type I interferons and other inflammatory cytokines. This activation leads to the transcription of interferon-stimulated genes (ISGs), including CXCL10, MxA, and IFIT2. We, thus, measured the expression of ISGs, which are typically overexpressed as a consequence of cGAMP production. We observed significant overexpression of CXCL-10 (~170-fold), IFIT-2 (~16-fold), and MxA (~20-fold) in infected samples treated with a high dose of PF74, compared to untreated infected THP-1 cells (ANOVA test, *p*-value * < 0.05) (Fig. 8A). Importantly, we also demonstrated that ISGs are overexpressed in MDMs isolated from healthy patients infected and treated with PF74 compared to infected and untreated MDMs (ANOVA test, MxA *p*-value *** 0.0007; CXCL10 *p*-value *** 0.0007; IFIT2 *p*-value ** 0.0043) (Fig. 8B). This underscores the physiological importance of CPSF6 condensates in protecting against innate immune responses. These results shed light on an unexplored role of HIV-1-MLOs, acting as a protective shield for newly synthesized viral DNA or hybrid forms vRNA and vDNA. If exposed, these viral forms can trigger cGAS activation and interfere with the viral spread.

## Discussion

HIV-1, a contagious and pathogenic virus, circumvents innate immunity (Cingoz and Goff, 2019). In this study, we provide evidence for a new mechanism used by HIV-1 to shield its genome from immune sensors, which relies on the establishment of viral nuclear membraneless organelles. Our study uses advanced technologies to investigate the molecular, structural and functional aspects of viral replication within these nuclear biomolecular condensates. This research involves in vitro and animal model experiments. We proposed a mechanistic model where condensates create a favorable environment induced by the viral infection, in which multiple viral core categories persist, likely representing different reverse transcription stages, aiding the progression of infection.

HIV-1 leverages the nuclear host factor CPSF6 to conceal the cores inside the host nucleus, presumably safeguarding viral nucleic acids via the formation of HIV-1-MLOs. We show that nearly all efficiently infected cells exhibit HIV-1-MLOs in their nucleus, supporting their importance for efficient progression of the post-nuclear entry steps. Our findings are reinforced by the observation of viral nuclear condensates in cells from humanized mice's bone

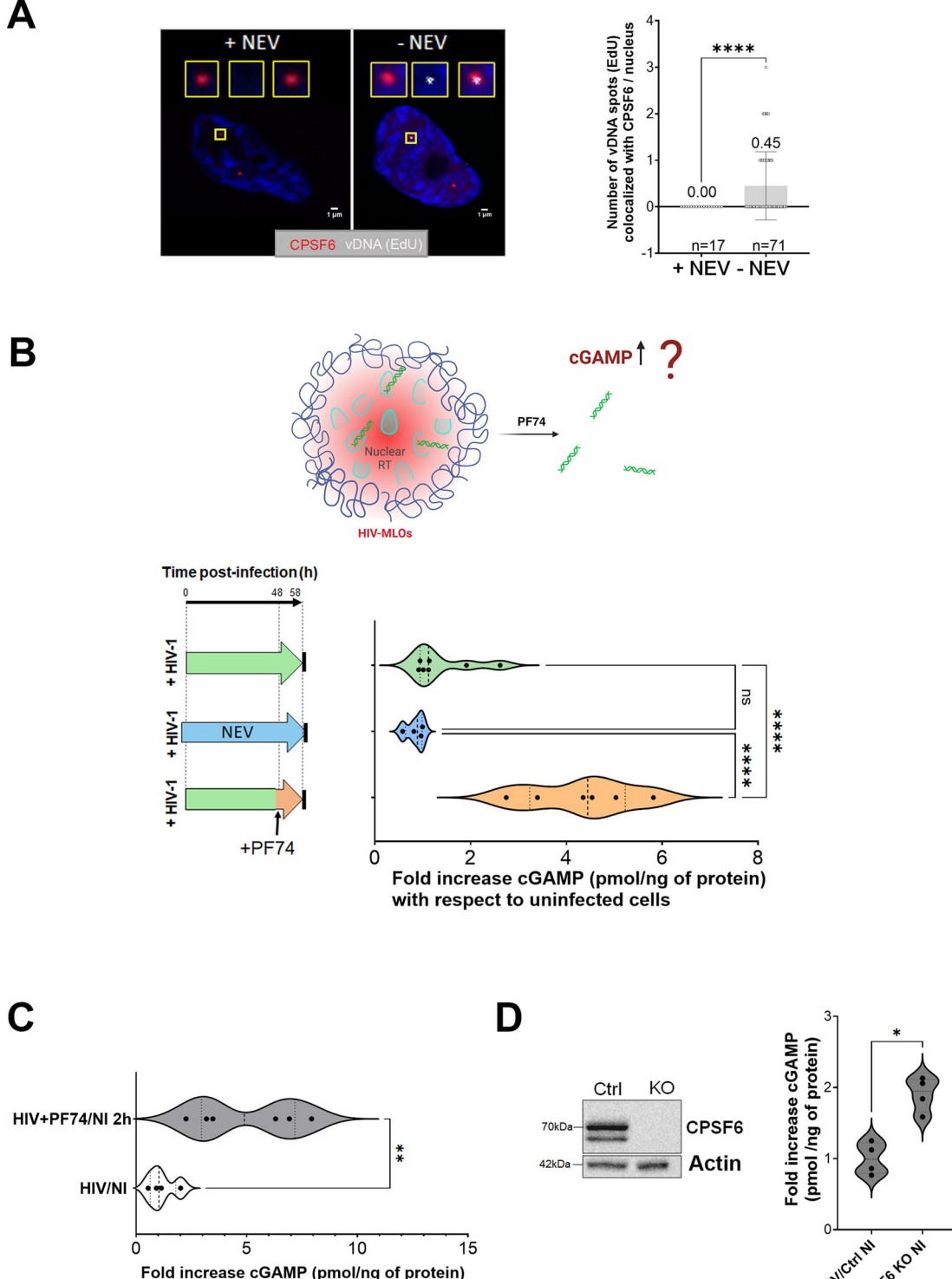

**Figure 7. CPSF6 condensates shield viral genomes from cGAS in the nucleus of infected cells.**

(A) THP-1 cells differentiated with PMA were infected with HIV-1 ΔEnv pseudotyped with VSV-G, in presence or absence of NEV, with EdU supplemented to the media. Fixed cells were labeled with an antibody to detect CPSF6 (in red) and viral DNA (vDNA) was revealed through EdU Click-chemistry (in gray). Nuclei were stained with Hoechst (in blue) (scale bar = 1 μm). The right panel shows the quantification of vDNA spots colocalizing with CPSF6 clusters per nucleus, in presence or absence of NEV (error bars are represented as ± SD, unpaired Welch's t-test; ****: $p$-value = 0.000002014358640). (B) Model illustrating the disassembly of HIV-1-MLOs potentially exposing vDNA (on the top) (made with BioRender). THP-1 cells differentiated with PMA were infected with HIV-1 ΔEnv pseudotyped with VSV-G, in presence or absence of NEV. 48 h p.i. cells were treated with PF74 or not (bottom left panel presents the various culture conditions). Ten hours post PF74 treatment, cells were lysed and cGAMP levels were measured by cGAMP ELISA assay (bottom right panel). The fold increase of cGAMP in each sample with respect to uninfected cells is represented. The data are derived from three independent experiments and shown as the mean ± SD. (Ordinary one-way ANOVA test; ns: $p$-value = 0.428237768384628; ****: $p$-value = 0.000006748324383, 0.000001340723954). (C) cGAMP measurements were taken at 2 h post PF74 treatment. Following treatment, cells were lysed, and cGAMP levels were quantified using a cGAMP ELISA assay. The fold increase in cGAMP levels, compared to uninfected cells, is depicted in a violin graph for THP-1 cells that were either untreated or treated with a high dose of PF74. Statistical analysis was conducted using unpaired Welch's t-test; **indicates a $p$-value = 0.0084. Data derive from two independent experiments and shown as the mean ± SD. (D) Role of CPSF6 in cGAS activation evaluated using THP-1 KO for CPSF6. Western blot showing control cells (Ctrl) and CPSF6 knock out (KO) THP-1 cells obtained by CRISPR Cas9 technology. Cells were infected (maintaining a similar time used for the samples treated with PF74 in (B)) and lysed. cGAMP levels were quantified using a cGAMP ELISA assay. Statistical analysis was conducted using t-test; * denotes a $p$-value = 0.0141. Data derive from two independent experiments and shown as the mean ± SD. Source data are available online for this figure.

marrow, underlining the significance of this post-nuclear entry step for in vivo viral replication.

Further characterization of these viral condensates has unveiled their role as main nuclear site for viral DNA synthesis. This insight is supported by immune-gold electron microscopy studies, which reveal the abundant presence of IN proteins outside viral cores but inside HIV-1-MLOs in infected cells, in contrast to cells where reverse transcription was inhibited by NEV treatment. This suggests that when the reverse transcription process is active, viral material, likely in the form of pre-integration complexes, is released from the viral cores. Conversely, the inhibition of reverse transcription is likely to influence the dynamics of CPSF6 condensation within HIV-1-MLOs leading to an increased concentration of CPSF6 proteins and the formation of larger organelles (up to 1.2 μm) through a fusion process typical of liquid–liquid phase separation (Scoca et al, 2022). Importantly, electron tomography allowed us to classify different core-like shapes within HIV-1-MLOs, including dense, lighter, and ghost or empty structures. These structures were surrounded by a low electro-dense signal, possibly representing an accumulation of proteins, particularly CPSF6, that may contribute to their stabilization. Interestingly, in NEV-treated cells, the cores appeared more intact (dense + lighter cores) compared to untreated cells (52.1% vs. 36%), although all three core categories were observed in both conditions. Next, to unravel the role of HIV-1-MLOs in the post-nuclear entry steps, we observed that when viral nuclear condensates are disassembled using a high dose of PF74, which interferes with the interaction between CPSF6 and the viral capsid (Buffone et al, 2018; Price et al, 2012), nuclear reverse transcription cannot be resumed upon NEV withdrawal, indicating their major role in nuclear reverse transcription.

In addition, our data shows that viral nuclear condensates act as barriers against the host cell's innate immune responses, which would be triggered shortly after viral DNA exposure. Indeed, disassembly of CPSF6 condensates by PF74 resulted in a significant increase of cGAS activity. Notably, under our experimental conditions, the effect on innate immunity activation at 48 h p.i. is only due to those viruses that are reverse transcribing a 48 h p.i. Thus, if all viruses, once reverse transcribed, are no longer protected by HIV-1-MLOs, the impact on viral replication could be dramatic. Another key aspect is the recent evidence that the reverse transcription can occur in the nucleus (Rensen et al, 2021) as well as the significant presence of cGAS in the cell nucleus (Hao et al, 2023; Volkman et al, 2019; Wu et al, 2022; Zhao et al, 2020) suggests the potential for viral sensing within the nucleus. In the case of HIV-2, cGAS is involved in recognizing DNA within the

nucleus, with assistance from the host factor NONO (non-POU domain-containing octamer binding protein) (Lahaye et al, 2018). This nuclear localization of cGAS implies that the detection of viral DNA could be a broader phenomenon, especially since many DNA viruses and retroviruses replicate in the nucleus, where their DNA may be more accessible during integration and replication phases, and it could be shielded by nuclear condensates. Apart from NONO, other host proteins may also play a role in virus detection, including the polyglutamine binding protein 1 (PQBP1), which is required for cGAS recruitment to incoming virus particles (Yoh et al, 2022). The recognition of viral DNA by cGAS appears to be a complex process, as cGAS, NONO, and PQBP1 are physically linked through a nuclear ribonucleoprotein complex containing the long non-coding RNA NEAT1 (Morchikh et al, 2017), which regulates cGAS–STING activation (Hopfner and Hornung, 2020). NONO and NEAT1 are located within paraspeckles, and a recent study also identify PQPB1 as a component of these nuclear bodies (Dyakov et al, 2024), which are closely associated with HIV-1-MLOs (Rensen et al, 2021). As a result, it is conceivable that the disruption of HIV-1-MLOs could facilitate the binding of multiple components, leading to the activation of the innate immune response. Notably, it has been observed that the dissociation of cGAS from the nucleosome, facilitated by the DNA-repair complex MRN11, might contribute to this process by enabling the mobilization of cGAS from genomic DNA to viral DNA in mitotic cells (Cho et al, 2024). Interestingly, in macrophages, where cGAS is crucial for fighting infections, cGAS predominantly resides in the nucleus (Volkman et al, 2019), despite the fact that these cells are not undergoing mitosis. That said, studying the underlying mechanisms of cGAS activation following the disassembly of HIV-1-MLOs is highly complex and necessitates a careful evaluation of multiple factors. One crucial aspect is the need to track endogenous proteins, which calls for further detailed investigation. While our study provides valuable insights, it also opens new avenues for more advanced research into the mechanisms of cGAS activation during the disassembly process of HIV-1-MLOs. Of note, our results collectively suggest that HIV-1-MLOs play a critical role as a shield against innate immunity, ensuring a canonical RT in a nuclear location that facilitates integration into speckle-associated chromatin domains (SPADs) (Francis et al, 2020; Scoca et al, 2022). Conversely, while the absence of CPSF6 does not hinder RT, it has been linked to a significant fitness disadvantage to the virus (Saito et al, 2016), likely due to the compromised protection of viral DNA from host innate immunity factors or their misregulation. In addition, our results could significantly contribute to the

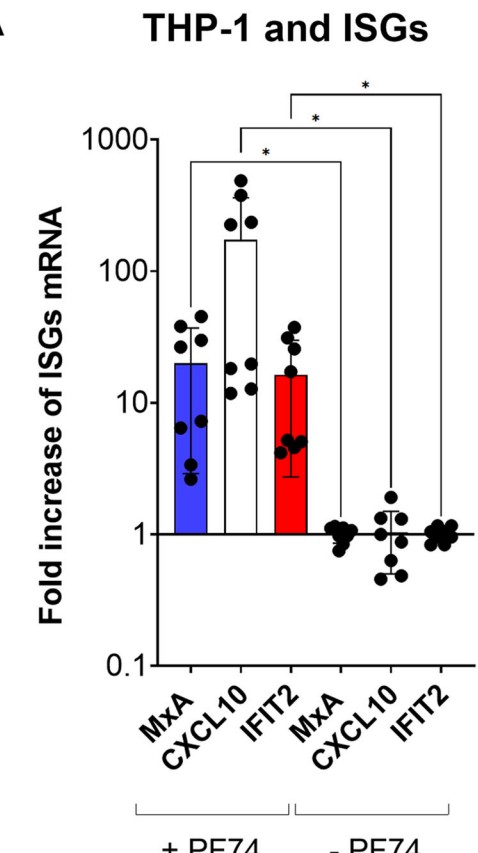

**A**     **THP-1 and ISGs**

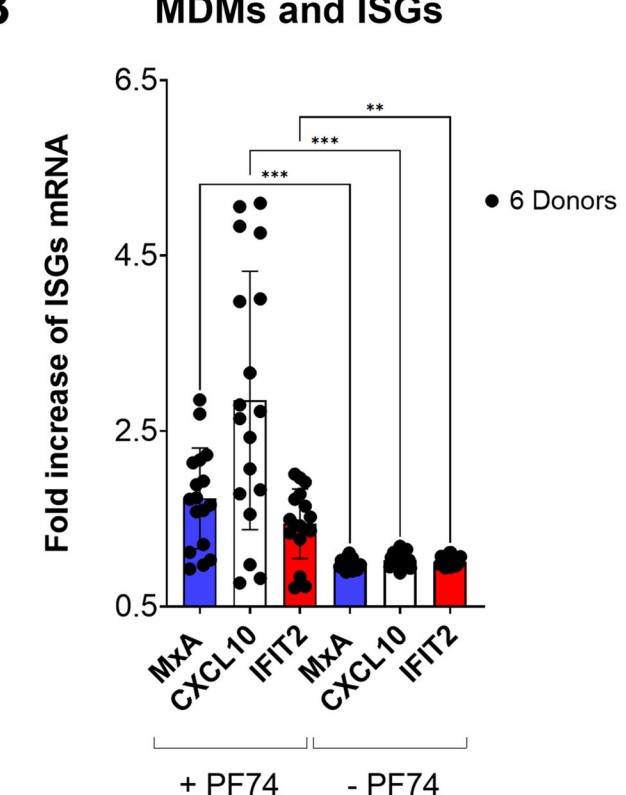

**B**     **MDMs and ISGs**

**Figure 8. Dissolution of CPSF6 condensates induces the activation of ISGs in both THP-1 cells and primary infected cells.**

(A) ISGs induced after PF74 treatment. Twenty-four hours after treatment with PF74, or without any treatment, HIV-1 infected THP-1 cells (MOI 10) were analyzed for the expression of interferon-stimulated genes (ISGs) by RT-PCR: MxA, CXCL10, and IFIT-2. Fold increase RT-PCR data on amplicons derived from infected cells treated with PF74 in comparison to infected untreated cells are displayed. Statistical analysis was performed using one-way ANOVA test; MxA * indicates a *p*-value = 0.0163; CXCL10 *$p$ value = 0.0354; IFIT2 *$p$ value = 0.0152. Data derive from two independent experiments with four replicates for each and shown as the mean ± SD. (B) MDMs were infected with MOI 30 and treated or not with high dose of PF74. The cellular RNA extracted was used as template to amplify ISGs by RT-PCR: MxA, CXCL10, and IFIT-2. Fold increase RT-PCR data on amplicons derived from infected cells treated with PF74 in comparison to infected untreated cells from 6 healthy donors. Statistical analysis was performed using a one-way ANOVA test, data are shown as the mean ± SD of three replicates from each donor. MxA *** indicates a *p*-value = 0.000944, CXCL10 *** indicates a *p*-value = 0.000969 and IFIT2 ** indicates a *p*-value = 0.00478. Source data are available online for this figure.

understanding of the mechanism of action of Lenacapavir, a derivative of PF-74 and the first-in-class anti-viral capsid drug recently approved by the FDA (Segal-Maurer et al, 2022). Lenacapavir could potentially enhance patient protection against viral infections by facilitating the exposure of the viral DNA to DNA sensors.

Our study highlights the crucial role of nuclear viral condensates in safeguarding the viral genome from innate immunity and enabling functional viral infection, serving as major hubs for nuclear viral reverse transcription. These findings provide new insights into the mechanisms and significance of HIV-1-MLOs, emphasizing their potential as therapeutic targets by disrupting their protective function.

## Limitations of the study

While our study has introduced new perspectives on the interaction of PF74 with viral cores within CPSF6 clusters, it also encounters specific limitations that must be acknowledged. The precise effects of PF74 at the atomic level, particularly concerning potential structural modifications within these clusters, remain partially unresolved due to the atomic-scale resolution limitations of our current imaging techniques.

Future studies employing cryo-electron microscopy (cryo-EM) could provide the necessary resolution to detect minor structural changes in the capsid that occur within CPSF6 clusters.

## Methods

### Reagents and tools table

| Reagent/Resource | Reference or Source | Identifier or Catalog Number |
|---|---|---|
| **Experimental models** | | |
| BRGS HIS (*M. musculus*) | Li et al, 2018 (PMID30065364) | N/A |
| HEK-293T cells (*H. sapiens*) | ATCC | CRL-3216™ |
| Primary CD34+ cells (*H. sapiens*) | This study | N/A |
| THP-1 cells (*H. sapiens*) | ATCC | TIB-202™ |
| **Recombinant DNA** | | |
| pCMV-VSV-G | Addgene | #8454 |
| pHIV-1(BRU)-Δenv-IN(HA) | Petit et al, 1999 (PMC112553) Petit et al, 2000 (PMC112230) | N/A |

| Reagent/Resource | Reference or Source | Identifier or Catalog Number |
|---|---|---|
| pHIV-1(BRU)-Δenv-IN(HA)-Δnef-MS2 | Scoca et al, 2022 (PMC10117160) | N/A |
| pHIV-1(BRU)-Δenv-Δnef-IRESGFP | Rensen et al, 2021 (PMC7780146) | N/A |
| pMS2-GFP-NLS | Tantale et al, 2016 (PMC4974459) | N/A |
| pSD-GP-NDK | PlasmidFactory | N/A |
| pSIV3+ | Durand et al, 2013 (PMC3536400) | N/A |
| **Antibodies** | | |
| Donkey anti-mouse IgG (CY3) | Jackson Lab | #715-165-150 |
| Donkey anti-rabbit IgG (CY3) | Jackson Lab | #711-165-152 |
| Goat-anti-mouse gold-conjugated (6 nm) | Aurion | #806.022 |
| Goat anti-rabbit IgG (A488) | Invitrogen | #A32731 |
| Goat anti-rabbit IgG (A647) | Invitrogen | #A32795 |
| Goat-anti-rabbit gold-conjugated (10 nm) | Aurion | #810.011 |
| Goat anti-rat IgG (A647) | Invitrogen | #A-21247 |
| Mouse anti-human CD3 (BV510) | BioLegend | #344828 |
| Mouse anti-human CD4 (APC) | BD Biosciences | #555349 |
| Mouse anti-human CD8/CD8a (BV785) | BioLegend | #103108 |
| Mouse anti-human CD19 (PE6CF594) | BD Biosciences | #562294 |
| Mouse anti-human CD45 (BUV395) | BD Biosciences | #563792 |
| Mouse anti-p24 | NIH reagent | #ARP-3537 |
| Rabbit anti-CPSF6 | Novus Biologicals | #NBP1-85676 |
| Rat anti-HA | Roche | #11867423001 |
| Rat anti-mouse CD45 (FITC) | BioLegend | #103108 |
| **Oligonucleotides and other sequence-based reagents** | | |
| Primary smiFISH probes against HIV-1 pol (x 24) | Integrated DNA Technologies | N/A |
| Secondary RNA FISH probe CY5 FLAP | Eurofins Genomics | N/A |

| Reagent/Resource | Reference or Source | Identifier or Catalog Number |
|---|---|---|
| **Chemicals, Enzymes and other reagents** | | |
| BD Cytofix Fixation Buffer | BD Biosciences | #554655 |
| Bovine Serum Albumin (BSA) | Sigma-Aldrich | #A9647 |
| Calcium Chloride solution | Sigma-Aldrich | #21115 |
| cOmplete, EDTA free (tablet) | Roche | #11873580001 |
| Deionized Formamide | Bio Basic | #FB0211 |
| DMEM (1X) + GlutaMAX™-I | Gibco | #31966 |
| DPBS (1X) | Gibco | #14190 |
| Ethylenediamine tetra-acetic acid disodium salt solution (EDTA) | Sigma-Aldrich | #E7889 |
| Ficoll-Paque™ PLUS | Cytiva | #17144003 |
| Fetal Bovine Serum (FBS) | Serana® | #S-FBS-SA-015 |
| Formaldehyde 4% | Sigma-Aldrich | #1004965000 |
| Gelatin | Merck | #104070 |
| Glutaraldehyde | Merck | #112179 |
| Glycine | Sigma-Aldrich | #G8898 |
| HEPES-buffered saline, pH 7.0 (2X for transfection) | Thermo Scientific Chemicals | #J62623.AK |
| HEPES solution (for cell culture) | Gibco | #15630 |
| Hoechst 33342 | Invitrogen | #H3570 |
| Human GM-CSF | Miltenyi Biotec | #130-093-862 |
| Human M-CSF | Miltenyi Biotec | #130-093-963 |
| Methylcellulose | Sigma-Aldrich | #M6385-250G |
| MEM Non-essential Amino Acid Solution | Sigma-Aldrich | #M7145 |
| NEBuffer™ 3 | New England BioLabs | #B7003S |
| Nevirapine | Sigma-Aldrich | #SML0097 |
| Paraformaldehyde 32% Aqueous Solution | Electron Microscopy Sciences | #15714 |
| PBS (RNase-free) | Invitrogen | #AM9625 |
| PBS | Gibco | #18912-014 |
| Penicillin Streptomycin (P/S) | Gibco | #15140 |
| PF74 | Sigma-Aldrich | #SML0835 |
| Phorbol 12-myristate 13-acetate (PMA) | Sigma-Aldrich | #P8139 |
| ProLong™ Diamond Antifade Mountant | Invitrogen | #P36970 |
| RIPA buffer | Sigma-Aldrich | #R0278 |
| RNaseZAP™ | Sigma-Aldrich | #R2020 |
| RPMI Medium 1640 (1X) + GlutaMAX™-I | Gibco | #61870 |
| Sodium Pyruvate | Gibco | #11360 |

| Reagent/Resource | Reference or Source | Identifier or Catalog Number |
|---|---|---|
| Stellaris® RNA FISH Hyb. Buffer | Biosearch™ Technologies | #SMF-HB1 |
| Stellaris® RNA FISH Wash Buffer A | Biosearch™ Technologies | #SMF-WA1 |
| Stellaris® RNA FISH Wash Buffer B | Biosearch™ Technologies | #SMF-WB1 |
| Sucrose | Sigma-Aldrich | #S0389-1KG |
| Triton™ X-100 | Sigma-Aldrich | #T8787 |
| Trypsin 0.05%-EDTA (1X) | Gibco | #253000 |
| TURBO DNase™ | Invitrogen | #AM2238 |
| UltraPure™ Distilled Water ($H_2O$) | Invitrogen | #10977 |
| Uranyl acetate | Merck | #8473 |
| Vanadyl Ribonucleoside Complexes (VRC) | Sigma-Aldrich | #94742 |
| 5-Ethynyl-2′-deoxyuridine (EdU) | Invitrogen | #A10044 |
| **Software** | | |
| Digital micrograph suite (V3.43.3213.0) | https://www.gatan.com/products/tem-analysis/gatan-microscopy-suite-software | |
| SerialEM (v4.0.11) | University of Colorado, https://bio3d.colorado.edu/SerialEM/ | RRID:SCR_017293 |
| IMOD (v4.11.64) | University of Colorado, https://bio3d.colorado.edu/imod/ | RRID:SCR_003297 |
| UCSF ChimeraX (v1.7.1) | UCSF, https://www.cgl.ucsf.edu/chimerax/ | RRID:SCR_015872 |
| Ilastik (v1.4.0) | https://www.ilastik.org/ | RRID:SCR_015246 |
| Fiji (v1.54f) | https://imagej.net/software/fiji/ Schindelin et al, 2012 (PMC3855844) | |
| FlowJo™ Software (v10.8.1) | https://www.flowjo.com/ BD Life Sciences | |
| Icy (v2.5.2) | https://icy.bioimageanalysis.org/download/ de Chaumont et al, 2012 (PMID22743774) | |
| Prism (v10.2.2) | https://www.graphpad.com/scientific-software/prism/ GraphPad Software | |
| **Other** | | |
| CD14+ Magnetic beads | Miltenyi Biotec | #130-050-201 |
| Click-iT™ Plus Alexa Fluor™ 647 picolyl azide toolkit | Invitrogen | #C10643 |
| Ficoll Paque Plus density gradient | Cytiva | #17-1440-03 |

| Reagent/Resource | Reference or Source | Identifier or Catalog Number |
|---|---|---|
| QIAamp® DNA Micro Kit | QIAGEN | #56304 |
| Rneasy Mini Kit | QIAGEN | #74106 |
| SepMate™-50 | Stemcell Technologies | #85450 |
| SuperScript™ III Platinum™ SYBR™ Green One-Step qRT-PCR | Invitrogen | #11736059 |
| 2',3'-Cyclic GAMP Enzyme Immunoassay Kit | Arbor Assays | #K067-H1 |
| Centrifuge | Thermo Scientific | Sorvall ST4 Plus |
| BD SYRINGE 60 ml (no needle) | Dutscher | #309653 |
| Confocal microscope | Zeiss | LSM700 inverted |
| Confocal microscope 63´ objective | Zeiss | Objective Plan-Apochromat 63x/1.4 Oil DIC M27 |
| EVOS® LED cube, GFP | Life Technologies | #AMEP4651 |
| EVOS® LED cube, RFP | Life Technologies | #AMEP4652 |
| EVOS® Live microscope | ThermoFisher Scientific | EVOS® FL Color |
| EVOS® microscope 10´ objective | ThermoFisher Scientific | EVOS™ 10X Objective, fluorite, LWD, 0.30NA/7.13WD |
| Falcon® Cell culture 24-well plate | Dutscher | #353047 |
| Flow Cytometer | ThermoFisher Scientific | Attune NxT Flow Cytometer |
| Formvar/carbon-coated nickel grids | Delta microscopies | #G150HEX-Ni |
| Glasstic™ Slide 10 with Grids | KOVA™ | #87144E |
| High tilt Holder | JEOL | #EM-21311 HTR |
| Transmission electron microscope 120 kV JEM-1400Plus | JEOL | JEM-1400Plus |
| Transmission electron microscope 200 kV Tecnai F20 (FEG-TEM) | FEI | Tecnai F20 D2216, Institut Pasteur |
| Falcon2 direct electron detector | FEI | Institut Pasteur |
| LS Columns | Miltenyi Biotec | #130-042-401 |
| Minisart™ NML Syringe Filter 0.45 μm | Sartorius | #16555 |
| MS Columns | Miltenyi Biotec | #130-042-201 |
| OctoMACS™ Separator | Miltenyi Biotec | #130-042-109 |
| OneView camera | Gatan | CMOS 4K camera |
| Open-Top Thinwall Polypropylene Konical Tube, 31.5 mL, 25 ´ 89 mm | Beckman Coulter, Inc. | #358126 |
| OPTILUX Petri dish - 100 × 20 mm | Dutscher | #353003 |
| Plate reader | Biorad | PR3100 TSC |

| Reagent/Resource | Reference or Source | Identifier or Catalog Number |
|---|---|---|
| Pre-Separation Filter 30 μm | Miltenyi Biotec | #130-041-407 |
| Precision cover glasses 12 mm Æ thickness No. 1.5H | Marienfeld | #0117520 |
| QuadroMACS™ Separator | Miltenyi Biotec | #130-090-976 |
| Refrigerated benchtop centrifuge | Eppendorf | Centrifuge 5415 R |
| Rotor for LE-80K | Beckman Coulter, Inc. | SW32Ti |
| Star-Frost slides 76 × 26 mm | Dutscher | #100204 |
| Stericup Quick Release-HV Sterile Vacuum Filtration System 0.45 μm | Millipore | #S2HVU05RE |
| Thermocycler | Eppendorf | Mastercycler® nexus |
| Thermocycler (qPCR) | Eppendorf | Mastercycler® realplex2 |
| Ultracentrifuge | Beckman Coulter, Inc. | Optima™ LE-80K |
| Cryo-Ultramicrotome | Leica | Leica EM UC7 FC7 |

## Methods and protocols

### *Human embryonic kidney 293 cells (HEK293T)*

HEK293T cells (obtained from the American Type Culture Collection (ATCC)) are human epithelial-like cells isolated from the kidney of a patient. They are used to produce and titer LVs and HIV-1 viruses. HEK293T cells were cultured at 37 °C in 5% $CO_2$ in Dulbecco's modified Eagle medium (DMEM) supplemented with GlutaMAX™-I, 10% fetal bovine serum (FBS), and 1% penicillin/streptomycin (P/S) (100 U/mL). Cells were negative to the mycoplasma test.

## THP-1 cells

THP-1 cells (obtained from the ATCC) are human monocytes isolated from the peripheral blood of an acute monocytic leukemia patient. Seeded THP-1 cells differentiate into macrophage-like cells within 48 h under phorbol 12-myristate 13-acetate (PMA) treatment (160 nM). THP1 cells were cultured at 37 °C in 5% $CO_2$ in RPMI Medium 1640 supplemented with GlutaMAX™-I, 10% FBS, and 1% P/S (100 U/mL). During long-term infections (experiments lasting over 10 days), the cells were cultured in RPMI Medium 1640 supplemented with GlutaMAX™-I, MEM Non-essential Amino Acid Solution (0.1 mM), Sodium Pyruvate (1 mM), HEPES buffer solution (10 mM), 10% FBS, and 1% P/S (100 U/mL). Cells were negative to the mycoplasma test.

## CPSF6 KO THP-1 cells

To target CPSF6, we used three distinct crRNAs with specific sequences—5'-TCGGGCAAATGGCCAGTCAAAGG-3', 5'-AGGA

CGGGGGCCGTTTTCCAGGGG-3', and 5'-CATGTAATCTCGG TCTTCTGGGG-3'—all supplied by Integrated DNA Technologies (IDT). We employed a pre-designed, non-specific crRNA from IDT as a control. Both crRNA and tracrRNA were prepared in IDT Duplex Buffer as per the manufacturer's instructions. On the nucleofection day, equimolar concentrations of crRNA and tracrRNA were mixed and annealed at 95 °C for 5 min to form duplexes. These RNA duplexes were then combined at a 1:2 ratio with TrueCut™ Cas9 Protein v2 (Thermo Fisher) for 10 min at room temperature, creating ribonucleoprotein (RNP) complexes. We resuspended $2 \times 10^5$ THP1 cells in P3 Primary Cell Nucleofector™ Solution (Lonza), mixed them with the RNP complexes and Alt-R® Cas9 Electroporation Enhancer (90 pmol, IDT), and nucleofected them using the 4D-Nucleofector™ System (Lonza) with the P3 Primary Cell 4D-Nucleofector™X Kit S (program FI-110). After nucleofection, the cells were cultured in complete RPMI medium with 20% FBS. Three days later, the cells were prepared for clonal selection by diluting in RPMI medium containing 20% FBS and plating in five 96-well plates at densities of 1 and 5 cells per well. One month later, selected microcolonies (50–100) were transferred to 24-well plates. As the wells neared confluence, the cells were transferred to 6-well plates. After another month of growth, the cells were processed for western blot analysis.

## PBMC purification and MDM differentiation

Peripheral blood mononuclear cells (PBMC) were purified from 40 ml of blood from healthy donors by Ficoll-Hypaque centrifugation using SepMate-50 tubes. CD14+ cells were purified from PBMC, isolated using positive magnetic sorting, and differentiated into MDM using M-CSF (20 ng/ml) in RPMI-1640 medium supplemented with GlutaMAX-I (Gibco), 10% fetal bovine serum, 1% penicillin-streptomycin (v/v) during 72 h then without M-CSF during 96 h. Human peripheral blood samples were collected from healthy volunteers through the ICAReB-Clin (Clinical Investigation platform) of the Pasteur Institute. All participants received oral and written information about the research and gave written informed consent in the frame of the healthy volunteers COSIPOP cohort after approval of the CPP Est II Ethics Committee (2023, Feb 20th).

## Lentiviral vectors (LV) and HIV-1 viruses

The non-replicative VSV-G ΔEnv HIV-1 (BRU) IN(HA) virus where the IN protein is fused to the HA tag was produced by co-transfecting the HEK293T cells with the HIV-1 (BRU) ΔEnv IN(HA) plasmid with the pCMV-VSV-G plasmid. The MCP-GFP lentiviral vector was produced by co-transfecting the HEK293T cells with the phage MS2 GFP NLS plasmid kindly gifted by Edouard Bertrand with plasmids that allow the expression of the VSV-G and of the gag-pol. The non-replicative VSV-G ΔEnv HIV-1 (BRU) IN(HA) ΔNef MS2 virus was produced by co-transfecting the HEK293T cells with the previously described HIV-1 (BRU) ΔEnv IN(HA) ΔNef MS2 plasmid. The non-replicative VSV-G ΔEnv HIV-1 (NL4.3) ΔNef GFP virus was produced by co-transfecting the HEK293T cells with the previously described HIV-1 (NL4.3) ΔEnv ΔNef IRES GFP plasmid with the VSV-G plasmid. HIV-1 (NL4.3-AD8) has been obtained from the NIH AIDS reagents.

All work with LVs and non-replicative HIV-1 viruses (VSV-G HIV-1 ΔEnv viruses) was performed under Biosafety Level 2 Plus (BSL2+) laboratory conditions, and all work with replicative HIV-1 viruses was performed under Biosafety Level 3 (BSL3) laboratory conditions. LVs and HIV-1 viruses were produced by transient transfection of HEK293T cells through calcium chloride coprecipitation. LVs were produced using a second-generation packaging system: $6 \times 10^6$ cells were co-transfected in dishes of 10 cm in diameter with the transfer vector (10 μg per cell), the pSD-GP-NDK packaging plasmid encoding for codon-optimized gag-pol-tat-rev (10 μg per cell), and the pCMV-VSV-G envelope plasmid (2.5 μg per cell). VSV-G HIV-1 ΔEnv viruses were produced by co-transfection of the cells with an HIV-1 ΔEnv plasmid (10 μg) and the pCMV-VSV-G envelope plasmid (2.5 μg). Replicative HIV-1 viruses were produced by transfection of the cells with an HIV-1 plasmid (10 μg). In addition, HIV-1 VSV-G pseudotyped viruses destined to infect THP-1 cells were produced in co-transfection with the pSIV3+ plasmid encoding for the $SIV_{MAC}$ protein Vpx (3 μg) kindly gifted by Andrea Cimarelli and VSV-G (2.5 μg). Then, the supernatant was collected 48 h post-transfection of the HEK293T cells, and lentiviral particles were concentrated by ultracentrifugation (22,000 rpm) at 4 °C for 1 h before being stored at −80 °C. All viruses and lentiviral vectors were tittered by qPCR using primers specific to the U5/R (viral DNA) and the CD3 (cellular DNA) sequences (see sequences in Appendix Table S1). LVs and HIV-1 viruses were tittered in 293T cells or THP-1 cells.

## Western blot

Proteins were isolated from THP-1 cells on ice using RIPA buffer, which includes 20 mM HEPES (pH 7.6), 150 mM NaCl, 1% sodium deoxycholate, 1% Nonidet P-40, 0.1% SDS, 2 mM EDTA, and a complete protease inhibitor from Roche Diagnostics. The protein concentration was measured using the Pierce BCA Protein Assay Kit (Thermo Fisher), a detergent-compatible (DC) method, with bovine serum albumin (BSA) serving as the calibration standard. We loaded ten micrograms of total protein lysate onto 4–12% Bis-Tris SDS-PAGE gels from Invitrogen. An anti-CPSF6 antibody (dilution used 1:400) was utilized for detection, and an anti-actin antibody was used for normalization. The proteins were visualized using the ECL Plus Western blotting detection system from Thermo Fisher.

## Immunofluorescence

Cells were washed with room temperature PBS once and then fixed with 4% PFA (diluted in PBS) for 15 min at room temperature and finally washed twice in PBS. Cells were then treated with 0.15% glycine (in PBS) for 10 min and washed once with PBS. Next, cells were permeabilized by incubation with 0.5% Triton X-100 (in PBS) for 30 min and washed once with PBS. Cells were then incubated for 30 min with a blocking buffer (1% BSA in PBS). Afterward, the blocking buffer was removed, and cells were incubated for 1 h at room temperature (RT) in a dark humid chamber with primary antibodies in the blocking buffer. After 5 washes in the blocking buffer, cells were incubated for 45 min at RT in a dark humid chamber with the secondary antibodies conjugated with fluorophores in the blocking buffer. After 5 washes in PBS, cells were fluorescently labeled by incubation with 'Hoechst 33342' diluted 10,000 times in $H_2O$ for 5 min at RT in a dark humid chamber. The cells were then washed 5 times in $H_2O$ before the coverslips were mounted on microscopy slides using ProLong™ Diamond Antifade

mounting medium. The mounting medium was cured overnight at RT under the chemical hood and away from light.

## Confocal microscopy

Confocal microscopy images were acquired using the Zeiss LSM700 inverted confocal microscope equipped with a 63×/1.4 Plan Apochromat oil immersion objective using diode lasers with excitation wavelengths of 405 nm (for Hoechst), 488 nm (for CPSF6), 555 nm (for p24), and 639 nm (for the HA tag and therefore the IN) with 4× averaging. A pixel size of 0.07 µm was used, and Z-stacks were acquired using an optimal Z-spacing of 0.33 µm. All images were acquired with the same laser gain.

## Immuno RNA-FISH

After infection, cells were washed with room temperature PBS once and then fixed with 4% PFA (diluted in RNase-free PBS) for 15 min at room temperature and finally washed twice in RNase-free PBS. Before the start of the immuno-RNA FISH process, all surfaces and tools were sprayed with RNaseZAP™. Cells were then incubated in permeabilization/blocking buffer (1% BSA, 2 mM vanadyl ribonucleoside complexes (VRC), and 0.3% Triton X-100, in RNase-free PBS) for 1 h at RT. Next, the permeabilization/blocking buffer was removed, and cells were incubated for 1 h at RT in a dark humid chamber with a primary antibody against CPSF6 using an antibody dilution of 1:400 in the permeabilization/blocking buffer. After 5 washes in RNase-free PBS, cells were incubated for 45 min at RT in a dark humid chamber with the secondary antibody 'donkey anti-rabbit Cy3', using an antibody dilution of 1:300 in permeabilization/blocking buffer. Following immunostaining, cells were fixed with 4% PFA (in RNase-free PBS) for 10 min and washed twice in RNase-free PBS. During fixation, primary smiFISH probes (24 smiFISH probes designed against HIV-1 pol sequence, see sequences in the Appendix Table S2) were hybridized with a secondary FLAP probe conjugated to a Cy5 fluorophore (see sequence in the Appendix Table S2) using a thermocycler: 40 pmol of primary probes and 50 pmol of secondary FLAP probe, in 1X NEBuffer 3 (diluted in RNase-free H$_2$O), incubated at 85 °C for 3 min then at 65 °C for 3 min and finally at 25 °C for 5 min. After fixation, cells were incubated in Wash A buffer (20% Stellaris® RNA-FISH Wash Buffer A, 10% Deionized Formamide, in RNase-free H$_2$O) at RT in a dark humid chamber for 5 min. Then, the Wash A buffer was removed, and cells were incubated overnight at 37 °C in a dark humid chamber with the hybridized smiFISH probes using a probe dilution of 1:50 in Hybridization buffer (90% Stellaris® RNA-FISH Hybridization Buffer, 10% Deionized Formamide) (50 µL of diluted probes per 12 mm size coverslip). The next day, the Hybridization buffer containing the probes was removed and cells were incubated in Wash A buffer at 37 °C in a dark humid chamber. After 30 min, the Wash A buffer was removed, and cells were incubated in 'Hoechst 33342' diluted 10,000 times in RNase-free H$_2$O at RT in a dark humid chamber for 5 min. Then the Hoechst was removed, and cells were incubated in Stellaris® RNA FISH Wash Buffer B at RT in a dark humid chamber for 5 min. Finally, the cells were washed 5 times in RNase-free H$_2$O before the coverslips were mounted on microscopy slides using ProLongTM Diamond Antifade mounting medium. The mounting medium was cured overnight at RT under the chemical hood and away from light.

*Confocal microscopy*: Confocal microscopy images were acquired using the Zeiss LSM700 inverted confocal microscope equipped with a 63×/1.4 Plan Apochromat oil immersion objective using diode lasers with excitation wavelengths of 405 nm (for Hoechst), 488 nm (for MCP-GFP), 555 nm (for CPSF6), and 639 nm (for the RNA) with 4× averaging. A pixel size of 0.07 µm was used, and Z-stacks were acquired using a Z-spacing of 0.33 µm. All images were acquired with the same laser gain.

## Image and association analysis

Images were analyzed using Fiji. The lower threshold of each channel was set using the negative control condition and the higher threshold of each channel was set using the condition with the highest signal (to avoid saturation). The association between proteins was determined by merging channels and observing overlapping signals. In addition, the association was also determined by overlapping intensity profiles. More specifically, the intensity profiles of each channel were measured using the 'Plot profile tool' in the 'Analyze' tab.

## Flow cytometry

Infected cells were detached from the wells with trypsin and fixed with BD CellFix (diluted 1/10 in PBS). Cells were then passed through an Attune NxT Flow Cytometer that evaluated their GFP expression levels. The raw data were then processed using the FlowJo™ software.

## Viral condensates persistence assay

To evaluate the stability of CPSF6 clusters in macrophages differentiated from THP-1 cells during HIV-1 infection we performed immunofluorescence and confocal microscopy as described below. *Infection with* VSV-G ΔEnv HIV-1 (BRU) IN(HA): 5 × 10$^5$ THP-1 cells were seeded on 12 mm ∅ No. 1.5H microscopy coverslips in a 24-well plate in complete growth medium (RPMI Medium 1640 GlutaMAX™-I, 10% FBS, and 1% P/S) and were differentiated with PMA (160 nM) at 37 °C in 5% CO$_2$ for 72 h. Once differentiated, the cells were infected with the VSV-G ΔEnv HIV-1 (BRU) IN(HA) virus at a multiplicity of infection (MOI) of 10. Following infection, cells were kept in culture in complete growth medium supplemented with PMA, treated with Nevirapine (10 µM), and incubated at 37 °C in 5% CO$_2$. Three days post-infection the culture media was changed, and cells were kept in culture in complete growth media without PMA and remained treated with Nevirapine always at 37 °C in 5% CO$_2$. The culture medium and Nevirapine were renewed every 2 to 3 days, until fixation. Samples were analyzed by immunofluorescence using confocal microscopy and images were analyzed by ImageJ (Fiji).

## Viral condensates/MCP-GFP association assay

To evaluate the association between CPSF6 and the MCP-GFP fusion protein in macrophages differentiated from THP-1 cells during HIV-1 infection, we performed immuno-RNA FISH and confocal microscopy as described below. Transduction with LV MCP-GFP: 5 × 10$^5$ THP-1 cells were seeded on 12 mm ∅ No. 1.5H microscopy coverslips in a 24-well plate in complete growth medium (RPMI Medium 1640 GlutaMAX™-I, 10% FBS, and 1% P/S) and were differentiated with

PMA (160 nM) at 37 °C in 5% $CO_2$ for 48 h. Once differentiated, the cells were transduced with the MCP-GFP lentiviral vector at an MOI of 2.5. Following transduction, cells were kept in complete growth medium supplemented with PMA and incubated at 37 °C in 5% $CO_2$. *Infection with VSV-G ΔEnv HIV-1 (BRU) IN(HA) ΔNef MS2*: the day after transduction, cells were infected with the VSV-G ΔEnv HIV-1 (BRU) IN(HA) ΔNef MS2 virus at an MOI of 30, in the presence or absence of Nevirapine (10 μM). Following infection, cells were kept in complete growth medium supplemented with PMA at 37 °C in 5% $CO_2$. The culture medium and Nevirapine were renewed 30 h post-infection.

## Immunogold labeling of cryosections according to Tokuyasu for immunoelectron microscopy

Twenty million cells were infected for seven days with VSV-G ΔEnv HIV-1 (BRU) IN(HA) (MOI 20) in the presence or absence of the NEV. Next, cells were fixed for 2 h with 4% paraformaldehyde in phosphate buffer (pH 7.6), washed with PBS (pH 7.6) for 2 × 5 min, and centrifuged at $300 \times g$ for 10 min. After removing the supernatant, cell pellets were included in gelatin 12% and infused with sucrose 2.3 M overnight at 4 °C. 80 nm ultrathin cryosections were made a −120 °C on a LEICA FC7 cryo-ultramicrotome (Leica, Wetzlar, Germany). Ultrathin sections were retrieved with Methylcellulose 2%/Sucrose 2.3 M mixture (1:1) and collected onto formvar/carbon-coated nickel grids. After the removal of gelatin at 37 °C, sections were incubated on drops of 1:100 anti-CPSF6 and 1:100 anti-p24 diluted in PBS. After six washes of 5 min each, grids were incubated on drops of PBS containing 1:30 goat-anti-rabbit (10 nm) and 1:30 goat-anti-mouse gold-conjugated (6 nm). Grids were finally washed with six drops of PBS (5 min each), post-fixed in 1% glutaraldehyde, and rinsed with three drops of distilled water. Contrasting step was performed by incubating grids on drops of uranyl acetate 2% (Agars Scientific, Stansted, UK)/methylcellulose 2% mixture (1:10). The sections were imaged in a transmission electron microscope at 120 kV (JEOL 1400 Plus, JEOL, Tokyo, Japan).

## Dual-axis electron tomography

Transmission Electron Microscopy took place either with a Tecnai F20 microscope operated at 200 kV and equipped with a Falcon2 direct electron detector, using SerialEM (RRID:SCR_017293), either with a JEOL microscope (JEM-1400Plus, Tokyo, Japan) operated at 120 kV and equipped with the OneView (Gatan) camera. In the case of SerialEM acquisitions, multiscale mapping of the grid took place to map positions of interest and collect the first tilt axis. The grid was then removed from the microscope, rotated manually at an approximate angle of 90 degrees, and reloaded to the microscope stage. A second grid map was collected in low resolution to map back the areas of interest and collect the second tilt axis in these areas. Dual-axis tilt series were collected by first step from −50° to +50° with a high tilt holder (JEOL, Tokyo, Japan) and managed by the TEM Tomography module in Digital Micrograph suite (Gatan, Pleasanton, CA, USA).

## Tomogram reconstruction, core annotation, classification, and visualization

Tilt angle information for each tilt series was extracted from Digital Micrograph (Gatan) header files and sorted with the help of a homemade script. Initial pairwise shifts of the tilt series were estimated using the tiltxcorr function from the IMOD software package (RRID:SCR_003297). Tilt series alignments were further optimized in IMOD using the tracing of the 10 nm immunogold beads present in the fields of view and by further optimizing the fiducial model to define the acquisition geometry. CTF correction was performed in cases where large defocus values had been used during the acquisitions. 3D reconstructions of unbinned data (pixel size = 4 Å) were calculated in IMOD by weighted back projection using the SIRT-like filter (12 iterations). The two volumes produced from the dual-axes tilt series were combined using the 10 nm CPSF6 immunogold beads. In case combinations of full volumes were difficult, the combined volumes were restricted to central areas that involved only the structures of interest.

Tomographic slices from the middle of the dual-axis tomographic volumes, where most of the cores were visible, were extracted, converted to tif using the mrc2tif function of IMOD, and imported to Ilastik (RRID: SCR_015246) for a 2D-pixel classification run. Intensity, edge, and texture features with sigma varying from 1 to 10 were selected. The assignment of thin labels took place as follows: one label (in magenta) to mark one dense core, one label (in blue) to mark one lighter core, one label (in yellow) for a ghost, and one label (in gray) for the nuclear background (Appendix Fig. S4E,F, left column). After a unique live prediction run for each 2D dataset using the initial labels assigned, tif snapshots were produced with the raw image pixels pseudo-colored according to the predictions. These tif snapshots were then converted to mrc files and assigned the correct pixel size as the original data using alterheader on IMOD. IMOD contours were used to manually trace the conical structures visible at the MLOs sites using the sculpt drawing tool of IMOD. Contours/Cores were assigned to 3 different objects according to the underlying 2D-pixel classification predictions/pseudo-coloring from the previous step (Appendix Fig. S4E,F, right column). Models of each dataset were then saved. All contours were transferred to the correct z value of the volume, from where the initial slice had been extracted, and re-opened on top of the volume data. The correctly assigned 2D contours in the 3D volume were then used to continue tracing the classified cores along z with the help of IMOD's spherical interpolation and by using manual corrections when needed.

Contours were then meshed into surfaces, and the 3 different objects were further sorted into separated surfaces using the imodsortsurf function of IMOD. CPSF6 immunogold beads were traced using the findbeads3d function of IMOD. Smaller CA beads were added manually as scattered points.

For visualization, volume data and imod surfaces were loaded to UCSF ChimeraX (RRID:SCR_015872). Imod surfaces were further used as masks to isolate subvolumes that correspond to the core interior. Beads were imported to ChimeraX as contours/points instead of surfaces.

## Tokuyasu sections for viral preparation and intensity analysis

Tilt series acquisitions of isolated mature virions prepared with the Tokuyasu method were collected with a Tecnai F20 microscope operated at 200 kV and equipped with a Falcon2 direct electron detector, using SerialEM (RRID:SCR_017293). Data were collected at a pixel size of 5.2 Å. Multiscale mapping of the grid took place to map positions of interest and collect the first tilt axis. The grid was then removed from the microscope, rotated manually at an

approximate angle of 90 degrees, and reloaded to the microscope stage. A second grid map was collected in low resolution to map back the areas of interest and collect the second tilt axis in these areas. Intensity analysis: Dual-axis tomograms were produced in full resolution using the default handling of intensity values in IMOD (RRID:SCR_003297) needed for volume combination. In the case of the Jeol/OneView tilt series acquisitions, 32-bit tomographic volumes were produced, while in the case of Tecnai/Falcon2 tilt series acquisitions, 16-bit tomographic volumes were produced. The 32-bit tomograms were converted to 16-bit tomograms and scaled to a common mean and standard deviation using new stack. The middle slice of representative cores was cropped and extracted from the tomograms and IMOD contours were prepared to indicate the core limits. Imodmop was then used to mask out the internal and external areas as indicated in Fig. EV3A,B. Intensity statistics were extracted using the clip stats function of IMOD (RRID:SCR_003297).

## Analysis for the assignment of core classes

Tomographic slices from the middle of the tomographic volume, where most of the cores were visible, were extracted, converted to tif using the mrc2tif function of IMOD, and imported to Ilastik for a 2D-pixel classification run. Intensity, edge, and texture features with sigma varying from 1 to 10 were selected. The assignment of thin labels took place as follows: one label (in magenta) to mark one dense core, one label (in blue) to mark one lighter core, one label (in yellow) for a ghost, and one label (in gray) for the nuclear background (Supplementary Figure R2.2, left column). After a unique live prediction run for each 2D dataset with the labels, tif snapshots were produced with the raw image pixels pseudo-colored according to the predictions. These tif snapshots were then converted to mrc files and assigned the correct pixel size as the original data using alterheader. IMOD contours were used to manually trace the conical structures visible at the MLOs sites. Contours/Cores were assigned to 3 different objects according to the underlying 2D-pixel classification predictions/pseudo-coloring from the previous step (Supplementary Figure R2.2, right column). Models of each dataset were then saved, and this time re-opened on top of their corresponding 3D tomographic volumes. All contours were transferred to the correct z value of the volume, from where the initial slice had been extracted. To achieve this, contours were copied through 3dmod's GUI to the correct z, and with the subsequent use of clipmodel of IMOD, the initial z = 0 contours were removed. The correctly assigned 2D contours in the 3D volume were then used to continue tracing the classified cores along z with the help of IMOD's spherical interpolation and by manual corrections when needed.

## CPSF6 cluster disassembly assay

To understand the impact of CPSF6 clusters in nuclear reverse transcription in macrophages differentiated from THP1, we performed flow cytometry, immunofluorescence and confocal microscopy, and late/early viral reverse transcript (LRT/ERT) qPCR.

## LRT/ERT qPCR

Three days after PF74 treatment, the cell's DNA was extracted using QIAGEN's 'QIAamp® DNA Micro Kit', following the manufacturer's

protocol intended for extracting DNA from small volumes of blood. The LRT, ERT, and actin sequences were then amplified using the primers listed in Appendix Table S3 and the SYBR green dye or probe. The qPCR cycle was as follows: 15 min at 95 °C, 40 cycles (15 s at 95 °C, 30 s at 58 °C, 30 s at 72 °C), 15 s at 95 °C, 15 s at 55 °C, 20 min melting curve until reaching 95 °C, 15 s at 90 °C.

## RT qPCR

Total viral RNA was extracted from the cells using the RNeasy Mini Kit, following the manufacturer's protocol, and eluted into 60 µL of DEPC. The quantities of CXCL10, IFIT2, and MxA RNA in 10 ng of total RNA were determined via the SuperScript™ III Platinum™ SYBR™ Green One-Step qRT-PCR Kit. Amplicons from each sample were analyzed against a standard curve, which was run in parallel using total RNA from uninfected cells. The results were normalized to the RNA quantity from the actin gene.

## Viral DNA staining with EdU/Click-chemistry

To visualize viral DNA in macrophage-like cells, we added 5-Ethynyl-2′-deoxyuridine (EdU) 5 µM during the entire time of the infection. Cells were washed with room temperature PBS once and then fixed with 4% formaldehyde for 15 min at room temperature and finally PBS-BSA 3%. Cells were then permeabilized by incubation with 0.5% Triton X-100 (in PBS) for 20 min and washed once with PBS-BSA 3%. Cells were then incubated for 30 min with Click-iT® Plus reaction cocktail prepared following the manufacturer's protocol and then washed 5 times in PBS-BSA 3%. The cells were then incubated for 15 min with blocking buffer (3% BSA in PBS) and washed 5 times in PBS-BSA 1%. Afterward, cells were incubated for 1 h at room temperature (RT) in a dark humid chamber with a primary antibody against CPSF6 diluted 1:400 in blocking buffer. After 5 washes in the blocking buffer, cells were incubated for 45 min at RT in a dark humid chamber with the secondary antibody 'cy3 anti-rabbit' diluted 1:300 in the blocking buffer. After 5 washes in PBS, cells were fluorescently labeled by incubation with 'Hoechst 33342' diluted 10,000 times in $H_2O$ for 5 min at RT in a dark humid chamber. The cells were then washed 5 times in $H_2O$ before the coverslips were mounted on microscopy slides using ProLong™ Diamond Antifade mounting medium and incubated overnight at RT under the chemical hood and away from light. *Confocal microscopy*: Confocal microscopy images were acquired using the Zeiss LSM700 inverted confocal microscope equipped with a 63×/1.4 Plan Apochromat oil immersion objective using diode lasers with excitation wavelengths of 405 nm (for Hoechst), 555 nm (for CPSF6), and 639 nm (for viral DNA) with 4× averaging. A pixel size of 0.07 µm was used. All images were acquired with the same laser gain. Images were analyzed using Fiji.

## cGAMP ELISA assay

To evaluate cGAMP levels following CPSF6 cluster disassembly, we performed a cGAMP ELISA assay. Briefly, $20 \times 10^6$ THP-1 cells were seeded in 10 cm culture dishes in complete growth medium (RPMI Medium 1640 GlutaMAXTM-I, 10% FBS, and 1% P/S) supplemented with MEM Non-essential Amino Acid Solution (0.1 mM), Sodium Pyruvate (1 mM), HEPES buffer solution (10 mM), and were differentiated with PMA (160 nM) at 37 °C in

5% $CO_2$ for 48 h. Once differentiated, the cells were infected with the VSV-G pseudotyped ΔEnv HIV-1 (BRU) IN(HA) virus at an MOI of 10. Forty-eight hours after infection, media was renewed, and certain cells were treated with PF74 (25 µM).

After PF74 treatment, cells were washed once with PBS and lysed using RIPA buffer (120 µL per dish, spread with a cell scraper) supplemented with a 'cOmplete, EDTA-free' tablet (one tablet in 10 mL of buffer) for 15 min at 4 °C. The lysates were centrifuged for 15 min at 13,000 rpm in a refrigerated benchtop centrifuge kept at 4 °C to eliminate debris. The ELISA assay was then performed following the manufacturer's protocol, using the lysis buffer as a diluent for the samples and the curve. Each well was loaded with 0.05 to 0.1 ng of protein. The plate was revealed using the BIORAD PR3100 TSC. cGAMP values were normalized by total proteins for each sample analyzed.

## Viral reactivation assay

To evaluate the rebound capacity of incoming vRNA located in stable CPSF6 clusters in macrophages differentiated from THP-1 cells during HIV-1 infection after temporary reverse transcription inhibition, we performed a live imaging experiment. $5 \times 10^5$ THP-1 cells were seeded on 12 mm Ø No. 1.5H microscopy coverslips in a 24-well plate in complete growth medium (RPMI Medium 1640 GlutaMAX™-I, 10% FBS, and 1% P/S) supplemented with MEM Non-essential Amino Acid Solution (0.1 mM), Sodium Pyruvate (1 mM), HEPES buffer solution (10 mM), and were differentiated with PMA (160 nM) at 37 °C in 5% $CO_2$ for 48 h. Once differentiated, the cells were infected with the VSV-G ΔEnv HIV-1 (NL4.3) ΔNef GFP virus at an MOI of 15 and were treated with Nevirapine (10 µM). Prior to infection, the virus was incubated with the TURBO DNase™ enzyme for 1 h at 37 °C to remove any plasmid DNA remaining from the viral production process. Following infection, cells were incubated in complete growth medium supplemented with PMA, MEM Non-essential Amino Acid solution, Sodium Pyruvate, and HEPES buffer solution, treated with Nevirapine (10 µM), and kept at 37 °C in 5% $CO_2$. The supplemented media and Nevirapine were renewed every 24 h. Three days post-infection, Nevirapine treatment was interrupted (except in the control condition that required uninterrupted Nevirapine treatment throughout the whole experiment). Again, the supplemented media (and Nevirapine in concerned wells) was renewed every 24 h.

## Live imaging

Three, seven, and twelve days post-infection, cells were imaged at the EVOS® FL Color equipped with a 10×/0.3 Plan Fluor objective using 'EVOS® LED cube, GFP' for GFP (a marker of viral re-activation), EVOS® LED cube, RFP for RFP (autofluorescence control) and no cube for bright-field (BF). A pixel size of 0.9 µm was used. All images were acquired with the same laser gain. Images were analyzed using Fiji. The lower threshold of each channel was set using the negative control condition and the higher threshold of each channel was set using the condition with the highest signal (to avoid saturation).

## In vivo observation of nuclear condensates

To investigate the behavior of nuclear viral condensates under in vivo acute infection conditions in BRGS HIS mice, we retrieved

bone marrow cells from infected mice and performed immune RNA-FISH and confocal microscopy as described below.

## Generation of HIS mice and infection

Balb/c Rag2−/−Il2rg−/−SirpaNOD strain (BRGS) mice were used as recipients to create HIS mice as previously described (Li et al, 2018). Briefly, fetal liver CD34+ cells (Advanced Bioscience Resources Inc., USA) were isolated with affinity matrices according to the manufacturer's instructions (Miltenyi Biotec) and subsequently phenotyped for CD38 expression. Newborn (3- to 5-d-old) pups received sublethal irradiation (3 Gy) and were injected intrahepatically with $5–10 \times 10^4$ CD34+CD38− human fetal liver cells. All manipulations of HIS mice were carried out under laminar flow conditions. The HIV-1NLAD8 molecular clone was obtained from the AIDS Research and Reference Reagent Program, NIAID, NIH. NLAD8 viruses were generated by transfecting 293T cells with the pNLAD8 plasmid (Trans IT, Mirus). The virus was quantified using the Reed-Muench method on activated peripheral blood mononuclear cells, and the p24 antigen was measured using the Alliance HIV-1 p24 ELISA kit (PerkinElmer). BRGS mice were inoculated with $10^5$ tissue culture infective dose required for 50% infection (TCID50) of HIVNLAD8 through intraperitoneal injection and housed in isolators. Viral RNA was extracted from 140 µL of plasma (EDTA-harvested blood, Microvette CB300, Sarstedt) with the QIAamp viral RNA kit (Qiagen) and a lentiviral vector preparation (pFlap) has been used as the standard curve. qRT PCR has been performed with the kit SuperScript III Platinum One-Step (Invitrogen) using the primers and probes for late reverse transcripts.

## Flow cytometry on the blood of humanized mice infected with HIV-1 and uninfected

Whole blood collected in EDTA coated Microvettes was directly stained with fluorochrome-conjugated antibodies at 4 °C for 30 min, erythrocytes were lysed and samples fixed with FACSTM Lysing solution. Washing steps were done in FACS buffer with 2% fetal calf serum, 2 mM EDTA.

## Cell isolation from bone marrow of infected HIS mice

Femur and tibia bones were crushed with a mortar and pestle for cell extraction. The bone marrow was lysed with Hybri-Max red blood cell lysing buffer (Sigma). PBMCs were isolated using Ficoll and resuspended in MACs buffer. Cells were counted and resuspended in RPMI 10% FBS. Five million cells were grown in coverslips in 24-well plates. Two hours later the media was changed with RPMI 10% FBS + MCSF 10 ng/mL. Cells were incubated at 37 °C for 3 days and then the media was changed with RPMI 10% FBS without MCSF and cultured for another 4 days.

## Cell retrieval

Mononuclear cell (MNC) buffy coat isolation from the bone marrow of the mice was performed and MNCs were resuspended in complete growth medium (RPMI Medium 1640 GlutaMAX™-I, 10% FBS, and 1% P/S). Cells were seeded on 12 mm Ø No. 1.5H microscopy coverslips in a 24-well plate and incubated at 37 °C in 5% $CO_2$. After 2 h, suspension cells were washed away from the wells, and adherent cells were incubated in complete growth medium supplemented with

MCSF (10 ng/mL) at 37 °C in 5% $CO_2$. After 3 days, the culture media was removed, and cells were incubated for 4 more days in complete growth media without MCSF at 37 °C in 5% $CO_2$ before fixation. Immuno RNA-FISH has been conducted in cells derived from bone marrow and cell images were acquired using the Zeiss LSM700 inverted confocal microscope equipped with a 63×/1.4 Plan Apochromat oil immersion objective using diode lasers with excitation wavelengths of 405 nm (for Hoechst), 488 nm (for CPSF6), and 639 nm (for the RNA) with 4× averaging. A pixel size of 0.07 μm was used, and Z-stacks were acquired using a Z-spacing of 0.33 μm. All images were acquired with the same laser gain. Images were analyzed using Fiji. The lower threshold of each channel was set using the negative control condition and the higher threshold of each channel was set using the condition with the highest signal (to avoid saturation). The association between proteins was determined by merging channels and observing overlapping signals. In addition, the association was also determined by overlapping intensity profiles.

## Ethics statement

Animals were housed in isolators under pathogen-free conditions and received humane care. To minimize suffering, they were anesthetized with isoflurane. All experiments involving the generation and characterization of humanized mice were conducted in accordance with ethical guidelines. The study was approved by the ethical committee at the Institut Pasteur (CETEA-2013-0131) and validated by the French Ministry of Education and Research (Reference # 02162.01).

## Quantification and statistical analysis

All image acquisition and quantification were done blinded. Unpaired T-tests or ordinary one-way ANOVA tests were utilized to analyze the significance of differences in all the following figure panels presenting statistical analysis.

# Data availability

Immuno-gold/EM/Tomography data and segmentation maps are deposited in EMDB accession code(s): EMD-51462 and EMD-51466.

The source data of this paper are collected in the following database record: biostudies:S-SCDT-10_1038-S44318-024-00316-w.

# Peer review information

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

## Acknowledgements

FDN is supported by the Institut Pasteur and ANRS grants (ECTZ192036, ECTZ137593) and Sidaction grant AAP31-1-AEQ 12633. FM is supported by grants from ANRS (*ECTZ136619*) and from Région Centre Val de Loire (APR-IA-2021). AY is supported by the ANRS fellowship ECTZ204694. YT is supported by the Vaccine Research Institute (Université Paris-Est, Créteil, France). We are also grateful for support for equipment from the French Government Programme Investissements d'Avenir France BioImaging (FBI, N° ANR-10-INSB-04-01). We gratefully acknowledge the UtechS Photonic BioImaging platform (Imagopole), C2RT at Institut Pasteur. We thank the NIH AIDS Reagents program to support us with precious reagents and Addgene. Thanks to ICAReB-Clin of the Medical Direction and ICAReB-biobank of the Biological Resource Center of the Institute Pasteur-Paris for providing blood samples from healthy volunteers, managing the participants' visits and preparing the blood samples from donors.

## Author contributions

**Selen Ay**: Data curation; Formal analysis; Investigation; Methodology. **Julien Burlaud-Gaillard**: Investigation; Methodology. **Anastasia Gazi**: Formal analysis; Investigation; Methodology; Writing—review and editing. **Yevgeniy Tatirovsky**: Investigation; Methodology. **Celine Cuche**: Data curation; Investigation; Methodology. **Jean-Sebastien Diana**: Methodology. **Viviana Scoca**: Methodology. **James P Di Santo**: Supervision; Investigation. **Philippe Roingeard**: Supervision. **Fabrizio Mammano**: Supervision; Writing —review and editing. **Francesca Di Nunzio**: Conceptualization; Data curation; Formal analysis; Supervision; Funding acquisition; Validation; Investigation; Methodology; Writing—original draft; Project administration; Writing—review and editing.

Source data underlying figure panels in this paper may have individual authorship assigned. Where available, figure panel/source data authorship is listed in the following database record: biostudies:S-SCDT-10_1038-S44318-024-00316-w.

## Disclosure and competing interests statement

The authors declare no competing interests.

# Expanded View Figures

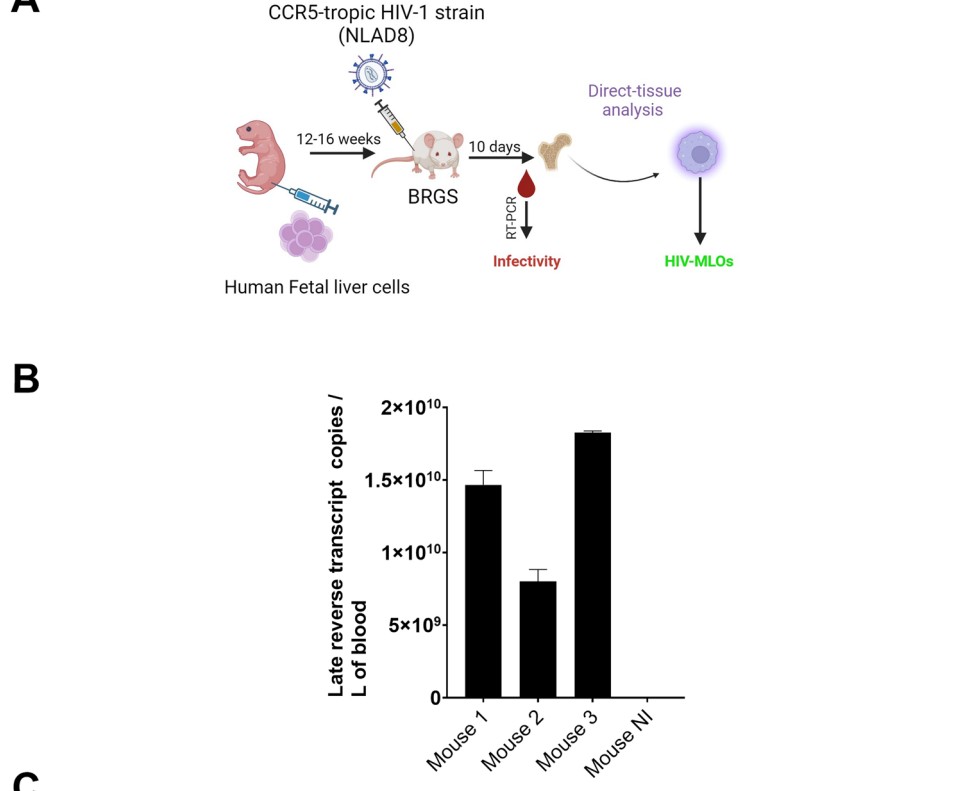

**Figure EV1.  HIV-1-MLOs build during in vivo infection, related to Fig. 3.**

(**A**) Schema of CCR5-tropic HIV-1 strain (NLAD8) infection in BRGS mice for 10 days. Bone marrow human CD4+ cells were immediately labeled and imaged without the need of cell culture passage. (**B**) Graph showing viral infectivity in BRGS mice by quantitative RT PCR (Data are shown as the mean ± SD of two replicates). (**C**) CD4+ cells derived from BM of infected BRGS mice were directly fixed and labeled for the detection of CPSF6 (in green). Nuclei were stained with Hoechst (in blue).

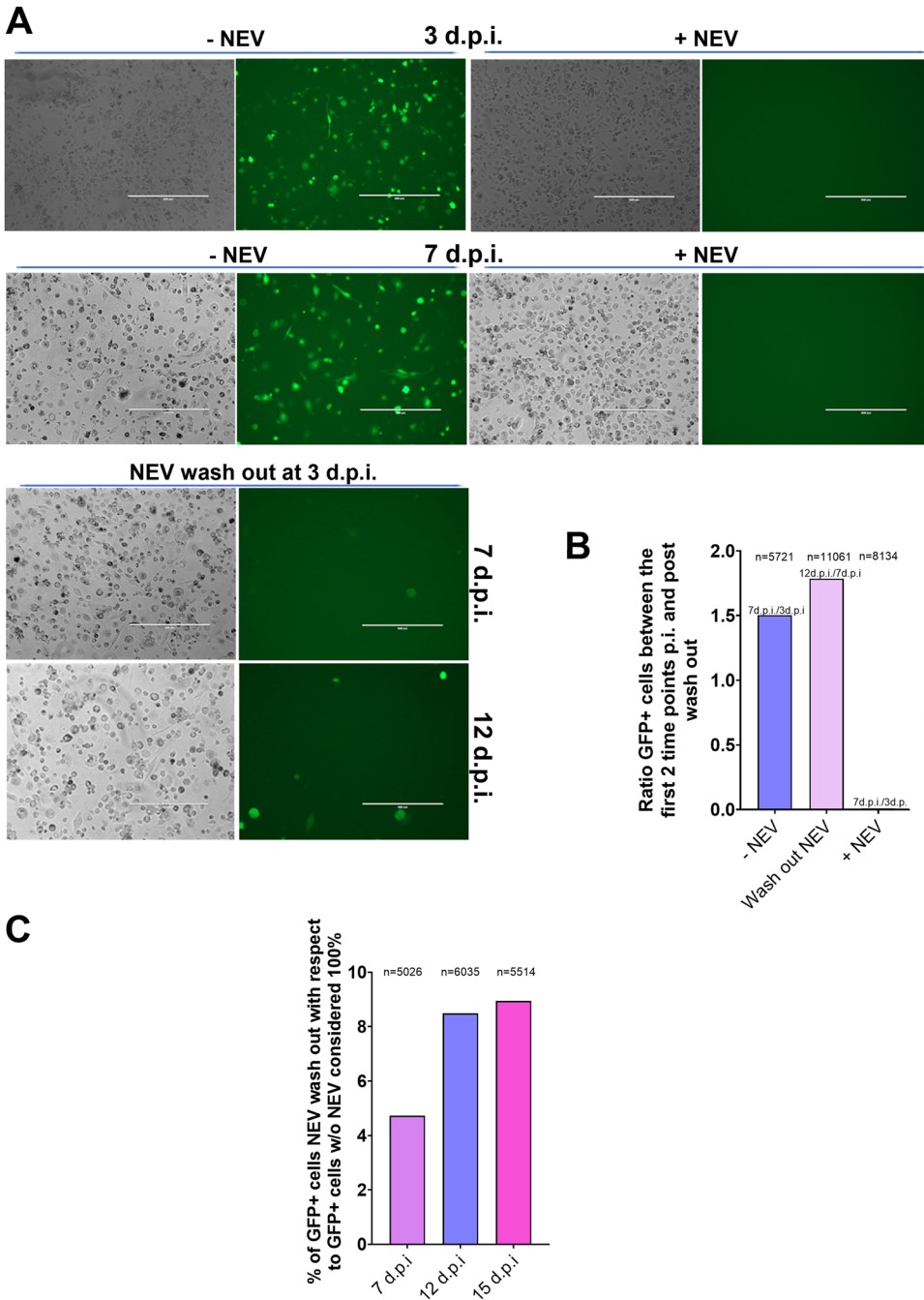

**Figure EV2.  Asynchronous viral reactivation in nuclear HIV-1-MLOs, related to Fig. 2 and 4.**

(A) THP-1 cells infected with HIV-1 ΔEnv pseudotyped with VSV-G and carrying the GFP as reporter gene at 3 and 7 d.p.i. +/− NEV or after washout of NEV at 3 d.p.i. and live imaging (EVOS microscope) were done at 7 and 12 d.p.i., respectively. (B) The increased ratio of GFP+ cells between the first 2 time points p.i. and post wash out. (C) Percentage of GFP after recovery of RT post wash out at different time p.i. Cells infected without NEV were considered as 100%.

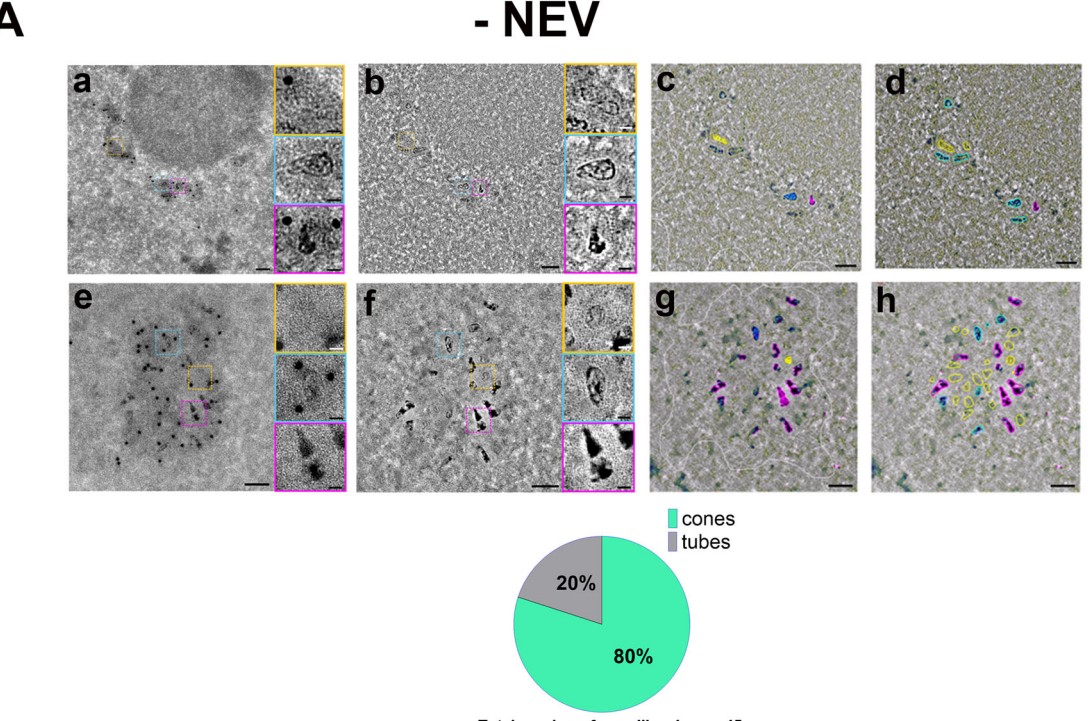

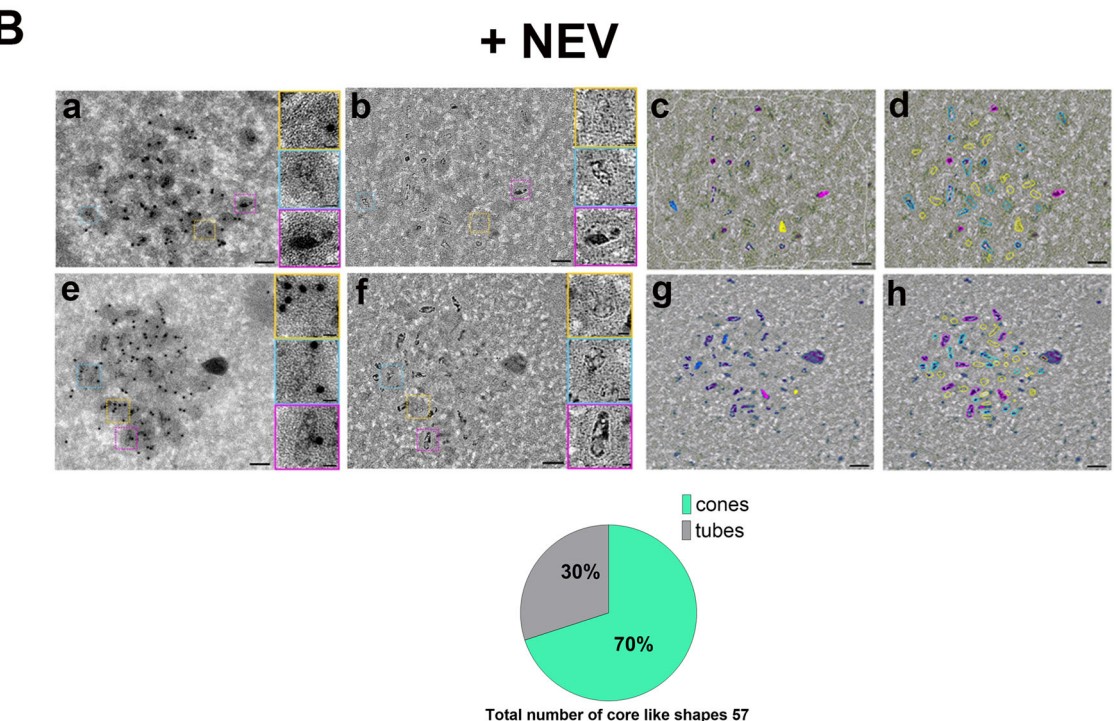

**Figure EV3.   Viral core classification in HIV-1-MLOs, related to Fig. 5.**

(A) Nuclear HIV-1-MLOs composition classes in absence of NEV: in the panels (a, e) projection images at sites of interest as recorded by the detector. Immunogold labeling (10 nm beads) are labeling the CPSF6 at HIV-1-MLOs. In the panels (b, f) a slice from the middle of the dual-axis tomographic volume where HIV cores are more easily observed. Examples of the three different classes as they identified according to the criteria below in dense cores (in magenta boxes), lighter cores (in blue boxes) and ghosts (in yellow boxes). Assigning core classes in absence of NEV: Pixel classification predictions for two different sites shown here (c, d, g, h). In the left column, initial user input (labels) are shown in the images as thin lines together with the predictions for the rest of the image (c, g). The prediction of ghosts was not possible using all features selected in Ilastik, and the pixels were most commonly classified as background (background was seeded as a separate longer label, here in white). In the right column (d, h), pixel classification predictions are overlaid with the manual traced contours of cores, colored according to the underline predictions. Magenta and blue predictions outside the cores are either due to immunogold beads that are still contributing to the slice under evaluation, or sometimes due to Tokuyasu artifacts (darker/thicker area in Bg). Scale bars = 100 nm. Percentage of HIV core shapes (pie chart): HIV cores were included in this analysis only when their longer axis was parallel or near parallel to the tomographic XY plane. The cores were classified as cones in the case an orientation (head to tip) could be assigned easily to them. The characteristic conical shape of the HIV was easily identifiable in most of the cases. In the rest of the cases, where the head could not be identified, HIV cores were assigned as of tubular shape. (B) Nuclear HIV-1-MLOs composition in presence of NEV (panels a, b, e, f), assigning core classes (panels c, d, g, h) and percentage of HIV core shapes (pie chart) have been processed as in (A).

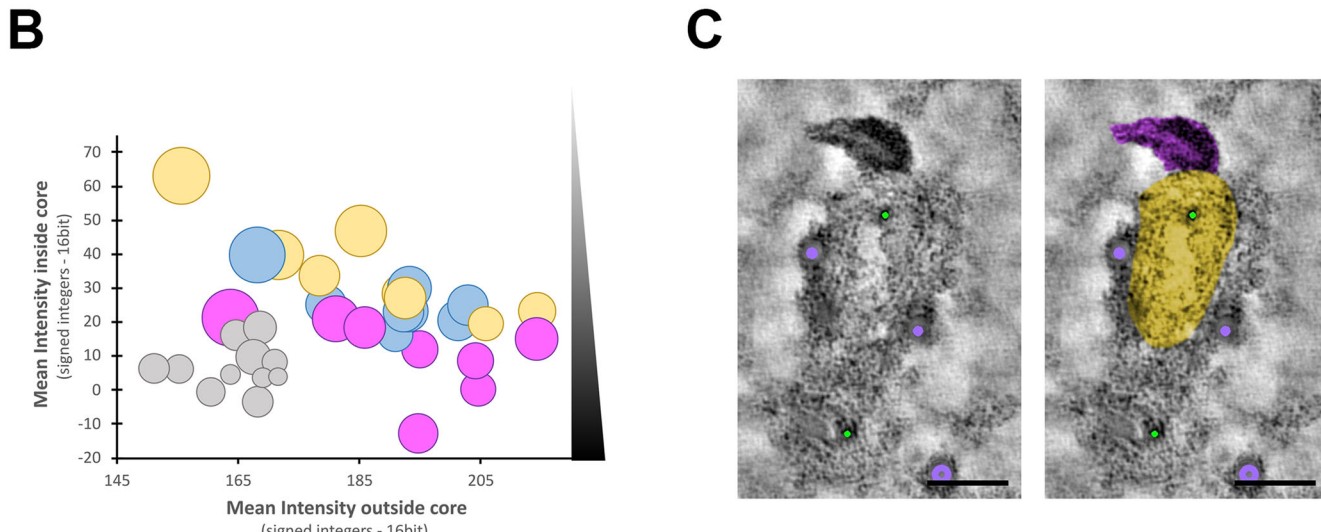

**Figure EV4. Viral core intensity analysis and comparison to free viruses, related to Fig. 5.**

(A) Intensity analysis of HIV cores. A representative core from mature virions before infection (first row), a representative of a dense core inside a nuclear MLO (second row), a representative core with lighter dark areas inside a nuclear MLO (third row) and a representative of a "ghost" or empty core in HIV-1-MLOs (forth row). In middle column only the inside of the core intensities are maintained, while in the third column only the exterior intensities are visible. (B) A scatter plot from the internal mean intensity (y axes) (negative values represent denser signal) vs external mean intensity (x axes) of individual cores. The diameter of the dots represents the standard deviation of the mean interior intensity for each individual core. In gray, the corresponding values from mature virions before the infection, in magenta the values from full cores inside the MLOs, cores in MLOs with intermediate intensities are in blue and the values from ghosts in MLOs are in yellow. (C) A ghost core detected by anti-CA antibody (green bead) is decorated with anti-CPSF6 antibody (purple beads). The left panel shows the original volume section, while the right panel is pseudocolored: yellow indicates the ghost core, and purple highlights the dark intensity near the core's head. Scale bar 50 nm.

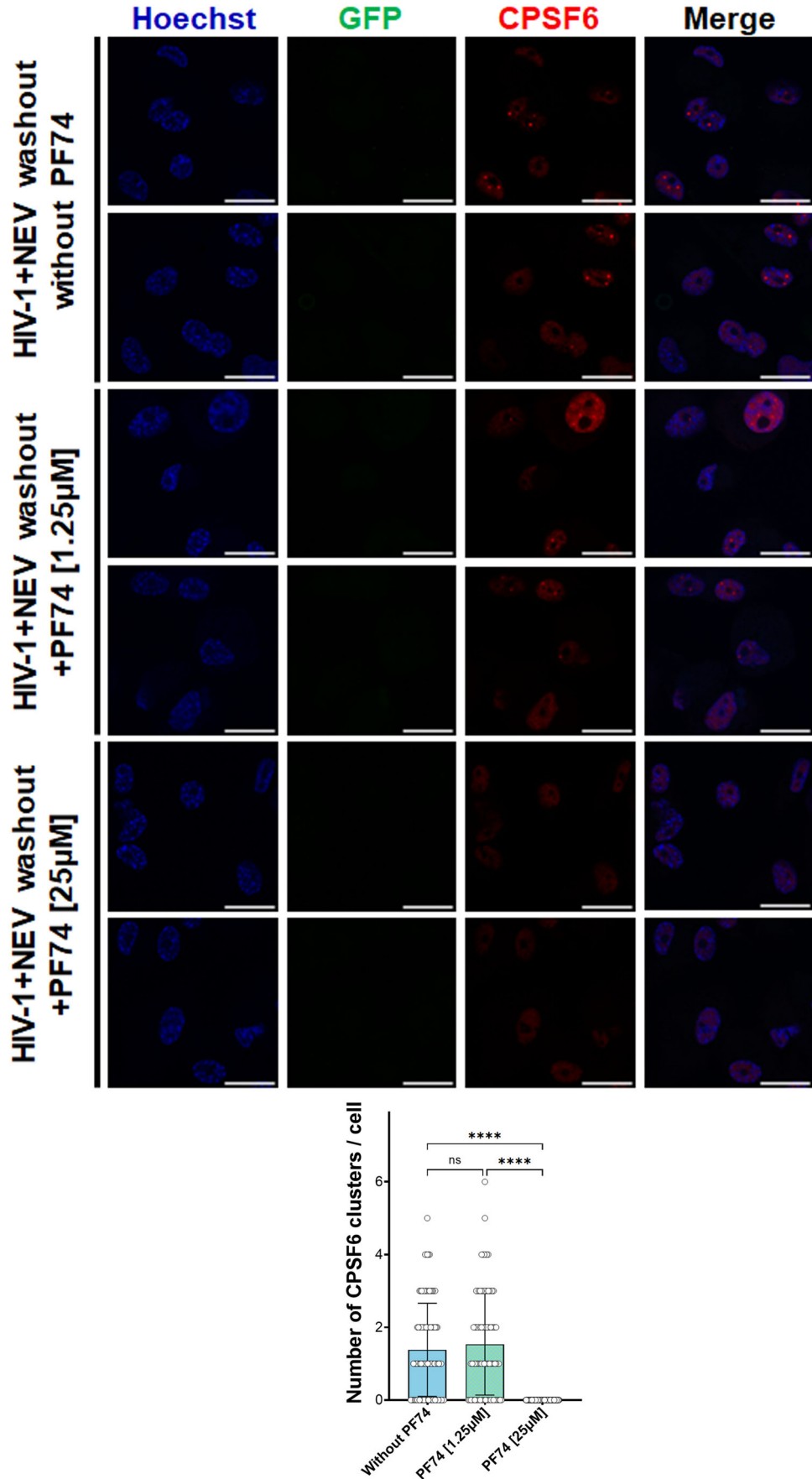

◀ **Figure EV5.  High PF74 dosage leads to the disassembly of CPSF6 condensates, related to Figs. 6, 7 and 8.**

Immunostaining of cells kept in the same culture conditions as in Fig. 6A and an additional condition with lower PF74 dosage (1.25 μM) (blue = Hoechst; green = GFP; red = CPSF6; scale bar = 20 μm) (top panel). Graph showing the quantification of CPSF6 clusters per cell in cells treated with PF74 (1.25 and 25 μM) or not (Data are shown as the mean ± SD of two datasets of biological replicates, statistical test: ordinary one-way ANOVA; ****: *p*-value= $6 \times 10^{-14}$; ns: *p*-value = 0.6040) (bottom panel).

