## [Peer Review File · The EMBO Journal]

***In vivo* HIV-1 nuclear condensates safeguard against cGAS and license reverse transcription**

Selen Ay, Julien Burlaud-Gaillard, Anastasia Gazi, Yevgeniy Tatirovsky, Celine Cuche, Jean-Sébastien Diana, Viviana Scoca, James Di Santo, Philippe Roingard, Fabrizio Mammano and Francesca Di Nunzio

Corresponding author: Francesca Di Nunzio (dinunzio@pasteur.fr)

Review Timeline:

Submission Date:	12th Jun 24
Editorial Decision:	23rd Jul 24
Revision Received:	16th Aug 24
Editorial Decision:	9th Oct 24
Revision Received:	24th Oct 24
Editorial Decision:	5th Nov 24
Revision Received:	5th Nov 24
Accepted:	7th Nov 24

Editor: Ieva Gailite

Transaction Report:

(Note: The initial review process for this manuscript took place with another journal. The initial reviewers' comments and authors' responses for this article have been made available. With the exception of the correction of typographical or spelling errors that could be a source of ambiguity, letters and reports are not edited. Depending on transfer agreements, referee reports obtained elsewhere may or may not be included in this compilation. Referee reports are anonymous unless the Referee chooses to sign their reports.)

Reviewer Expertise:

Referee #1: HIV, Gene expression

Referee #2: Mol. Biology

Reviewers Comments:

Reviewer #1 (Remarks to the Author):

In this study, Ay et al. performed imaging studies using fluorescent microscopy, thin sections EM, EM tomography and FISH to measure co-localisation of CPSF6 membraneless condensates (MLOs) with intranuclear HIV-1 capsid cores, viral RNA (vRNA) and integrase in macrophages or macrophage-like cells. They use the reverse transcriptase inhibitor nevirapine to examine the effect of viral DNA synthesis on the formation of MLOs, observing that blocking reverse transcription results in larger MLOs and in fewer empty-like, or “ghosts”, cores, possibly a sign of uncoating upon DNA synthesis. Notably, the authors also show that MLOs form in vivo in macrophages from humanized mice infected with HIV-1. Using a combination of imaging and qPCR, the authors report that MLOs are sites of HIV-1 reverse transcription and protect the viral DNA in the capsid core from cGAS-mediated sensing.

The paper is interesting and timely. It provides further support for the functional relevance of CPSF6-containing MLOs and, importantly, demonstrate that these exist also in vivo in humanised mouse macrophages. This is a novel and exciting area in the HIV-1 field. The imaging results are generally of high quality, but some conclusions of the paper need further evidence.

We express our gratitude to the referee for his/her interest in our study and appreciate the insightful comments provided. Below, we address each point, aiming to enhance the accessibility of our study for readers. Additionally, we have seized this opportunity to amplify the significance of our findings by including further experiments.

1. Figure 2D and E. Here the authors compare the localisation of the CPSF6 signal in cells treated or not with nevirapine and perform vRNA FISH and MCP-GFP labelling to demonstrate co-localisation of the viral complex with MLOs. The difficulty is that the background signal for MCP-GFP in the +Nev image is high and it is hard to say if there are no MCP-GFP puncta. The authors should also clarify if the MCP-GFP recognises both vRNA and vDNA, or only vRNA, or preferentially vDNA.

The MCP-MS2 system, developed by Edouard Bertrand (IGH, France), is designed for the specific detection of viral RNAs. It utilizes a genetically encoded reporter system based on RNA stem-loop multimerization derived from bacteriophage MS2 (PMID:9809065). MCP-GFP has been designed to exclusively label RNA and not DNA (PMID:9809065) and many studies use this approach to exclusively label RNA (Bertrand et al., 1998; Dufourt et al., 2018; Faló-Sanjuan et al., 2019; Fukaya et al., 2017; Fusco et al., 2003; Garcia et al., 2013; Hoppe et al., 2020; Koromila and Stathopoulos, 2019; Lee et al., 2019; Lionnet et al., 2011; Lucas et al., 2013; Scholes et al., 2019; Yamada et al., 2019; etc.). This approach, leveraging repeat MS2 RNA hairpins, has been widely applied in tagging RNAs for live-cell imaging (PMID:9809065; PMID:18303053; PMID:21613549; PMID:31990126), enabling the real-time, high-resolution detection of individual mRNA molecules. The system's efficacy stems from the high-affinity interaction between the fluorescently labeled MS2 Coat Protein (MCP) and MS2 hairpins integrated into the HIV genome. While we acknowledge the referee's observation regarding the high

expression levels of MCP-GFP protein, we note that MCP-GFP puncta are observed exclusively in conditions without NEV, and these puncta do not colocalize with CPSF6 clusters harboring incoming vRNA genomes. This suggests that the genomes are likely shielded by viral structures, as evidenced by the observed variety of capsid shapes within CPSF6 clusters (Fig.4). Following the reviewers' suggestions, we have provided further clarification in this section. Our data suggest that MCP-GFP is distributed randomly within the nucleus and does not bind to the incoming vRNA located inside CPSF6 clusters. We observed that 100% of MCP-MS2 dots detected did not colocalize with CPSF6 clusters (Fig. 2B-D). Instead, it binds only to the vRNA outside CPSF6 clusters, which likely represents vRNA transcripts or newly formed vRNA genomes poised for generating new progeny. This is evidenced by the green puncta observed outside CPSF6 clusters (New Fig.2B-C, new movies 1 and 2).

2. Line 127, authors state that the presence of incoming viral RNA genomes clustered in HIV-1 MLOs is a pre-requisite for productive infection, which requires integration. However, the authors should also mention the fact that un-integrated DNA can be transcribed, albeit at low efficiency but eventually un-integrated viral DNA will be lost (it is an open question how long this will take in macrophages).

We thank the referee, and we followed his/her advice changing the sentence Line139 "Taken together, our data indicate that the presence of incoming viral RNA genomes clustered in HIV-1-MLOs is established prior to RT and integration as a pre-requisite for canonical post-nuclear RT events".

3. Figure Supplementary 1C, same high background problem mentioned above.

Supplementary Figure 1C shows the presence of CPSF6 clusters in cells recovered from organs and immediately imaged without the need of cell culture passage. The staining shows that CPSF6 is randomly distributed in the nucleus of cells from uninfected humanized mice while in infected cells isolated from bone marrow of infected mice we can detect CPSF6 condensates through a direct tissue analysis. The CPSF6 puncta are clearly visible. We can find these cells carrying CPSF6 clusters only in infected mice.

4. Figure 3C. Please explain why in the upper panel the signal is defined as incoming viral RNA and in the lower panel it is described as vRNA transcription focus.

Our *in-cellulo* data (Figure 2) demonstrates that the viral RNA (vRNA) detected by RNA Fluorescence in situ Hybridization (FISH) within CPSF6 puncta originates from incoming viruses. Indeed, in presence of NEV, a reversible inhibitor of RT, there is only the incoming vRNA that is detected exclusively inside CPSF6 clusters. This observation is supported by previous studies from our lab (Rensen et al., EMBO J. 2021 doi.org/10.15252/emboj.2020105247; Scoca et al., JMCB 2022 DOI: 10.1093/jmcb/mjac060). The vRNA detected outside CPSF6 clusters represents newly transcribed vRNAs, indeed it can be detected only in cells infected in absence of NEV, as we previously published (Scoca et al., JMCB 2022 DOI: 10.1093/jmcb/mjac060).

These *in cellulo* findings enable us to present in Figure 3C the identification of two types of cells within the bone marrow of humanized mice during acute infection. We observed cells likely containing only incoming vRNA within CPSF6 clusters, indicative of early stages of infection. Another possibility is that these cells contain pre-integration reservoirs. Conversely, other cells displayed vRNA puncta outside

CPSF6 condensates and, in some cases, vRNA in the cytoplasm, suggesting efficient infection and/or progression to a late stage of infection (refer to Fig 3C and Suppl. Fig. 1D; new movies 3 and 4).

5. Figure 4B. There is hardly any 6nm gold labelling decorating the viral capsid cores. Gold particles of different sizes on the core-like structures should be counted and the data presented.

We concur with the reviewer's observation that most of the gold particles detected are attributable to CPSF6 rather than capsid detection. To scrutinize the state of the viral capsid within CPSF6 clusters, we employed CPSF6 immuno-gold labeling. It is well-documented that viral capsid epitopes are obscured from CPSF6 and are scarcely recognized by antibodies against the capsid in immunofluorescence assays (DOI: 10.7554/eLife.64776). Transmission electron microscopy (TEM) sections are not permeabilized, which limits epitope accessibility to only a few on one side of the slices. Therefore, it is unsurprising that the viral capsid is minimally decorated with gold particles.

The primary objective of this experiment was to investigate the presence of capsid structures within CPSF6 clusters. This is evidenced in our findings presented in Figure 4 (panels B, C, D, and F), Supplementary Figure 2E, and in movies 5 and 6.

6. Figure 5B. PF74 at 25 uM can block reverse transcription by acting directly on the capsid core itself outside MLOs and so it is difficult to attribute the absence of early and late HIV-1 reverse transcripts only to the disruption of MLOs. The authors should use a lower concentration of PF74 which does not directly inhibit RT. This could be tested by adding PF74 alone at different concentrations followed by washout.

We fully understand the concerns raised by the reviewer.

In supplementary Figure 4D we compared the effects of low and high doses of PF74 on CPSF6 clusters in HIV-infected THP-1 cells. Our observations indicate that a low dose of PF74 does not influence the number of MLOs. This outcome suggests that low doses cannot be utilized to explore the role of MLOs in the post-nuclear entry steps of the virus. It should be noted, however, that although high doses of PF74 lead to the disassembly of CPSF6 clusters, they conversely stabilize viral cores *in vitro*, as documented in the following studies: *Molecules* (2020, 25(8), 1895, DOI: 10.3390/molecules25081895), *Journal of Virology* (2017, 91(23), e01732-17, DOI: 10.1128/JVI.01732-17), and *Proceedings of the National Academy of Sciences* (2015, 112(5), 1504-1509, DOI: 10.1073/pnas.1419945112). Thus, a destabilizing effect on the capsid would not be expected based on the literature.

However, to address to the reviewer's concern, we conducted further experiments. We infected THP-1 cells for 48 hours, which coincides with the timing of PF74 addition as shown in Figure 5 and at this time of infection all viruses are mainly accumulated in HIV-1-MLOs. In this set of experiments, we treated some samples with PF74 at a concentration of 25 μ M for 2 hours. Subsequently, we performed immunofluorescence (IF) assays. Our observations revealed that CPSF6 clusters were present exclusively in the infected cells that had not been treated with PF74. Conversely, in cells treated with PF74, CPSF6 clusters were completely disassembled. However, we observed numerous capsid puncta (the Ab against capsid can only detect it if the epitope is not occupied by CPSF6 as published in doi: 10.7554/eLife.64776) (New Figure 5C), indicating that the capsids were not disassembled by the high dose of PF74, in line with previous *in vitro* data indicated above.

In agreement with these results, HIV-1 cores were solely observed within the confines of HIV-1 MLOs, as substantiated by transmission electron microscopy (TEM) detailed in Supplementary Figures 2B-C-E; Figure 4A. This exclusive localization within MLOs argues against any attribution of nuclear reverse transcription (RT) products to cores situated externally. Consistent with these findings, our previous work (Rensen et al., 2021 doi.org/10.15252/embj.2020105247) demonstrated that newly synthesized viral DNA (vDNA) is exclusively observed within CPSF6 clusters, as illustrated in the image below.

Primary macrophages infected with HIV-1 single round infection. Removing Nevirapine restores the reverse transcription in the nucleus (vRNA, RNA-FISH in green; vDNA, EDU in magenta; merge vRNA/vDNA)

In the submitted study, we demonstrate that disassembly of CPSF6 clusters prevents the restart of reverse transcription.

However, our transmission electron microscopy (TEM)/tomography data do not enable the exploration of nanoscale structural information. Indeed, the precise effects of PF74 at the atomic level on the structure of viral cores within these clusters are yet to be fully elucidated, particularly concerning potential structural modifications, we have highlighted this promising area for further research in the limitations section of our revised manuscript.

7. The idea that the viral DNA in the MLOs is protected from cGAS and sensing is very interesting but needs some clarification and additional evidence. In normal conditions, cGAS must be inhibited in the nucleus, and so how do the authors think it is activated by the viral DNA? Although higher cGAMP levels suggest cGAS activation, to demonstrate physiological activity it is important to measure IFN production, or CXCL10, or other chemokines/ISGs induced by cGAS activation.

We sincerely appreciate the reviewer's constructive comments, which have significantly enhanced the clarity and impact of our study's message. In response to the suggestion for demonstrating the disruption of nuclear HIV-1-MLOs and its effect on exposing dsDNA to activate nuclear cGAS, we have conducted additional experiments using THP-1 cells but we importantly demonstrated this phenomenon also in primary macrophages from healthy donors.

First, to monitor cGAS activation, we quantified cGAMP levels at various time points following PF74 treatment. THP-1 cells, known to be suboptimal for producing new viral particles, form few pre-integration complexes within CPSF6 nuclear clusters, as reported in our previous publication (Scoca et al., JMBC 2023). Our data demonstrate that viral cores typically persist within CPSF6 puncta for an extended period of time (see Fig. 4F-G). Notably, the viral genome, reverse-transcribed within these cores (Suppl. Fig.3A-D), is eventually released from these nuclear sites. Indeed, the formation of dsDNA in the nucleus—a pre-requisite for cGAS activation—has been well-documented by multiple research groups, including our own work (Scoca et al., JMBC, 2023) and others (Selyutina et al., Cell Reports 2020; Dharan et al., Nature Microbiology 2020; Muller et al., eLife 2022). We observed that after 2

hours of PF74 treatment, the viral capsid remains as visible puncta, albeit CPSF6 clusters are fully disassembled (Fig.5C, see above in the answer to the point 6). It is during this disassembly that we first detect an increase of cGAMP, likely due to the exposure of the newly formed dsDNA. Further analysis of cGAMP induction was conducted at 10h post- PF74 treatment, with results detailed in the new Figures 6C.

Second, to evaluate the role of CPSF6 cluster to shield the dsDNA from cGAS , we generated THP-1 cells knockout (KO) of CPSF6 employing the CRISPR-Cas9 system. We then infected both KO and control cells to assess the presence of cGAMP, a critical marker for the activation of nuclear cGAS by dsDNA. Notably, in the CPSF6 KO cells—which lack the ability to form CPSF6 clusters that according to our data shield the viral genome from DNA sensing mechanisms (Fig.6A-C) we noted a marked enhancement in cGAMP induction in KO cells when compared cGAMP production from infected control cells, as illustrated in the new Figure 6D.

These findings substantiate our hypothesis that the disruption of nuclear HIV-1-MLOs facilitates the exposure of dsDNA, thereby activating nuclear cGAS. Notably, research from Wes Sundquist's lab (University of Utah) has demonstrated that dsDNA is released from viral cores as soon as it forms (Christensen et al., 2020, Science). We think that this additional data strengthens our study's message and underscores the critical role of CPSF6 in modulating DNA sensing pathways during HIV infection.

Third, following the reviewer's suggestion, we show in the new manuscript version the induction of interferon-stimulated genes (ISGs), such as CXCL10, MxA, and IFIT2, triggered by cGAS activation through RT-PCR. Specifically, in THP-1 cells infected for 48 hours, a significant increase in these ISGs was observed after the addition of PF74 (New Figure 6E).

Figure 6E

THP-1 and ISGs

Significantly, we successfully confirmed these findings in primary monocyte-derived macrophages (MDMs) from healthy donors. This underscores the physiological importance of CPSF6 condensates in protecting against DNA sensing, as demonstrated in the new Figure 6F and new suppl. Fig.6.

Figure 6F

MDMs and ISGs

Reviewer expertise: molecular biology

This manuscript is a follow-up study of the previous work by the Di Nunzio lab, which proposed nuclear CPSF6-containing HIV-1 membraneless organelles (HIV-1 MLOs) as hubs for HIV-1 reverse transcription in macrophage-like cells (Scoca et al., 2022, *J. Mol. Cell Biol.*). Here, the authors confirm the occurrence of these HIV-1 MLOs by immunofluorescence microscopy *in vivo* in isolated BM-derived cells from HIV-1-infected humanized mice. The authors demonstrate that VSV-G pseudotyped replication-incompetent HIV-1 viruses induce HIV-1 MLOs that can persist over at least 25 days during pharmacological reverse transcriptase inhibition *in vitro* in macrophage-like cells. These MLOs might constitute a relevant dormant virus reservoir complicating HIV-1 clearance in the host. Interestingly, the authors could visualize different classes of viral core particles within the MLOs inside the nucleus by TEM, indicating the coexistence of different stages during the early viral replication cycle within these MLOs. The condensates might shield viral DNA from the innate immune sensor cGAS recognition.

Overall, the manuscript provides an interesting perspective on the early steps of the HIV replication cycle and is presented clearly. The correlations between HIV-1 MLOs and infection efficiency and immune evasion are shown. Nevertheless, it remains challenging to demonstrate causality. I have provided some comments for revision below.

The term "cocoon"

I strongly recommend removing the term "cocoon" from headings, the result part, and data figures for the following reasons: Firstly, "cocoon" already has a precise meaning in biology and is misleading here as it is used metaphorically. The analogy does not truly match as a "cocoon" suggests a shell architecture, for which no evidence is provided here. Previously, the authors described the MLOs as liquid-liquid phase-separated compartments, which would somewhat contradict the "cocoon"

analogy. For clarity and to avoid scientific term confusion, the authors should stick to the established terminology of Condensate.

We thank the reviewer for his/her constructive comment. We fully agree and have replaced the term 'cocoon' with 'condensate' throughout the manuscript, including the title.

Correlating MLOs and replication using the capsid-binding PF74

The authors demonstrate the correlation between HIV-1 MLOs and effective infection / immune evasion. However, it remains challenging to establish causality. Ay et al. use a pharmacological approach, i.e., the compound PF74, "known for its ability to dissolve CPSF6 condensates, at a high dose" (l. 246). It should be noted, however, that this drug appears to have pleiotropic effects and predominantly directly interacts with the viral capsid proteins, which should be more clearly stated and referenced in the manuscript. This disassembly/CA binding likely prevents interactions required for further virus propagation as the capsid protein also participates in reverse transcription. Hence, reported effects are not necessarily due to the disruption of the mesoscale condensates, which should be explicitly stated as a limitation.

We thank the reviewer for his/her comments. We include in the revised text more info about the mechanism of action of PF74. As indicated in the reply to question 6 of the 1st reviewer, although high doses of PF74 lead to the disassembly of CPSF6 clusters, they conversely stabilize viral cores *in vitro*, as documented in the following studies: DOI: 10.3390/molecules25081895), DOI: 10.1128/JVI.01732-17), and DOI: 10.1073/pnas.1419945112).

Of note, the mechanism of action of high dose of PF74 on viruses trapped in HIV-1-MLOs is not clear. In our study we added high dose of PF74 after 48 h of infection (a time point where almost all viruses are already in the nucleus in CPSF6 puncta). To answer to the reviewer's concern, we conducted further experiments. We infected THP-1 cells for 48 hours, which coincides with the timing of PF74 addition as shown in Figure 5. In this set of experiments, we treated some samples with PF74 at a concentration of 25µM for 2 hours. Subsequently, we performed immunofluorescence (IF) assays. Our observations revealed that CPSF6 clusters were present exclusively in the infected cells that had not been treated with PF74. Conversely, in cells treated with PF74, CPSF6 clusters were completely disassembled. However, we observed numerous capsid puncta (the Ab against capsid can only detect it if the epitope is not occupied by CPSF6 as published in doi: 10.7554/eLife.64776) (new Figure 5C), indicating that the capsids were not disassembled by the high dose of PF74. Interestingly, we did not observe any significant difference between the amount of CPSF6 puncta before PF74 treatment and CA puncta after PF74 treatment. This finding is in line with previous *in vitro* data (see: DOI: 10.3390/molecules25081895; DOI: 10.1128/JVI.01732-17; DOI: 10.1073/pnas.1419945112).

Figure 5C

Significantly, disassembling CPSF6 clusters with a high dose of PF74 leads to an induction of cGAMP, attributed to the exposure of dsDNA. We hypothesize that dsDNA, once formed, exits from viral cores, which then appear empty, as supported by our TEM/tomography data (Fig.4F-G). These observations align with *in vitro* findings from Wes Sundquist's lab (doi: 10.1126/science.abc8420). Furthermore, dsDNA formed within CPSF6 clusters, as previously shown by Scoca et al., 2023, and Muller et al., 2022, seems to be responsible of the activation of cGAS—a mechanism now illustrated in our new Figure 6C.

Figure 6C

While the precise effects of PF74 at the atomic level, particularly concerning potential structural modifications within these clusters, remain partially unresolved due to the atomic-scale resolution limitations of our current imaging techniques.

Acknowledging the referee's insight, we concur that our study introduces new perspectives while also facing certain limitations. This caveat is addressed in the 'Limitations of the Study' section of our revised manuscript. Overall, it is important to specify that the PF-74 drug, which is the precursor of Lenacapavir approved in clinics, has an effect on the activation of innate immunity once the HIV-1-

MLOs in the nucleus are disassembled. This represents a new perspective and an unexplored mechanism of action for these compounds.

The origin of the 3 virus core classes

The demonstration of the three virus core classes is indeed interesting – the authors mention that it is unclear whether those constitute steps in the viral life cycle or have been present in the original virus prep. It would help to add references here, as other labs have seen cone-shaped virus cores and their rupture (rather than complete cooperative disassembly) in the nucleus before (e.g., Li et al., PNAS, 2021;118(10), doi: 10.1073/pnas.201946711; Müller et al., Elife, 2021, doi: 10.7554/eLife.64776, and Zila et al., Cell.2021;184(4):1032-1046.e18. doi:10.1016/j.cell.2021.01.025). Fixation artefacts should be considered as well. Providing TEM images of the virus preps and classifying virus cores might be informative.

We have listed in the revised version of our manuscript other studies that also found viral cores in the host nucleus (line 180-181). We have also described in material and methods that immune-gold TEM experiments have been conducted on fixed samples, because cells were infected with HIV. We have also mentioned in the new revised version that we exclude the fact that empty cores were already present in the virus prep as it has been elegantly published from John Briggs (Mattei et al., Science 2016) see figure below extracted from his study.

In addition to our initial analyses, we utilized TEM and tomography to classify intact viruses from viral preparations based on their core-density profiles, as compared to viral cores observed in host nuclei. These new data, included as new supplementary Figures 2F-G, show that capsids in the preparation share similar electrodense characteristics with dense cores observed in HIV-1-MLOs.

Supplementary Figure 2F-G

Suppl. Figure 2

Intensity analysis of HIV cores. Top panel. A representative core from mature virions before infection (first row), a representative of a dark core inside a nuclear MLO (second row), a representative core with lighter dark areas inside a nuclear MLO (third row) and a representative of a “ghost” or empty core in HIV-1-MLOs (forth row). In middle column only the inside of the core intensities are maintained, while in the third column only the exterior intensities are visible. **Bottom panel.** A scatter plot from the internal mean intensity (y axes) (negative values represent denser signal) vs external mean intensity (x axes) of individual cores. The diameter of the dots represents the standard deviation of the mean interior intensity for each individual core. In grey, the corresponding values from mature virions before the infection, in magenta the values from full cores inside the MLOs, cores in MLOs with intermediate intensities are in blue and the values from ghosts in MLOs are in yellow.

Together, these results suggest that empty cores are not present in the initial viral preparations, indicating that they likely form during their journey through the cytoplasm or nucleus. Interestingly, a recent paper published on bioRxiv, which utilizes coarse-grained molecular dynamics simulations, supports our hypothesis. It suggests that some viral cores might lose their content during nuclear translocation (doi.org/10.1101/2024.04.23.590733).

Cytoplasmic reverse transcription sites
I understand that the authors focused on nuclear events here. Most figure panels show only single magnified nuclei. Nevertheless, the authors should comment on the potential contribution of cytoplasmic events (e.g., relating to integrase, vRNA, and vDNA signals). E.g., in Fig. 3, there is a clear cytoplasmic vRNA signal in mouse bone marrow cells. Ideally, Fig. 2A would show the entire cell.

Figure 3 shows monocyte-derived macrophages (MDMs) from the bone marrow of mice infected with a replication-competent virus. We interpret the predominant RNA signal in the cytoplasm of some cells as likely representing viral transcripts, which are abundant in the cytoplasm. This does not, however, rule out the detection of incoming viral RNAs (vRNAs). Notably, the presence of vRNAs in the cytoplasm and a focus of viral transcription near a CPSF6 cluster in the nucleus in these cells suggests active transcription from a provirus within the cell.

We included two additional supplementary figures that show multiple entire cells regarding experiments cited in Fig.2A and Fig.2D (New Suppl. Fig. 3E-F).

Establishing the necessity of CPSF6 clusters for productive infection
The authors correlate the productive infectivity and the occurrence of CPSF6 clusters (l. 104 / Fig. 2B). Do the authors observe CPSF6 clusters in GFP-negative cells, and if so, how many? The authors should report these numbers as well and comment on them. A comparison to NEV-treated THP-1 cells infected by the GFP reporter HIV-1 would be helpful here. This is particularly relevant, as the authors mention that their "data indicate that the presence of incoming viral RNA genomes clustered in HIV-1-MLOs is a prerequisite for productive infection" (l. 127), but still observe 16 % of GFP-positive cells which do not display CPSF6 clusters 4 d.p.i., which seems to differ from the pseudotyped HIV-1 virus infection (Fig. 11B, 2E). The authors should also state how the MOI for all HIV-1 virus preps was determined.

Our research confirms that CPSF6 clusters are consistently present in both ~83% of GFP-positive cells and in 70.13% of GFP-negative cells four days post-infection (New Figure 2F). This indicates that a significant number of cells may host early-stage viral components or be affected by restriction factors like SAMHD1, which inhibit complete infection. Notably, the majority of transcriptionally active (GFP+) cells contain these clusters, corroborating their role in viral replication as observed in prior studies (Rensen et al., 2021; Francis et al., 2020).

We also noted a disassembly of CPSF6 clusters post-reverse transcription, aligning with findings by Li et al., 2021 PNAS (doi.org/10.1073/pnas.2019467118). This phenomenon could explain the 16% of GFP+ cells lacking CPSF6 clusters. Our data reveal that CPSF6 clusters decrease from 2.2 to an average of 0.87 by day four post-infection in the absence of NEV, as shown in Figure 5B. This suggests that CPSF6 cluster numbers reduce as the viral lifecycle advances.

Experimentally, we employed VSV-G pseudotyped viruses in cell-based tests, adjusting the multiplicity of infection (MOI) according to viral efficiency—10 for standard conditions and increased to 30 for less efficient MS2 loop-bearing viruses. These adjustments explain the variations in CPSF6 cluster counts observed between Figure 1B and new Figure 2B. Full-length viruses were used for *in vivo* studies. We've revised our virus cartoons in Figure 2A for better clarity and detailed MOI specifics in the figure legends.

MOIs were calculated by titrating the viruses using RT PCR, as detailed in the materials and methods section.

We have incorporated these points into the manuscript to address the comments raised by the referee, enhancing the clarity and depth of our findings.

Incoming vRNA versus total vRNA and MLO permeability

The authors should be careful when using statements about the vRNA origin, like "incoming vRNA" (e.g. in l. 105 or in l. 148 / Fig. 3: "Interestingly, we observed a positive correlation between the percentage of cells presenting CPSF6 clusters and GFP-positive cells (...) and between the incoming vRNA genome located in CPSF6. ") – How can the authors truly discriminate between the incoming vRNA and de novo-produced vRNA at this point when the cells are not treated with a reverse transcriptase inhibitor? I agree that these clusters seem to upconcentrate incoming vRNA, but it does not exclude the possibility of upconcentrating de novo-produced vRNA. To err on the side of caution, I recommend using "vRNA" if the origin of the RNA cannot be demonstrated.

I am not convinced that the authors show here that the MCP-GFP reporter could not enter the mesoscale HIV condensates per se, which would indicate a permeability barrier. In that case, we would expect a decrease in signal intensity in Fig 2D, which is not seen here (intensity profile). Along these lines, the scheme in Fig. 2D would be obsolete. The line along which the intensity profile was plotted must be indicated in the inset (Fig. 2D). L. 113 "These results suggest that the incoming vRNA is protected by CPSF6 condensates, which may act as a filter to select only a few molecules to enter inside the clusters." – an alternative explanation is that protein-vRNA interactions inside the condensate mask the vRNA sequence / sterically hinder the large MCP complex probe binding, which would agree with the TEM images of near-intact viral cores.

Figure presentation

Individual microscopy image panels should be clearly separated from each other (Fig. 1 A, 2D, 3D). The authors should avoid adding a common black background for several microscopy panels (Suppl. Fig 1C, D, 1E). Furthermore, coloured text on merged-channel images should be avoided as this is confusing (e.g., Fig. 3C, Suppl. Fig 1). Single-channel images should be provided for critical panels (Suppl Fig. 1E, Fig 6 A).

The authors should consider labelling the capsid protein labelled by immunogold with "CA" and the core capsid identified by its electron density architecture with another name (e.g., "viral core" in Fig. 4) for consistency.

We thank the reviewer for the constructive comments, which have greatly assisted in refining our manuscript. In alignment with these observations, we concur that MCP-GFP diffuses broadly within the nucleus but does not accumulate within CPSF6 clusters. Indeed, we have used two approaches to detect vRNA: RNA-FISH and MCP GFP-MS2. In cells treated with NEV, only incoming vRNA is present and it is detected by the RNA-FISH approach, but not by MCP-GFP. This discrepancy stems from the RNA-FISH technique's ability to detect vRNA, facilitated by sample permeabilization step. MCP-GFP lacks this capability due to the protective barrier of viral cores. Indeed, our observations indicate that

MCP-GFP does not colocalize with CPSF6 clusters, suggesting it cannot bind to the vRNA shielded by viral capsids, a conclusion supported by our TEM/tomography findings (Fig.4F-G). This vRNA is thus a *bona fide* incoming genome.

This implies that newly transcribed vRNAs do not accumulate freely within CPSF6 clusters; otherwise, they would be detected by MCP-GFP, as observed by MCP-GFP puncta outside of these clusters. This finding highlights a potential viral strategy to compartmentalize and safeguard its genetic material from host cellular mechanisms. Following the referee's recommendation, we have revised Figure 2D now Figure 2B and we eliminated the illustrative cartoon. Additionally, the manuscript text has been updated to accurately reflect these changes, emphasizing our agreement with the reviewer insight regarding MCP-GFP's diffusion and its inability to detect protected vRNA within CPSF6 clusters. These updates have been carefully incorporated into the revised manuscript. In addition, we have made adjustments to the other figures based on the feedback. However, we chose to retain the merged channels in Figure 3C. We believe that maintaining these channels allows readers to more easily understand the results, presenting a clearer and more direct visual representation of our findings. We have also included 3D movies (movies 3 and 4).

Figure 6A: This figure includes insets showing individual channels for CPSF6 (colored red) and viral DNA labeled with EdU (colored white), alongside a merged image. The merge visually represents the spatial relationship between CPSF6 and the newly synthesized viral DNA within the cell.

Supplementary Figure 1E: this supplementary figure displays individual channels for CPSF6 (colored green) and viral RNA (colored red), in addition to a composite image. The individual and merged images together illustrate the interaction or co-localization of CPSF6 with viral RNA, shedding light on cellular processes during viral infection. Following the referee's recommendation, we have revised Figure 4D to more accurately depict the core-like shape in place of what was previously labeled as CA.

Statistics/data

visualization

Information about the error bars should be included throughout the manuscript. The error band in Fig. 2B is not explained. Clarify whether r (Pearson correlation coefficient) (-1,1) or R (coefficient of determination) (0-1) or R^2 was calculated (Fig 2). Adding the data for MOCK controls to the plot in Fig. 1B would be beneficial. Single data points should be added to all plots (e.g., Fig 3B, 5B, Suppl Fig 3B-C, Fig. 6B).

We specified that an R^2 has been calculated for the new figure 2F in the text and in the figure legend. The error bars represent the 95% confidence interval of the simple linear regression curve calculated using Prism. We added data from mock (non-infected cells) in both graphs in the figures 1B-C. About the graph 3B on the left it represents a qPCR for each mouse. Each sample has been run in duplicate. The graph in the middle is a percentage of cells presenting CPSF6 clusters calculated on more than 150 cells per each mouse. We added single data points to the plots in Figure 5B as well as to figure 6B where we also increased the number of biological replicates. Supplementary figure 3B shows a ratio and Figure 3C a percentage.

Several plots need a description of the sample size within the plot or figure legend.

We added the missing information in the legends.

Fig. 5: The heatmaps are not easily accessible and do not offer advantages over a scatter plot for this

data type. The authors should plot the data as a scatter plot or a histogram, which would be easier to interpret in this case.

There is not heatmap in figure 5, the reviewer probably refers to Figure 4E that we have modified as suggested.

Fig 6 B: what was the sample size and what was the type of replicate? Single data points should be plotted. The x-axis title needs to be clarified. I suppose it is a "fold induction" and not pmol/ng protein? Since cGAMP has been quantitatively assessed, the original values should be reported for the infected and uninfected cells.

The figure was reporting data from two biological replicates and now in the new Figure 6 B we have increased the number of biological replicates to 3. The graph shows the fold increase cGAMP with respect to uninfected cells (cGAMP values are pmol/ng of protein). We report single data points. Original values are reported in the table 1 related to Fig.6B.

Clarity and context
For consideration to be published in Nature Microbiology, I strongly suggest to clarify some approaches to make them more accessible to a broader audience: The authors should clearly state that they used a chimeric replication-incompetent virus at the beginning of the results section to avoid the impression that wild-type HIV-1 was used (especially in the first results paragraph and in l. 164). The VSV-G pseudotyping has implications for the virus infectivity, uptake/entry route, and potentially nuclear translocation and virus fate and should be mentioned with appropriate references. For accessibility to a broader audience, please mention how the viruses are rendered non-replicative in the method section (with reference). This is important in the context of this study as this has implications on the incoming versus the potentially de novo synthesized viral components.

We added a sentence on the beginning of the paragraph (line 183) and in material and methods we included details and references. Cartoons of viruses used have been included in several figures.

Please explain abbreviations when they are mentioned first (like NEV (l. 80)). Please mention the cell line THP-1 name directly in the main text (l. 75). Mention the macrophage / CCR5 tropism of HIV-1 NLAD8 on its first mention and provide a reference for it (l. 134). This virus needs to be included in the materials and methods section. l. 157: "Since condensates are spherical, we evaluated the sphericity of these CPSF6 clusters." Condensates are not necessarily spherical; sphericity is a measure of material properties. Please rephrase.

We followed the advice of the reviewer and modified the sentence (line76, 82). HIV-1 NLAD8 is described in material and methods with reference. We explain why we performed the sphericity analysis in line 171-177.

"At the same time, the persistence of ghosts can contribute to generate a favorable microenvironment for viral DNA synthesis, by maintaining viral nuclear cocoons" – the authors should lower their tone here as this is speculative at this point. "Ghost" virus cores could be remnants without further function

or serve as a reservoir for some viral compounds without a protective function. Causality is not clear to me here.

We added more explanations (line 239-247) and modified the sentence as suggested by the referee (line 258-260).

L. 277 "cGAS is mainly localized in the nucleus" – this statement (and its consequences) is still debated in the field and could be rephrased to "cGAS also localizes to the nucleus".

We changed the sentence in the updated manuscript version as suggested by the referee (line 351). We also performed additional experiments to corroborate the role of CPSF6 condensates as shield against innate immunity as we mentioned in response to the first referee (see point 7 first referee).

Methods section

Many conclusions are based on microscopy images describing clusters and colocalization. The authors should precisely describe the image analysis procedures, how clusters and colocalization were determined / how thresholds were set, and how background and noise were handled.

Image acquisition and analysis procedure have been added to the material and methods

The list of viruses and references needs to be completed in the reagents table. Some full references need to be included (e.g., Petit et al., 1999, 2000).

All PMCID numbers were added to the citations in the reagents table in the Methods session.

The software version numbers need to be included.

The software version numbers have been included in the Methods.

Details on transfection strategies should be added.

Details on transfection strategies have been included in the Methods.

The rotor should be mentioned to determine the g value: "and lentiviral particles were concentrated by ultracentrifugation (22000 rpm) at 4°C for 1 hour".

The Rotor Beckman Coulter, Inc. model SW32Ti has been added to the table in the methods section.

More details on the fixation procedures (wash steps, fixation temperature) should be added. How much DMSO was added to the cells for NEV? Was DMSO added to NEV- cells as MOCK treatment control?

NEV has been solubilized in DMSO at a concentration of 10mM and we use a final concentration of 10µM, so the drug is diluted 1000 times and we use the same DMSO amount for the mock.

The cGAMP ELISA assay procedures should include potential washing steps, the precise volume of lysis buffer used per well and centrifugation steps (if applicable).

cGAMP assay has been detailed in the methods section.

How was the MOI determined?

Viruses have been tited by qPCR to obtain a title as number of transducing units (TU). The MOI has been calculated as number of TU/number of cells.

-

Dear Francesca,

Thank you for submitting your manuscript together with the reviews from another journal and your point-by-point response to them to The EMBO Journal. I have now received input from one of the original reviewers (reviewer #1) and two arbitrating advisors on the revised manuscript. I have copied their comments below.

As you can see, all reviewers are generally positive in their assessment of the revised version, while mentioning several remaining aspects that would need to be addressed in the final revision before I can accept the manuscript for publication here. Specifically, advisor #1 finds that the statements on the causal link between reverse transcription and condensate formation should be appropriately toned down, as well as raises concerns regarding potential effects of P74 on infectivity. Reviewer #2, who assessed the structural and cryo-electron microscopy aspect of study, which was not evaluated during the previous assessment round, indicates that further information into how the condensate size and CPSF6 labelling were measured is needed, and asks for additional quantification. Similarly, he/she finds that better description of viral core classification and its re-assessment with a secondary parameter would be needed. Finally, reviewer #3 evaluated your responses to the original reviewer #2, which he/she finds satisfactory, while also adding a suggestion to expand on the involvement of downstream innate immunity signalling pathway activation, which he/she finds to be feasible within a minor revision. I agree that this would be valuable and would like to discuss with you to what extent this would be possible from your side.

I would therefore invite you to address these remaining comments in a revised manuscript. I think that it would be useful to discuss the revision in more detail via email or phone/videoconferencing - please let me know which option you prefer.

We generally allow three months as standard revision time. As a matter of policy, competing manuscripts published during this period will not negatively impact on our assessment of the conceptual advance presented by your study. However, please contact me as soon as possible upon publication of any related work to discuss the appropriate course of action. Should you foresee a problem in meeting this deadline, please let us know in advance to discuss an extension.

When preparing your letter of response to the referees' comments, please bear in mind that this will form part of the Review Process File and will therefore be available online to the community. For more details on our Transparent Editorial Process, please visit our website: <https://www.embopress.org/page/journal/14602075/authorguide#transparentprocess>. Please also see the attached instructions for further guidelines on preparation of the revised manuscript.

Please feel free to contact me if have any further questions regarding the revision. Thank you for the opportunity to consider your work for publication, and I look forward to discussing your revision with you.

With best regards,

Ieva

At EMBO Press we ask authors to provide source data for the main manuscript figures. Our source data coordinator will contact you to discuss which figure panels we would need source data for and will also provide you with helpful tips on how to upload

and organize the files.

We realize that it is difficult to revise to a specific deadline. In the interest of protecting the conceptual advance provided by the work, we recommend a revision within 3 months (21st Oct 2024). Please discuss the revision progress ahead of this time with the editor if you require more time to complete the revisions.

Referee #1:

The revised paper by Selen Ay et al. is significantly improved. The additional evidence provided in Figure 5 shows the loss of CPSF6/viral condensates upon treatment with the drug PF74. This nicely links an almost complete loss of infectivity to the disappearance of viral condensates.

The revised Figure 6 convincingly demonstrates that loss of viral condensates activates cGAS signalling and induces an innate immune response.

When one considers all the data together, the evidence provided does indeed support the hypothesis that viral condensates promote reverse transcription and shield viral DNA from cGAS sensing before integration.

This concept is novel and interesting and will add a new perspective to the field.

Concerning some remaining limitations, it should be noted that the link between reverse transcription and the viral condensates is correlative, and Figure 6 begs the question if treatment with PF74 at 48h post-infection, which causes the disassembly of the viral condensates, also reduces infectivity. Nonetheless, at this stage I would not delay this paper any further, instead the authors may want to mention this issue at the end of the Discussion.

Referee #2:

The manuscript is a valuable contribution and the authors have addressed most of the reviewer comments in a satisfactory manner.

The electron microscopy data showing the presence of nuclear cores surrounded by CPSF6 is generally convincing and the low CA labelling is indeed typical of anti-CA antibodies bound to cores. My comments only relate to this part of the study.

Are the increased in IN labelling and the decrease in condensate size in the absence of NEV consistent between preparations/experiments? It is not clear whether the data are presented from a single experiment or from more than one. What is the variability in labelling efficiency or on condensate size between experiments? This also goes for the change in CPSF6 labelling. This information is necessary to assess whether the suggested changes in condensates through the lifecycle are real

or not.

How were cores classified into dense, light and ghosts? Was this done by eye? The quantification in S2G suggests that classification may be difficult and there may be a continuum of densities? Overall, I am not sure to what extent the authors are observing heterogeneity in local fixation and staining, and to what extent they are observing changes in nucleic acid content. To strengthen these conclusions, I first suggest to classify the cores on a second parameter, for example could they do a quantitative measure of the granularity of the core density (dense cores perhaps contain a small number of large densities, while light cores a large number of small densities?) to see if this better separates the core classes? Secondly, the authors could clarify that the differences between NEV treated and untreated cells is observed in independent preparations?

Is "density of gold labellin for CPSF6" per area, or is this (as the figure would suggest) per cluster?

All three "+HIV" panels in Fig 4A show the same region at different magnifications. Instead, different examples should be shown.

The movies of the tomograms should also show the full volume before the colored cones appear, so that readers can really see the data.

Referee #3:

The study by Ay and colleagues provides a detailed description of the morphology and function of incoming HIV-1 capsids in the nucleus of myeloid cells. I was asked to substitute for reviewer 2, who had provided very detailed and constructive comments to a previous version of the manuscript. In my view, the authors added a number of new data sets as well as textual clarification that address all these comments and the manuscript presents a valuable addition to this dynamic area of research.

I have one additional comment that does not relate to those made by reviewer 2. Considering that the supposed shielding of RT products from cGAS sensing is one of the main conclusions of this study and is highlighted in the manuscript title, I find this part relatively underdeveloped. The data reporting induction of cGAMP production and ISG expression are convincing but, as the authors discuss, many questions remain open and could easily be addressed. Experiments that assess to role of cGAS, STING, NONO and PQBP1 in cGAMP and ISG induction and attempts to visualize potential co-localization of these factors with viral structures in the nucleus would be straightforward and an important addition to substantiate the innate sensing aspect of the study.

Referee #1:

The revised paper by Selen Ay et al. is significantly improved. The additional evidence provided in Figure 5 shows the loss of CPSF6/viral condensates upon treatment with the drug PF74. This nicely links an almost complete loss of infectivity to the disappearance of viral condensates.

The revised Figure 6 convincingly demonstrates that loss of viral condensates activates cGAS signalling and induces an innate immune response.

When one considers all the data together, the evidence provided does indeed support the hypothesis that viral condensates promote reverse transcription and shield viral DNA from cGAS sensing before integration.

This concept is novel and interesting and will add a new perspective to the field.

Concerning some remaining limitations, it should be noted that the link between reverse transcription and the viral condensates is correlative, and Figure 6 begs the question if treatment with PF74 at 48h post-infection, which causes the disassembly of the viral condensates, also reduces infectivity. Nonetheless, at this stage I would not delay this paper any further, instead the authors may want to mention this issue at the end of the Discussion.

We thank the referee for the insightful comments. We agree that our experiments correlate the presence of CPSF6 condensates with viral infectivity. Our data show that if PF74 is added at the time of NEV washout, reverse transcription cannot be restored (Fig. 5B), resulting in the absence of infectivity (Fig. 5A). Figure 6 demonstrates that if PF74 is added at 48 hours post-infection, the final products of reverse transcription formed in HIV-1 membraneless organelles (HIV-1-MLOs) are no longer protected due to their disassembly. We thank the referee for his/her comments on Figure 6 about the question if treatment with PF74 at 48h post-infection, which causes the disassembly of the viral condensates, also reduces infectivity. In this study, we discussed that the effect on innate immunity activation at 48 hours post-infection is only partial because it affects only those viruses that are reverse transcribing after 48 hours p.i.. As shown in Figure 6B, these are the products that activate cGAS. Of note, we decided to wait 48 hours post-infection before adding the drug, to ensure that all viruses had entered the nucleus. However, it is possible that if all viruses, once reverse transcribed, are no longer protected by HIV-MLOs from the onset of reverse transcription, the impact on viral replication could be dramatic. We have further clarified our results in the text (line 424-431).

We particularly thank this referee for understanding the high competitiveness of this study and for helping us to publish it, thus making these peer-reviewed results available to the entire scientific community.

Referee

#2:

The manuscript is a valuable contribution and the authors have addressed most of the reviewer comments in a satisfactory manner.

The electron microscopy data showing the presence of nuclear cores surrounded by CPSF6 is generally convincing and the low CA labelling is indeed typical of anti-CA antibodies bound to cores. My

comments only relate to this part of the study.

Are the increased in IN labelling and the decrease in condensate size in the absence of NEV consistent between preparations/experiments? It is not clear whether the data are presented from a single experiment or from more than one. What is the variability in labelling efficiency or on condensate size between experiments? This also goes for the change in CPSF6 labelling. This information is necessary to assess whether the suggested changes in condensates through the lifecycle are real or not.

We thank the referee for his/her comments and appreciation of our paper.

Several biological replicates were performed for the immune-gold labeling. To strengthen the result that condensates are larger in the presence of NEV, we have now analyzed additional samples from independent experiments and independent labeling (New Figure 4E). Below, we report graphs showing the size of HIV-1 MLOs and the number of CPSF6 dots of two independent data sets, which have been merged into the new Figure 4E that is now more statistically significant. Of note, independent data sets indicate that the size of HIV-1-MLOs are larger, and the number of CPSF6 dots is more abundant in the presence of NEV compared to its absence.

Data set 1

Data set 2

Similarly, Fig. 5B presents results obtained using immunofluorescence, showing that there are half as many CPSF6 clusters in cells where the viral cycle is progressing compared to cells treated with NEV, where the viral cycle is halted.

How were cores classified into dense, light and ghosts? Was this done by eye? The quantification in S2G suggests that classification may be difficult and there may be a continuum of densities? Overall, I am not sure to what extent the authors are observing heterogeneity in local fixation and staining, and to what extent they are observing changes in nucleic acid content. To strengthen these conclusions, I first suggest to classify the cores on a second parameter, for example could they do a quantitative measure of the granularity of the core density (dense cores perhaps contain a small number of large densities, while light cores a large number of small densities?) to see if this better separates the core classes? Secondly, the authors could clarify that the differences between NEV treated and untreated cells is observed in independent preparations?

As reviewer #2 highlights, heterogeneity in local fixation, sample drying and staining during sample preparation (classical Tokuyasu protocol) as well as non-uniform shrinkage due to high dose imaging is highly probable in classical TEM imaging. This is the reason a global level analysis was limited in Figure EV3C-D.

The classification of the cores in 3 different classes per site/tomogram took place using the following criteria:

- i. Cores that were not identifiable at all in the projection images (tilt series collected) were classified as Ghosts when later these became observable in the dual-axis tomographic volumes. In these cores their interior presents a similar or close to the pattern/texture and intensity level with their exterior peripheral space.

- ii. Cores that were extremely easy to be identified in the projection images were classified as Dense cores. After reconstruction of the dual-axis tomogram, these were easily identified by eye and had several dark areas that were usually extended to the entire core.
- iii. Finally, intermediate cores that were less easy to identify in the projection images and that after reconstruction they were presenting dark spots in some part(s) of the core were classified as 'Lighter Cores'.

The classification took place by tracing cores manually and classifying them according to the above criteria (Fig. R2.1 cores without tracing are shown, now shown as Fig.EV3E,F). Heterogeneity per sample preparation can be observed between samples.

A

-NEV

+NEV

Fig.R2.1. Nuclear HIV-1-MLOs composition with (in B) and without (in A) nevirapine treatment. In the left column projection images at sites of interest as recorded by the detector. Immunogold labelling (10 nm beads) are labelling the MLOs. In the right column a slice from the middle of the dual axis tomographic volume where HIV cores are more easily observed. Examples of the three different classes as they identified according to the above criteria in dense cores (in magenta boxes), lighter cores (in blue boxes) and ghosts (in yellow boxes).

To further follow the classification uniformly within volumes, and in order to include more parameters like detection of edges and texture, we used the pixel classification method as this has been implemented in ilastik (<https://www.nature.com/articles/s41592-019-0582-9>). Intensity, edge and texture features were selected and a limited input from the user was used: marking one Dense Core in magenta, one Lighter Core in blue/cyan, one or two ghosts in yellow, the nucleus background in grey (Fig. R2.2 and in manuscript it is shown as Appendix Fig.S4E,F), to get the prediction maps. Pixel classification was performed in 2D (one of the middle slices of the tomographic volume). Good prediction maps were possible for the labels of the Dense Cores and the Lighter Cores. Ghosts are usually having intensities near the values of their surroundings and present similar texture. For these reasons they are hard to be identified by eye or pixel classification which gave to this label very high

uncertainty values. Manual tracing was instead possible in tomographic slices due to an existed clear border of conical shape, although not always easy to be followed to its entire length. Probably ghosts are underestimated in our datasets as they are less visible, despite our efforts to complete our datasets as much as possible by performing dual-axis ET. The difficulty in their detection explains why they have never been reported before. They are, in fact, visible only in the dual-axis tomographic volumes (see Fig.4, Fig.EV3A-BE-F, movies 5,6,7,8,9,10,11,12).

Fig. R2.2. Pixel classification predictions for six different sites shown in Fig.EV3E,F. In the left column, initial user input (labels) is shown in the images as thin lines together with the predictions for the rest of the image. The prediction of ghosts was not possible using all features selected in ilastik, and these pixels were most commonly classified as background (background was seeded as a separate longer label, here in white). In the right column pixel classification predictions are overlaid with the manual traced contours of cores, colored accordingly to the underline predictions. Magenta and blue predictions outside the cores are due to immunogold beads that are still contributing to the slice under evaluation, sometimes due to missed manual tracing of cores and on rare cases, like in the top left ghost core of A, that seems like its interior is just released outside (see also Figure EV3B and movies 9 and 10). Scale bars = 100 nm. A) samples without NEV, B) samples with NEV.

In response to the question on changes in nucleic acid content, in rare cases, we did notice the presence of dense material outside the cores. In Fig. R2.3 you can see a zoom on the upper left corner of tomogram from Fig.R2.2.A where dense material is observed next to a ghost, that has been identified by three CPSF6 immunogold beads (marked in purple and computationally erased from the data during reconstruction) and one CA immunogold (marked in green). These data have been included in the manuscript as Figure EV3B and movies 9 and 10.

Fig. R2.3 A ghost core detected by anti-CA antibody (green bead) is decorated with anti-CPSF6 antibody (purple beads). The left panel is unstained, while the right panel is stained with artificial colors: yellow indicates the ghost core, and purple highlights the dark intensity near the core. Scale bar 50nm.

In addition, when possible, we quantified and differentiate conical shapes versus tubular shapes based on viral preparation core shapes. We included this analysis in Fig. EV3G.

Figure EV3

Fig.EV3G. Percentage of HIV core shapes found with and without Nevirapine. HIV cores were included in this analysis only when their longer axis was parallel or near parallel to the tomographic XY plane. The cores were classified as cones in the case an orientation (head to tip) could be assigned easily to

them. The characteristic conical shape of the HIV was easily identifiable in most of the cases. In the rest of the cases, where the head could not be identified, HIV cores were assigned as of tubular shape.

Here we reported an additional example from an independent biological replicate for the identification of the three categories of viral cores:

Without NEV (5 dark, 8 lighter, 19 ghosts)

With NEV (9 dark, 11 lighter, 22 ghosts)

These cores categories could potentially represent different stages of nuclear RT as shown in different field of view and samples analyzed.

Is "density of gold labellin for CPSF6" per area, or is this (as the figure would suggest) per cluster?

We confirm that Figure 4 E bottom panel shows the number of CPSF6 gold particles per condensate.

All three "+HIV" panels in Fig 4A show the same region at different magnifications. Instead, different examples should be shown.

We modified the Figure 4A according to the referee' suggestions and erased the supplementary fig.2B.

The movies of the tomograms should also show the full volume before the colored cones appear, so that readers can really see the data.

We followed the advice of the referee and in the submitted version we added the movies of the tomograms that show the full volume before the coloured cones appear, so that readers can really see the data (movies 7 and 8).

Referee #3:

The study by Ay and colleagues provides a detailed description of the morphology and function of incoming HIV-1 capsids in the nucleus of myeloid cells. I was asked to substitute for reviewer 2, who had provided very detailed and constructive comments to a previous version of the manuscript. In my view, the authors added a number of new data sets as well as textual clarification that address all these comments and the manuscript presents a valuable addition to this dynamic area of research. I have one additional comment that does not relate to those made by reviewer 2. Considering that the supposed shielding of RT products from cGAS sensing is one of the main conclusions of this study and is highlighted in the manuscript title, I find this part relatively underdeveloped. The data reporting induction of cGAMP production and ISG expression are convincing but, as the authors discuss, many questions remain open and could easily be addressed. Experiments that assess to role of cGAS, STING, NONO and PQBP1 in cGAMP and ISG induction and attempts to visualize potential co-localization of these factors with viral structures in the nucleus would be straightforward and an important addition to substantiate the innate sensing aspect of the study.

We thank the reviewer for his/her pertinent comments. We agree with the referee that the understanding of the underlying mechanism to the activation of cGAS after HIV-MLOs disassembly is very important but at the same time this takes time, and it constitutes a new study by itself. Immunofluorescence might not be effective, and this field has several controversies that require considerable time to address them appropriately. This is why we decided to include the following paragraph in the discussion to offer new perspectives (line 436-454): "*Apart from NONO, other host proteins may also play a role in virus detection, including the polyglutamine binding protein 1 (PQBP1), which is required for cGAS recruitment to incoming virus particles (64). The recognition of viral DNA by cGAS appears to be a complex process, as cGAS, NONO, and PQBP1 are physically linked through a*

nuclear ribonucleoprotein complex containing the long non-coding RNA NEAT1(65), which regulates cGAS–STING activation (66). NONO and NEAT1 are located within paraspeckles, and a recent study also identify PQP1 as a component of these nuclear bodies (67), which are closely associated with HIV-1-MLOs (13). As a result, it is conceivable that the disruption of HIV-1-MLOs could facilitate the binding of complex components, leading to the activation of the innate immune response. Notably, it has been observed that the dissociation of cGAS from the nucleosome, facilitated by the DNA-repair complex MRN11, might contribute to this process by enabling the mobilization of cGAS from genomic DNA to viral DNA in mitotic cells (68). Interestingly, in macrophages, where cGAS is crucial for fighting infections, cGAS predominantly resides in the nucleus (62), despite the fact that these cells are not undergoing mitosis. That said, studying the underlying mechanisms of cGAS activation following the disassembly of HIV-1-MLOs presents significant complexity, requiring careful evaluation of multiple factors. One crucial aspect is the need to track endogenous proteins, which calls for further detailed investigation. While our study provides valuable insights, it also opens new avenues for more advanced research into the mechanisms of cGAS activation during the disassembly process of HIV-1-MLOs.”.

Of note, the title of our study indicates that we found that the HIV genome is protected by HIV-1-MLOs from cGAS and does not mention the study of the underlying mechanism:

“In Vivo HIV-1 Nuclear Condensates Safeguard Against cGAS and License Reverse Transcription.”

We completely agree with the referee's perspective and will continue to perform additional experiments on this topic. However, the data obtained in a few months will be preliminary to include in this study, and we fear that it may weaken the study rather than strengthen it.

We hope the referee can appreciate and understand our perspective.

Dear Francesca,

Thank you for submitting a revised version of your manuscript. I sincerely apologise for the protracted assessment process due to the high manuscript submission rate to our office at the moment. We have now received input from one of the original reviewers, who finds that their previous concerns have been addressed satisfactorily and now recommends implementation of minor textual edits before acceptance of the manuscript.

Additionally, there remain a few editorial points that need addressing before I can extend official acceptance of the manuscript:

1. Please check if the email addresses provided for the authors Jean-Sebastien Diana (diana@pasteur.fr) and Viviana Scoca (scoca@pasteur.fr) are correct, as the emails sent to them were returned.
2. Please ensure that the funding information is correct and identical both in the manuscript and our online system.
3. Please submit up to five keywords.
4. Please submit a complete author checklist, which you can download from our author guidelines (<https://www.embopress.org/pb-assets/embo-site/EMBO%20Press%20Author%20Checklist-1642513524327.xlsx>). Please insert information in the checklist that is also reflected in the manuscript. The completed author checklist will also be part of the Review Process File.
5. Please make sure that the order of the sections in the manuscript is as follows: abstract, introduction, results, discussion, materials & methods, data availability section, acknowledgments, disclosure statement and competing interests, references, main figure legends, tables, expanded figure legends.
6. Please upload the main and EV figures as individual production quality figure files in the .eps, .tif, or .jpg format (one file per figure). Please note that all panels should fit into a single A4 page.
7. All Materials and Methods need to be described in the main text using our 'Structured Methods' format. According to this format, the Methods section includes a Reagents and Tools Table (listing key reagents, experimental models, software and relevant equipment and including their sources and relevant identifiers). The aim is to facilitate adoption of the methodologies across labs. Please download and fill our Reagents and Tools Table template (.docx), which you can find in our author guidelines:
<https://www.embopress.org/page/journal/14602075/authorguide#structuredmethods>
An example of a Method paper with Structured Methods can be found here:
<https://www.embopress.org/doi/10.15252/msb.20178071>
When submitting your revised manuscript, please upload it as a separate file choosing the file type "Reagent Table". The information currently provided in Key Resources Table could be adapted to this format.
8. Please add a "Disclosure and competing interests statement" section. Further info:
<https://www.embopress.org/page/journal/14602075/authorguide#conflictsofinterest>.
9. Please rename the movies into Movie EV1-EV12 and update the callouts accordingly. The legends should be removed from the manuscript text file and zipped with each movie file. Further information is available here:
<https://www.embopress.org/page/journal/14602075/authorguide#expandedview>
10. Please update references according to The EMBO Journal style - where there are more than 10 authors on a paper, the first 10 should be listed, followed by 'et al.' Please see further information here:
<https://www.embopress.org/page/journal/14602075/authorguide#referencesformat>
11. In our standard image integrity check, we noted a reuse of the image panels between the figure panels listed below:
 - Fig. 4A, +HIV - Appendix Fig. S4B, +HIV
 - Fig. 4F - Fig. EV3E, -NEV
 - Fig. 4F - Fig. EV3F, +NEV
 - Fig. 4F - Appendix Fig S4E-F, +/-NEV
 - Fig. EV3A - Fig. EV3E, -NEV
 - Fig. EV3A - Fig. EV3F, +NEVWhile we allow figure panel reuse in the manuscript when clearly indicated in the figure legends (currently mentioned in the legend for Appendix Fig. S4B with regard to Fig. EV3), most of electron microscopy images presented appear to be derived from a small number of field views. Since this reuse appears excessive, please replace some of the images with independent data.
12. Additionally, we noted reuse of figure panels in Fig 5A +NEV, 3 d.p.i. and 7 d.p.i. Please correct and provide a clarification.
13. In our routine textual plagiarism check, we noted a sentence (lines 154-156 and in the attachment) that appears identical to an earlier publication. Please rephrase.
14. Our data editors have flagged the following issues in figure legends that need correcting:
 - Please define the annotated p values */** as well as provide the exact p-values for the same in the legend of figure 4d-e; as appropriate.
 - Please provide the exact p values in the legends of figures 1b; 2c, f; 5b, c; 6c-f; EV 4.
 - Please note that in figures 1b; 2c; 5b; 6b there is a mismatch between the annotated p values in the figure legend and the annotated p values in the figure file that should be corrected.
 - Please provide information on the number and nature of replicates in the legends of figures 1b-c; 2c; 3b (left); 4d-e; 6a-f; EV 1b; EV 4.
 - Please define the error bars in the legends of figures 1b-c; 2c; 3b; 4d; 5b; 6a, e-f; EV 1b; EV 4.

- Please note that the scale bar is missing for figure 4g.
- Please note that scale bar and its definition are missing for figures EV 3c, e-f.
- Please define the yellow arrowheads in the legend of figure 5c.

With best wishes,

leva

leva Gailite, PhD
Senior Scientific Editor
The EMBO Journal
Meyerhofstrasse 1
D-69117 Heidelberg
Tel: +4962218891309
i.gailite@embojournal.org

We realize that it is difficult to revise to a specific deadline. In the interest of protecting the conceptual advance provided by the work, we recommend a revision within 3 months (7th Jan 2025). Please discuss the revision progress ahead of this time with the editor if you require more time to complete the revisions.

Referee #2:

The manuscript is improved. I have only minor requests for changes to be made before publication.

Please make clear in the text, legend or methods that the observed differences in immunogold labelling (especially for integrase in 4D), have been observed in more than one experiment.

Regarding the size and the CPSF6 labelling of the condensates: Please remove the pie charts from figures 4E, and please also remove the statements about the numbers smaller or larger than an arbitrary threshold from the text (lines 218-219 and 223-224). Comparison to arbitrary thresholds is not an appropriate way to present statistical data. Instead, I think it would be more helpful to include numbers in the text. Mean and standard deviation for the +/- drug option would be one possibility followed by t-test pvalue.

The new analysis of the appearance of the cores is better. The differences between the two conditions are subtle, and the authors should be sure that they are happy that they have been cautious enough in discussing differences.

Point by point

Dear Francesca,

Thank you for submitting a revised version of your manuscript. I sincerely apologise for the protracted assessment process due to the high manuscript submission rate to our office at the moment. We have now received input from one of the original reviewers, who finds that their previous concerns have been addressed satisfactorily and now recommends implementation of minor textual edits before acceptance of the manuscript.

Dear editor Dr. Ieva Gailite,

We thank you for your support and guidance during the revision of our manuscript. Please see below the change that we made together with our answers to the comments.

Additionally, there remain a few editorial points that need addressing before I can extend official acceptance of the manuscript:

1. Please check if the email addresses provided for the authors Jean-Sebastien Diana (diana@pasteur.fr) and Viviana Scoca (scoca@pasteur.fr) are correct, as the emails sent to them were returned.

We used the new emails of the co-authors.

2. Please ensure that the funding information is correct and identical both in the manuscript and our online system.

Yes, we did it.

3. Please submit up to five keywords.

Yes, we did it.

4. Please submit a complete author checklist, which you can download from our author guidelines (<https://www.embopress.org/pb-assets/embo-site/EMBO%20Press%20Author%20Checklist-1642513524327.xlsx>). Please insert information in the checklist that is also reflected in the manuscript. The completed author checklist will also be part of the Review Process File.

Yes, we did it.

5. Please make sure that the order of the sections in the manuscript is as follows: abstract, introduction, results, discussion, materials & methods, data availability section, acknowledgments, disclosure statement and competing interests, references, main figure legends, tables, expanded figure legends.

Yes, we did it.

6. Please upload the main and EV figures as individual production quality figure files in the .eps, .tif, or .jpg format (one file per figure). Please note that all panels should fit into a single A4 page.

Yes, we did it.

7. All Materials and Methods need to be described in the main text using our 'Structured Methods' format. According to this format, the Methods section includes a Reagents and Tools Table (listing key reagents, experimental models, software and relevant equipment and including their sources and relevant identifiers). The aim is to facilitate adoption of the methodologies across labs. Please download and fill our Reagents and Tools Table template (.docx), which you can find in our author guidelines:

<https://www.embopress.org/page/journal/14602075/authorguide#structuredmethods>

When submitting your revised manuscript, please upload it as a separate file choosing the file type "Reagent Table". The information currently provided in Key Resources Table could be adapted to this format.

Yes, we did it.

8. Please add a "Disclosure and competing interests statement" section. Further info:

<https://www.embopress.org/page/journal/14602075/authorguide#conflictsofinterest>.

Yes, we did it.

9. Please rename the movies into Movie EV1-EV12 and update the callouts accordingly. The legends should be removed from the manuscript text file and zipped with each movie file. Further information is available here:

Yes, we did it.

10. Please update references according to The EMBO Journal style - where there are more than 10 authors on a paper, the first 10 should be listed, followed by 'et al.' Please see further information here: <https://www.embopress.org/page/journal/14602075/authorguide#referencesformat>

Yes, we did it.

11. In our standard image integrity check, we noted a reuse of the image panels between the figure panels listed below:

- Fig. 4A, +HIV - Appendix Fig. S4B, +HIV

In the new submission, there is no reuse of images. Initially, we aimed to present different magnifications, but in response to your feedback, we have modified Appendix Fig. S4B as requested. Additionally, we have included several extra EM images in the updated version of Fig. S4B in the appendix.

- Fig. 4F - Fig. EV3E, -NEV

- Fig. 4F - Fig. EV3F, +NEV

Figures EV3E and EV3F were intended to compare two analytical approaches. To better present the results, we have now revised the figures according to the journal's guidelines, and they are updated as Figures EV3A and EV3B.

- Fig. 4F - Appendix Fig S4E-F, +/-NEV

We incorporated data from the previous Fig. S4E-F into the updated Fig. EV3A and Fig. EV3B. This representation will help readers better understand the imaging analysis we performed on the tomograms.

- Fig. EV3A - Fig. EV3E, -NEV

- Fig. EV3A - Fig. EV3F, +NEV

The previous Fig. EV3A has now been replaced with the new Fig. S4E in the appendix.

While we allow figure panel reuse in the manuscript when clearly indicated in the figure legends (currently mentioned in the legend for Appendix Fig. S4B with regard to Fig. EV3), most of electron microscopy images presented appear to be derived from a small number of field views. Since this reuse appears excessive, please replace some of the images with independent data.

We avoid the reuse of images in the revised version of the paper and included additional electron microscopy images, as indicated in the aforementioned figures.

12. Additionally, we noted reuse of figure panels in Fig 5A +NEV, 3 d.p.i. and 7 d.p.i. Please correct and provide a clarification.

In the new submission, there is no reuse of images. We understood that there was a reuse of images between the Fig.5A and the Fig.S5 in appendix. Initially, we aimed to show a large field of view of what is shown in Fig.5A. We apologize and we have now replaced it with additional fields of views, as shown in the new Fig. S5 in the appendix.

13. In our routine textual plagiarism check, we noted a sentence (lines 154-156 and in the attachment) that appears identical to an earlier publication. Please rephrase.

Sorry for that we rephrased it as follows: "Xenotransplantation of human CD34+ hematopoietic stem cell progenitors into BRGS hosts results in the development of various human lymphocyte populations (B, T, and NK cells) as well as myeloid cells".

14. Our data editors have flagged the following issues in figure legends that need correcting:

- Please define the annotated p values */** as well as provide the exact p-values for the same in the legend of figure 4d-e; as appropriate.

- Please provide the exact p values in the legends of figures 1b; 2c, f; 5b, c; 6c-f; EV 4.
- Please note that in figures 1b; 2c; 5b; 6b there is a mismatch between the annotated p values in the figure legend and the annotated p values in the figure file that should be corrected.
- Please provide information on the number and nature of replicates in the legends of figures 1b-c; 2c; 3b (left); 4d-e; 6a-f; EV 1b; EV 4.
- Please define the error bars in the legends of figures 1b-c; 2c; 3b; 4d; 5b; 6a, e-f; EV 1b; EV 4.
- Please note that the scale bar is missing for figure 4g.
- Please note that scale bar and its definition are missing for figures EV 3c, e-f.
- Please define the yellow arrowheads in the legend of figure 5c.

We have considered all the feedback and revised the text accordingly.

With best wishes,

Ieva

We would like to thank you once again for your guidance in improving our manuscript, which we are confident that it now meets the standards of EMBO Journal.

Ieva Gailite, PhD
Senior Scientific Editor
The EMBO Journal
Meyerohofstrasse 1
D-69117 Heidelberg
Tel: +4962218891309
i.gailite@embojournal.org

Further information is available in our Guide For Authors:

We realize that it is difficult to revise to a specific deadline. In the interest of protecting the conceptual advance provided by the work, we recommend a revision within 3 months (7th Jan 2025). Please discuss the revision progress ahead of this time with the editor if you require more time to complete the revisions. Use the link below to submit your revision:

Referee #2:

The manuscript is improved. I have only minor requests for changes to be made before publication.

We thank the reviewer for his/her high professionalism in helping us to improve our manuscript.

Please make clear in the text, legend or methods that the observed differences in immunogold labelling (especially for integrase in 4D), have been observed in more than one experiment.

We specified it in the legend of the Figure 4D.

Regarding the size and the CPSF6 labelling of the condensates: Please remove the pie charts from figures 4E, and please also remove the statements about the numbers smaller or larger than an arbitrary threshold from the text (lines 218-219 and 223-224). Comparison to arbitrary thresholds is not an appropriate way to present statistical data. Instead, I think it would be more helpful to include numbers in the text. Mean and standard deviation for the +/- drug option would be one possibility followed by t-test pvalue.

We modified the Figure 4E as indicated by the reviewer and included numbers with p-values in the main text.

The new analysis of the appearance of the cores is better. The differences between the two conditions are subtle, and the authors should be sure that they are happy that they have been cautious enough in discussing differences.

We thank the referee for his/her comment.

We carefully discussed our results and included the following sentences in the text: "Thus, we quantified and **classified the most visible core shapes in our tomograms** using a combination of pixel classification with manual tracing" (line 253-254). "Interestingly, in NEV-treated cells, the cores appeared more intact (dense + lighter cores) compared to untreated cells (52.1% vs. 36%), **although all three core categories were observed in both conditions.**" (line 415-416).

Dear Francesca,

Thank you for addressing most of the remaining editorial issues. I apologise for the delay in the processing of your manuscript due to school holidays here in Germany. Upon checking your manuscript, I am afraid that I noticed a few remaining issues that need to be fixed before I can accept it for publication:

1. I apologise for the error in the point 12 in the previous decision letter. The image reuse was in figure EV2A, not figure 5A. As mentioned previously, bright field images for the panels +NEV, 3 d.p.i. and 7 d.p.i are identical. This should be corrected in the final version.

2. I would like to propose minor textual edits in the synopsis and abstract. I have also written a short blurb that will accompany the title of your manuscript in our online table of contents. Please take a look at the text below and in the attached file and let me know if any edits or corrections are needed.

Blurb:

HIV-1 membraneless organelles are the main sites of viral RNA reverse transcription in the host cell nucleus.

Synopsis:

In vitro infection studies have shown that HIV-1 entry into the nucleus triggers the formation of dynamic structures called HIV-1 membraneless organelles (HIV-1-MLOs). This study reveals their existence in vivo and highlights their multifaceted role in the post-nuclear viral entry steps.

- Viral nuclear condensates form in a humanized mouse model of HIV-1 infection.
- Viral nuclear condensates host distinct viral core structures for several days post-infection.
- Nuclear condensates are the main sites of nuclear reverse transcription of the HIV-1 genome.
- Viral nuclear condensates protect the newly synthesized viral DNA from recognition by cGAS.

Thank you again for giving us the chance to consider your manuscript for The EMBO Journal. I look forward to receiving your input on these final points.

With best wishes,

Ieva

We realize that it is difficult to revise to a specific deadline. In the interest of protecting the conceptual advance provided by the work, we recommend a revision within 3 months (3rd Feb 2025). Please discuss the revision progress ahead of this time with the editor if you require more time to complete the revisions.

Dear Ieva,

Thank you for your efficient and dedicated handling of our study. We sincerely apologize for the error in Figure EV2, where we inadvertently duplicated the same panel while preparing the image. We also misunderstood the comment in the point-by-point review referring to Figure 5, which led us to overlook our previous mistake. We have now uploaded the corrected version of Figure EV2.

Additionally, we have prepared higher-resolution images for Figure 2 and Figure 3, which have been included in the updated version of the manuscript.

We are grateful for your valuable input on the manuscript, synopsis, and blurb. We appreciate all your comments and suggested changes, and we have made a very few modifications to better convey the study's message.

Please find the revised text with a few changes marked in red and comments included in the attached document.

We have submitted the manuscript incorporating the requested changes and sincerely hope that this version meets your expectations for publication.

Thank you once again for your careful consideration and support.

Sincerely yours,

Francesca

Dear Francesca,

Thank you for fixing these final issues and for your input on the textual edits. I am happy to incorporate your suggested changes in the final text. I am now pleased to inform you that your manuscript has been accepted for publication in the EMBO Journal.

If you have any questions, please do not hesitate to contact the Editorial Office. Thank you for your contribution to The EMBO Journal and congratulations on a nice study!

With best wishes,

Ieva
